# Test-Time Learning of Causal Structure from Interventional Data

Wei Chen[* 1 2]   Rui Ding[† 3]   Bojun Huang[4]
Yang Zhang[5]   Qiang Fu[3]   Yuxuan Liang[† 1]   Shi Han[3]   Dongmei Zhang[3]

## Abstract

Supervised Causal Learning has shown promise in causal discovery, yet it often struggles with generalization across diverse interventional settings, particularly when intervention targets are unknown. To address this, we propose `TICL` (Test-time Interventional Causal Learning), a novel method that synergizes Test-Time Training with Joint Causal Inference (JCI). Specifically, we design a self-augmentation strategy to generate instance-specific training data at test time, effectively avoiding distribution shifts. Furthermore, by integrating JCI, we developed a PC-inspired two-phase supervised learning scheme, which effectively leverages self-augmented data while ensuring theoretical identifiability. Extensive experiments on bnlearn benchmarks demonstrate `TICL`'s superiority in multiple aspects of causal discovery and intervention target detection.

## 1. Introduction

*Causal discovery*, the identification of causal relations from data, underpins modern scientific progress (Reichenbach, 1956). Recently, Supervised Causal Learning (SCL) has emerged as a promising modeling paradigm for causal discovery (Lorch et al., 2022; Ke et al., 2023a;b; Wu et al., 2025). By training on synthetic datasets spanning diverse causal structures, SCL models aim to operationalize causal discovery as a structured prediction task where empirical accuracy under realistic conditions is the central objective.

Despite these successes, existing SCL works has primarily focused on *observational data*. However, experimentation remains the gold standard for causality (Hill, 1952; Pearl,

---

[*]work done during his internship at MSRA. [1]The Hong Kong University of Science and Technology (Guangzhou) [2]The Hong Kong University of Science and Technology [3]Microsoft Research [4]Sony Research [5]National University of Singapore. Correspondence to: Rui Ding <juding@microsoft.com>, Yuxuan Liang.

*Proceedings of the 43rd International Conference on Machine Learning*, Seoul, South Korea. PMLR 306, 2026. Copyright 2026 by the author(s).

2009). *Interventional data*, in particular, allows for the identification of more causal relations with fewer assumptions (Hauser & Bühlmann, 2012). Applying SCL under interventional settings is thus of considerable importance.

Unfortunately, the SCL paradigm faces two main challenges when interventions are involved, particularly if the specific intervention actions are unknown, a common scenario in real world third-party experiments (Eaton & Murphy, 2007a):

*Challenge I: Versatility.* Different from the observational setting (which enjoys a simple and uniform problem formulation), interventional settings may vary widely, including factors such as known / unknown targets, hard / soft interventions, and single / multiple-variable interventions. This diversity complicates learning versatile models, and existing SCL methods (Lorch et al., 2022; Ke et al., 2023b) are typically limited to specific settings (*e.g.*, hard + known), limiting their applicability to diverse practical scenarios.

*Challenge II: Generalizability.* As a supervised learning approach, SCL methods fundamentally face the core challenge of generalization – models trained on data following a specific distribution (*e.g.*, one generated by a configured simulator) may perform poorly on real-world test data. The formulation diversity as mentioned above may further exacerbate the generalization issue in the interventional setting.

To circumvent the generalization bottleneck, we investigate Test-Time Training (TTT) paradigm (Sun, 2023), an emerging that trains instance-specific models at test/inference time in an on-the-fly manner rather than seeking a single universally generalizable model. Thus, we explore three key questions: (1) When should training data be acquired? (2) What kind of training data is effective for SCL? (3) How can such data be acquired in our setting? Specifically,

**When:** We identify and exploit the window between accessing test data and performing the actual inference. In this window of timing, we generate free training data via test data and train the model specifically for the final testing.

**What:** We observe that the *posterior estimation* of causal graphs, $P(\mathcal{G}|\mathcal{D})$, provides highly effective training data. By sampling graphs $(\mathcal{G}_1, \ldots, \mathcal{G}_n)$ from the posterior and generating compatible datasets via forward sampling, we create paired instances $\langle \mathcal{G}_i, \mathcal{D}_i \rangle$ for *self-augmentation* (Fig. 2 Left).

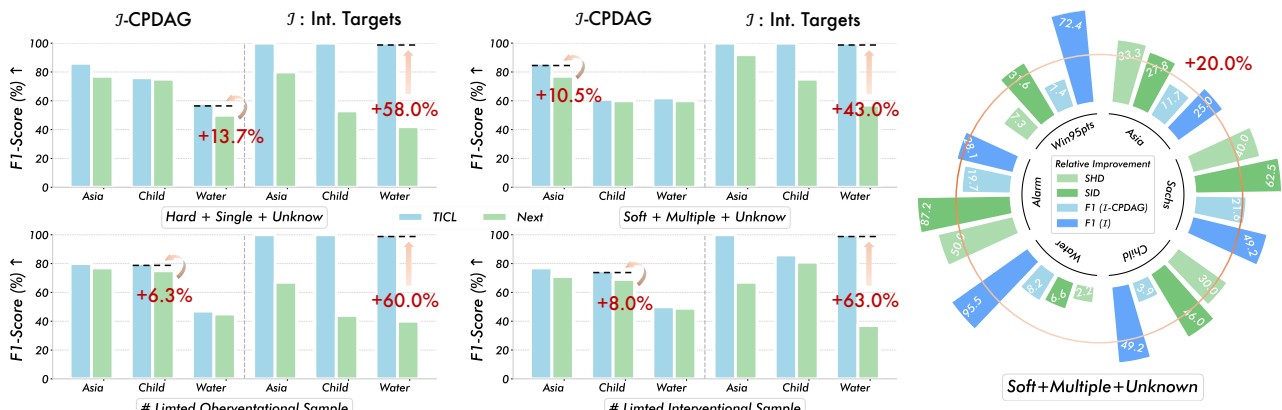

*Figure 1.* (Empirical dominance across interventional SCL tasks). `TICL` consistently outperforms all SoTA methods on both $\mathcal{I}$-CPDAG discovery and intervention targets detection, across diverse intervention families and limited-sample regimes.

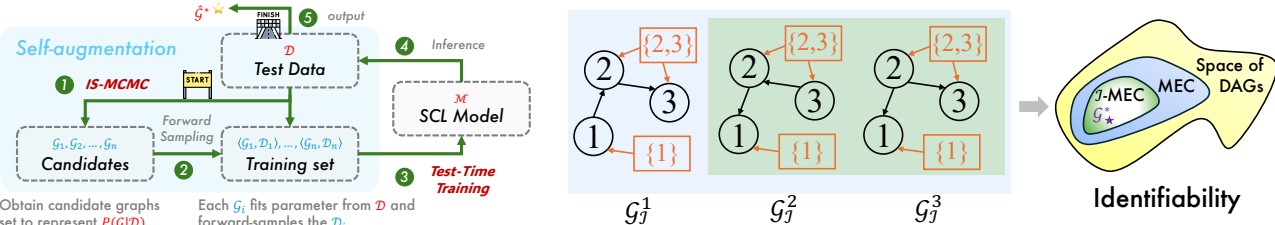

*Figure 2.* Left: The workflow of test-time learning of causal structure. Right: An identifiability example.

**How:** We introduce the IS-MCMC algorithm, which constructs a Markov chain over the augmented graph structure space with *intervention constraints* and performs multiple optimizations. This enables efficient sampling from the posterior $P(\mathcal{G}|\mathcal{D})$ to fuel the self-augmentation process.

To address the versatility challenge, we advocate for the adoption of Joint Causal Inference (JCI) framework (Mooij et al., 2020). JCI unifies diverse interventional settings by pooling datasets into an augmented representation, allowing algorithms to treat them as observational. While JCI simplifies the input formulation, adapting SCL to this framework still requires to answer two critical questions: (1) What should be the appropriate learning target? (2) How should the learning process be designed? Specifically,

**What:** We prioritize theoretical identifiability, emphasizing that models should predict identifiable causal structures, *i.e.*, $\mathcal{I}$-CPDAG. This ensures the model only predicts relations that are structurally identifiable from the training data.

**How:** We introduce the two-phase SCL algorithm. Inspired by the PC algorithm (Spirtes & Glymour, 1991), we focus on identifying the identifiable components of the $\mathcal{I}$-CPDAG, namely the skeleton and v-structures. This approach ensures asymptotic correctness while promoting systematic feature characterization and improved classification mechanisms.

By incorporating these ideas together, we propose `TICL`, a novel method for interventional causal discovery, focusing on discrete data to illustrate its effectiveness. Specifically,

`TICL` establishes a TTT + JCI paradigm, employing a self-augmentation algorithm for training data acquisition and a two-phase SCL algorithm for designing training targets.

**Evaluation Highlights**: Figure 1 summarizes our main empirical finding: `TICL` establishes a new state-of-the-art for interventional SCL. These improvements are hold consistently across diverse benchmark causal graphs and multiple evaluation criteria. In summary, our key contributions are:

- We propose `TICL`, a novel method for SCL under interventions. Our `TICL` consistently outperforms existing state-of-the-art methods in experiments on two tasks: causal discovery and intervention target detection.

- We introduce a specific test-time training technique to the SCL domain, which (self-)augments the training data by efficient sampling causal graphs from the posterior distribution via an optimized IS-MCMC process.

- Our systematic study highlight JCI as a promising direction for unifying interventional settings in SCL. Within this framework, our two-phase learning algorithm provides a concrete solution for what and how to learn.

## 2. Preliminaries

### 2.1. Interventional Causal Discovery

A *Causal Graphical Model* (CGM) $\mathcal{M} =< \mathcal{G}, P >$ over $d$ random variables $\mathbf{X} := \{X_1, \ldots, X_d\}$ comprises: *(i)* a directed acyclic graph (DAG) $\mathcal{G}$ with nodes corresponding to the variables $\mathbf{X}$ and edges encoding direct causal

relations between them, and *(ii)* a joint probability distribution $P_{\mathbf{X}}$ that is *Markov compatible* with $\mathcal{G}$, *i.e.*, $P_{\mathbf{X}} = \prod_{i=1}^{d} P(X_i|pa(X_i))$, where $pa(X_i)$ are the parents of $X_i$.

Given an unknown causal model $< \mathcal{G}, P >$, the *Observational Causal Discovery* problem asks to infer about the causal graph $\mathcal{G}$ based on an i.i.d. sample of the joint distribution $P_{\mathbf{X}}$. It is however well known that under the observational setting, in the general case, we can only identify a causal graph up to its Markov Equivalent Class (MEC), even if with the *faithfulness assumption* (that $X \perp_P Y|Z \Rightarrow X \perp_{\mathcal{G}} Y|Z$) and with the Markov compatibility condition , where the MEC of the $\mathcal{G}$ is the set of graphs with the same *skeleton* and *v-structure*s with $\mathcal{G}$ (Verma & Pearl, 1990).

The identifiability limit in the observational setting can be effectively mitigated by conditioning the observations upon interventions, *i.e.*, actions that purposefully perturb the causal model. In general, an intervention can apply to a subset of variables $I \subset \mathbf{X}$, called the *intervention targets*, which can either be a single variable or contain multiple variables. For each target variable $X_i$ in $I$, the intervention replaces the conditional distribution $P(X_i|pa(X_i))$ with a new one: $P^{(I)}(X_i|pa(X_i))$. The joint distribution after the intervention thus becomes $P_{\mathbf{X}}^{(I)} = \prod_{i \notin I} P(X_i|pa(X_i)) \prod_{j \in I} P^{(I)}(X_j|pa(X_j))$. The intervention is called a *hard* (*a.k.a. perfect, structural*) intervention (Eaton & Murphy, 2007b) if it eliminates the intervention targets' causal dependence on their parents entirely, *i.e.*, if $P^{(I)}(X_i|pa(X_i)) = P^{(I)}(X_i), \forall X_i \in I$; otherwise it is called a *soft* (*a.k.a. imperfect, parametric*) *intervention* (Tian & Pearl, 2001) as it maintains at least part of the original causal dependence. It is an *unknown intervention* if the target set $I$ is unknown.

The *Interventional Causal Discovery* problem asks to infer about the causal graph $\mathcal{G}$ based on a collection of data samples $\mathcal{D} = \{D_0, D_1, \dots, D_K\}$, each obtained under a different intervention (Cooper & Yoo, 1999; Hauser & Bühlmann, 2012; Brouillard et al., 2020). Let $\mathcal{I} := \{I_0, I_1, \dots, I_K\}$ be the intervention targets for each of the interventions here. $\mathcal{I}$ is called the *intervention family* of the dataset $\mathcal{D}$. It is often useful to obtain $D_0$ as a sample of the observational distribution $P_{\mathbf{X}}$ without any actual intervention (Ke et al., 2023a), and in this case we denote $I_0 = \emptyset$. In this paper, we consider the situation that the interventions are unknown, and we need to infer both about the causal graph $\mathcal{G}$ and about the intervention family $\mathcal{I}$ from the given data collection $\mathcal{D}$.

## 2.2. Joint Causal Inference Framework

Our method is based on the JCI framework (Mooij et al., 2020), which reduces an interventional causal discovery problem into an observational causal discovery problem over an augmented causal model. The basic idea is to treat a data sample under intervention as an observation under

an imposed condition. The data-sample collection $\mathcal{D}$ in the intervention setting can then be seen as a single data sample under a variety of observation conditions. More specifically,

*Augmented Data.* Given data-sample collection $\mathcal{D} = \{D_0, D_1, \dots, D_K\}$ obtained under intervention family $\mathcal{I} := \{I_0, I_1, \dots, I_K\}$, an *augmented data* $D_{\mathcal{I}}$ can be constructed by stacking the $K+1$ data samples in $\mathcal{D}$ together by rows, then appending $K$ columns, each corresponding to a newly added *environment variable* (or *intervention variable*) $X_{I_k}$ which takes binary value and $X_{I_k} = 1$ in and only in data points (= rows) originally from $D_k$. Please refer to the data table at the top of Figure. 3 for an example.

*Augmented Graph.* Accordingly, an *augmented causal graph* $\mathcal{G}_{\mathcal{I}}$ can be constructed by adding nodes $X_{I_k}$, and adding edges $X_{I_k} \to X_i$ for all $X_i \in I_k$ if the intervention targets $I_k$ are known. In the $\mathcal{G}_{\mathcal{I}}$, nodes corresponding to the original variables $X_i \in \mathbf{X}$ are called *system nodes*, and those corresponding to $X_{I_k}$ are called *environment nodes*.

In the JCI framework, we first convert data collection $\mathcal{D}$ to augmented data $D_{\mathcal{I}}$, optionally incorporating edges from known interventions regime $\mathcal{I}$, then infer the augmented graph $\mathcal{G}_{\mathcal{I}}$, which encodes both the information of the original causal graph $\mathcal{G}$ over the system variables $\mathbf{X}$, and reveals unknown intervention targets as edges from environment nodes to system nodes are identified.

Moreover, the causal inference over the augmented graph in the JCI framework may be facilitated by some *a priori* constraints when prior knowledge / assumptions about the interventions are available. For example, *exogeneity* assumption requires no causal edges from any system variable $X_i \in \mathbf{X}$ to environment nodes. Similarly, *complete randomized context* and *generic context* assumption assume no confounding between system and environment nodes, and no causation between environment nodes, respectively. For more details, please refer to the paper (Mooij et al., 2020).

## 2.3. Identifiability with Intervention Data

Two causal DAGs $\mathcal{G}_1$ and $\mathcal{G}_2$ are indistinguishable under an intervention family $\mathcal{I}$ with $I_0 = \emptyset$ if and only if their interventional graphs $\mathcal{G}_{I_k}^1$ and $\mathcal{G}_{I_k}^2$ have the same skeleton and v-structures for all $I_k \in \mathcal{I}$ (Yang et al., 2018). It is thus clear that with interventional data, we may identify the true causal graph up to a smaller equivalence class – called the $\mathcal{I}$-MEC. Fig. 2 (Right) gives a simple example that illustrates how interventions can help with causal identifiability: The three DAGs on the left belong to the same MEC. Considering the intervention family $\mathcal{I} = \{\emptyset, \{1\}, \{2, 3\}\}$, $G_{\mathcal{I}}^1$ is not in the same $\mathcal{I}$-MEC as $G_{\mathcal{I}}^2$ and $G_{\mathcal{I}}^3$ due to the absence of the v-structure $X_2 \to X_1 \leftarrow \{1\}$. The schematic on the right further elucidates the relationships among the true DAG $\mathcal{G}^{\star}$, the $\mathcal{I}$-MEC, the MEC, and the space of all possible DAGs.

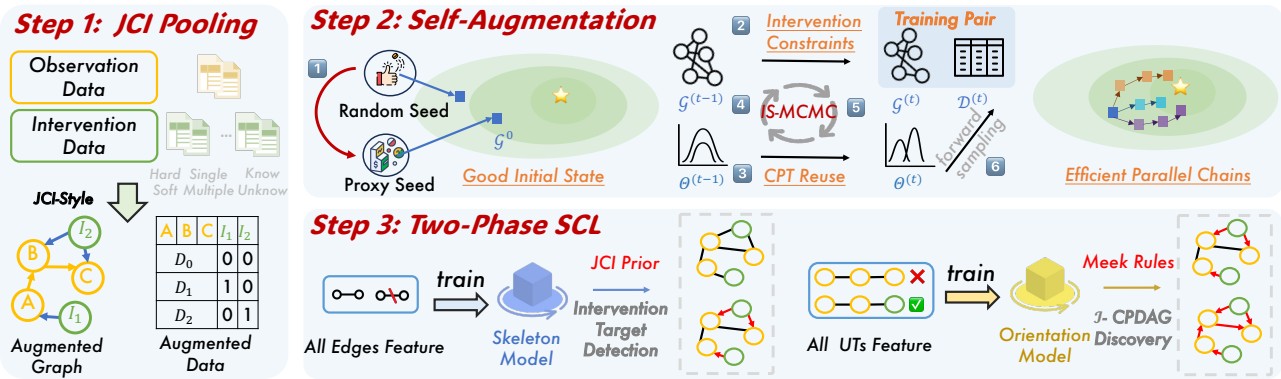

*Figure 3.* The overall workflow of `TICL` for test-time learning of causal structure from interventional data.

# 3. Test-Time Learning of Causal Structure

In this section, we present the details of `TICL` method (see Figure 3). We first formally summarize the problem and outline its solution, then delve into the two essential components of our `TICL`: *self-augmentation strategy* and *two-phase supervised causal structure learning*.

**Problem Summary.** Given a data collection $\mathcal{D}$ generated by an unknown causal model $(\mathcal{G}, P)$ under an intervention family $\mathcal{I}$ (potentially multi-variable, soft, and unknown), we adopt standard causal discovery assumptions: i) $P$ is *Markovian* and *faithful w.r.t* the causal graph and *causal sufficiency*, and ii) *exogeneity*, *complete-randomized-context*, and *generic-context* property for the intervention family. We aim to predict all causal relations entailed by the given data $\mathcal{D}$ (subject to the above assumptions), which correspond to the invariant causal structures in the $\mathcal{I}$-MEC set of the causal graph $\mathcal{G}$ behind $\mathcal{D}$, as explained in Section 2.3. Such invariant causal structures can be computationally encapsulated as a partial DAG, called the *Interventional-Complete Partial Directed Acyclic Graph* ($\mathcal{I}$-CPDAG), in which each directed edge indicates an invariant causal relation in the $\mathcal{I}$-MEC set. Besides the $\mathcal{I}$-CPDAG Discovery task thus discussed, we also want to identify the unknown intervention targets in the intervention family $\mathcal{I}$ (*Intervention Target Detection*).

**Solution Outline.** `TICL` proceeds in three steps: *(1) JCI Pooling:* First, we convert the given data collection $\mathcal{D}$ into an augmented data $\mathcal{D}_{\mathcal{I}}$ following the JCI protocol. *(2) Self-Augmentation:* Then, we use the self-augmentation strategy to construct a Markov chain over the space of augmented graph structures constrained by interventions, where each $\mathcal{G}_i$ fits the parameters in $\mathcal{D}_{\mathcal{I}}$, and forward sampling to get $\mathcal{D}_i$ as the training instance. *(3) Two-Phase SCL:* Last, we extract edge and triplet features of training data to enable two-phase SCL for the skeleton and orientation models. The former leverages JCI priors (*i.e.*, prior knowledge of environment-system variable relations) to identify unknown targets, while the latter employs Meek Rules to enhance causal discovery.

## 3.1. Training Data Acquisition via Self-Augmentation

We propose leveraging the posterior estimation of causal graphs, $P(\mathcal{G}_{\mathcal{I}}|\mathcal{D}_{\mathcal{I}})$ to generate training data. Specifically, we sample causal graphs $(\mathcal{G}_1, \ldots, \mathcal{G}_n)$ from $P(\mathcal{G}_{\mathcal{I}}|\mathcal{D}_{\mathcal{I}})$ using a tailored Markov Chain Monte Carlo (MCMC) designed for interventional data. For each $\mathcal{G}_i$, the parameters governing the conditional distributions of variables, given their parents in $\mathcal{G}_i$, are re-estimated using the dataset $\mathcal{D}$. Forward sampling is then applied with these parameters to generate a dataset $\mathcal{D}_i$ compatible with $\mathcal{G}_i$ These paired instances $\langle \mathcal{G}_i, \mathcal{D}_i \rangle$ are used for subsequent two-phase SCL.

This self-augmented data offers two key advantages: First, since the true causal graph $\mathcal{G}^*$ is unknown, the posterior estimation $(\mathcal{G}_1, \ldots, \mathcal{G}_n)$ provides a diverse range of plausible causal structures, capturing epistemic uncertainty. Second, by re-estimating parameters and forward sampling compatible datasets $\mathcal{D}_i$ from each $\mathcal{G}_i$, we generate accurately labeled instances $\langle \mathcal{G}_i, \mathcal{D}_i \rangle$, where $\mathcal{G}_i$ likely retains similarity to the augmented graph $\mathcal{G}_{\mathcal{I}}$, which entails the similarity properties of the $\mathcal{D}_i$. More importantly, this generation mechanism is free, and we can generate as much training data as we want.

**Implementation Details.** We summarize the key steps here:

*I. Initialization*: The initial graph $\mathcal{G}^{(0)}$ is randomly initialized with the *augmented graph*'s node number.

*II. Mutation*: In each iteration, a candidate graph $\mathcal{G}^{cand}$ is generated from $\mathcal{G}^{(t-1)}$ by mutating the graph structure (adding, deleting, or reversing an edge).

*III. Parameters Re-Estimation*: For each $\mathcal{G}^{cand}$, the conditional probability table (CPT) is re-estimated via maximum likelihood estimation (MLE), ensuring alignment with the augmented data $\mathcal{D}_{\mathcal{I}}$. If a node's parents remain unchanged, the corresponding CPT is inherited from the previous graph.

*IV. Evaluation*: The candidate graph $\mathcal{G}^{cand}$ is evaluated using a goodness-of-fit score, such as log-likelihood, computed based on its alignment with the observed data.

*V. Acceptance*: The acceptance probability of $\mathcal{G}^{cand}$ is computed as: $\alpha(\mathcal{G}^{cand}|\mathcal{G}^{(t-1)}) = \min\left(1, \frac{\text{score}(\mathcal{G}^{cand})}{\text{score}(\mathcal{G}^{(t-1)})}\right)$, $\mathcal{G}^{cand}$ is accepted as $\mathcal{G}^{(t)}$ with probability $\alpha$, else it retains $\mathcal{G}^{(t-1)}$.

*VI. Forward Sampling*: Once $\mathcal{G}^{cand}$ is accepted, the corresponding $\mathcal{D}^{(t)}$ is generated by forward sampling from the graph, and the pair $\langle \mathcal{G}^{(t)}, \mathcal{D}^{(t)} \rangle$ is stored for future use.

There are, however, several critical considerations to address. First, since the IS-MCMC process operates on augmented graphs rather than standard causal graph, additional constraints must be imposed to ensure validity. Furthermore, managing time complexity is crucial for efficient sampling. Accordingly, we implement the following optimizations:

- *Good Initial State*: Using a proxy algorithm to generate an initial graph significantly reduces convergence time compared to starting from a purely random graph.
- *Intervention Constraints*: The mutation process must respect system-environment (*sys – env node*) constraints that are specific to the interventional setup. Specifically, edges from system to environment nodes (*sys → env*) are not permitted, and interactions between environment nodes are excluded. This ensures that the augmented graphs remain consistent under the JCI framework.
- *Parameters Reuse*: Only the nodes affected by structural changes need to have their CPTs re-estimated. The rest of the graph can retain its parameters from the previous.
- *Efficient Parallel Chains*: Running multiple parallel IS-MCMC chains allows for faster exploration of the augmented graph space, improving the sampling efficiency.

**Remark.** By addressing these factors, we ensure that the generation of training data is both effective and efficient, thereby enhancing the task performance. We detail optimization strategies in Appendix B.1 and summarize the procedure in Algorithms 1 and 2. *Theoretically*, our IS-MCMC follows a standard structured MCMC framework (Madigan et al., 1995; Su & Borsuk, 2016; Kuipers & Moffa, 2017) and converges to the posterior distribution $P(\mathcal{G}_\mathcal{I}|\mathcal{D}_\mathcal{I})$, thus avoiding convergence to incorrect graph structures. (See formal discussion and convergence visualization in Appendix C). *Empirically*, we also demonstrate its task effectiveness in Section 4.2 with experimental evidence.

### 3.2. Two-Phase Supervised Causal Learning

With the generated training data, a straightforward idea is to train a model that directly predicts the causal graphs. However, we argue that this may not be the best choice, and that the model should instead predict the *identifiable* causal structures, *i.e.*, the $\mathcal{I}$-CPDAG. Recall that a directed causal edge is identifiable if and only if it appears consistently in every causal graph compatible with the given data. In other words, there does not exist stable association at all between

the given data and the unidentifiable causal edges, despite their presence in the true causal graph. For this reason, we choose to focus on identifying the identifiable components in the $\mathcal{I}$-CPDAG, namely, skeletons and v-structures.

Inspired by the PC algorithm (Spirtes & Glymour, 1991), we implement the supervised causal learning as a two-phase process, with phase 1 focusing on identifying the skeleton, followed by orientation predictions in phase 2. This shift implies that for each phase, we must define the learning target, the feature set, and the classification mechanism.

**Implementation Details.** We first review the PC algorithm, which consists of two main stages: Phase 1 identifies the skeleton and the separating sets $\mathcal{SS}$, determining the existence of edges. Starting from an undirected complete graph, edges are removed iteratively through conditional independence (CI) tests. Specifically, an edge $X_i - X_j$ is removed if $X_i$ is conditionally independent of $X_j$ given a subset $\mathcal{S}$ of other variables in the current $k$-order graph. Phase 2 orients the unshielded triples $\mathcal{U}$ in the skeleton based on the $\mathcal{SS}$, assigning edge directionality. Here, the triple $\langle X_a, X_c, X_b \rangle$ is oriented into a v-structure $X_a \to X_c \leftarrow X_b$ if $X_c$ is not in the separating set of $X_a$ and $X_b$.

Then, we establish a formal connection between PC and SCL. Due to limited space, we focus on phase 2 as an example. Phase 1 and more details are provided in B.2.

*Task*: For all unshielded triples $\mathcal{U}$, classify whether each triple $\langle X_a, X_c, X_b \rangle$ is a v-structure.

*Featurization*: Query $\mathcal{SS}$ satisfying $X_a \perp X_b | \mathcal{SS}$, and calculate existence feature: $F_{\langle X_a, X_c, X_b \rangle} = \begin{cases} 1 & X_c \in \mathcal{SS} \\ 0 & X_c \notin \mathcal{SS} \end{cases}$

*Classifier*: Train a binary classifier using features:
$$C_o(F_{\langle X_a, X_c, X_b \rangle}) := \begin{cases} v\text{-}structure & F_{\langle X_a, X_c, X_b \rangle} = 0 \\ non\text{-}v\text{-}structure & F_{\langle X_a, X_c, X_b \rangle} \neq 0 \end{cases}$$

In summary, detecting v-structures can be framed as a binary classification task, with the PC algorithm viewed as feature engineering combined with a static classifier. It's clear that this approach can be extended by enriching the feature set, such as considering all possible separating sets or more conditional dependency information. Crucially, it enables us to replace heuristic searches and potentially erroneous CI tests with robust classification mechanisms trained on data.

**Remark.** Building on this insight, we introduce two classifiers: one to detect the existence of edges between nodes, and the other to determine whether an unshielded triple is a v-structure. We detail feature engineering in Appendix B.2 and summarize the learnable components in Algorithms 4 and 5. Moreover, the interventional setting (augmented graph) offers further benefits. During inference, prior knowledge from the augmented graph allow pre-identifying certain edges (*e.g.*, *env → sys* node), facilitating

*Table 1.* Performance comparison of $\mathcal{I}$-CPDAG (SHD ↓ / SID ↓ / F1-score ↑). ● : Best, ● : Next. /: Crashes, –: Timeout.

| METHODS | TICL | | | JCI-PC | | | JCI-BLIP | | | JCI-HC | | | JCI-GOLEM | | | AVICI* | | | ENCO* | | | IGSP* | | | UT-IGSP | | | GIES | | |
|---|---|---|---|---|---|---|---|---|---|---|---|---|---|---|---|---|---|---|---|---|---|---|---|---|---|---|---|---|---|---|
| | SHD | SID | F1 | SHD | SID | F1 | SHD | SID | F1 | SHD | SID | F1 | SHD | SID | F1 | SHD | SID | F1 | SHD | SID | F1 | SHD | SID | F1 | SHD | SID | F1 | SHD | SID | F1 |
| EARTHQUAKE | 0 | 0 | 1.00 | 0 | 0 | 1.00 | 0 | 0 | 1.00 | 5 | 14 | 0.40 | 6 | 19 | 0.00 | 3 | 6 | 0.40 | 3 | 12 | 0.67 | 0 | 0 | 1.00 | 0 | 0 | 1.00 | 1 | 0 | 0.89 |
| SURVEY | 0 | 0 | 1.00 | 5 | 18 | 0.40 | 1 | 4 | 0.91 | 0 | 0 | 1.00 | 7 | 23 | 0.40 | 8 | 18 | 0.22 | 5 | 21 | 0.29 | 1 | 4 | 0.91 | 1 | 4 | 0.91 | 1 | 4 | 0.91 |
| ASIA | 2 | 13 | 0.86 | 4 | 19 | 0.75 | 3 | 18 | 0.77 | 6 | 28 | 0.46 | 15 | 36 | 0.00 | 7 | 28 | 0.31 | 6 | 13 | 0.74 | 4 | 17 | 0.62 | 5 | 24 | 0.73 | 4 | 14 | 0.77 |
| SACHS | 3 | 6 | 0.90 | 19 | 56 | 0.52 | 7 | 41 | 0.67 | 5 | 29 | 0.74 | 18 | 61 | 0.11 | 37 | 46 | 0.22 | 41 | 42 | 0.26 | 10 | 16 | 0.70 | 13 | 21 | 0.61 | 15 | 38 | 0.48 |
| CHILD | 7 | 67 | 0.81 | 40 | 299 | 0.62 | 10 | 188 | 0.78 | 16 | 211 | 0.65 | 41 | 330 | 0.21 | / | / | / | 110 | 251 | 0.06 | 29 | 124 | 0.63 | 33 | 214 | 0.45 | 46 | 197 | 0.29 |
| INSURANCE | 17 | 295 | 0.78 | 66 | 537 | 0.53 | 24 | 360 | 0.68 | 32 | 342 | 0.60 | 63 | 687 | 0.23 | / | / | / | 172 | 505 | 0.14 | 80 | 455 | 0.34 | 82 | 442 | 0.31 | 87 | 261 | 0.52 |
| WATER | 45 | 470 | 0.53 | – | – | – | 46 | 538 | 0.49 | 48 | 503 | 0.46 | 81 | 564 | 0.22 | / | / | / | 58 | 527 | 0.46 | / | / | / | / | / | / | / | / | / |
| MILDEW | 29 | 239 | 0.76 | 58 | 766 | 0.76 | 35 | 523 | 0.51 | 41 | 500 | 0.51 | – | – | – | / | / | / | 262 | 820 | 0.10 | 130 | 482 | 0.22 | 123 | 430 | 0.25 | 116 | 81 | 0.41 |
| ALARM | 9 | 96 | 0.91 | 67 | 481 | 0.76 | 18 | 152 | 0.75 | 38 | 416 | 0.61 | – | – | – | / | / | / | 236 | 612 | 0.11 | 65 | 247 | 0.43 | 70 | 274 | 0.37 | 88 | 219 | 0.39 |
| BARLEY | 55 | 948 | 0.63 | 127 | 1789 | 0.57 | 39 | 919 | 0.70 | 78 | 1328 | 0.39 | – | – | – | / | / | / | 98 | 1059 | 0.55 | / | / | / | / | / | / | – | – | – |
| HAILFINDER | 42 | 509 | 0.66 | 157 | 1152 | 0.33 | 43 | 558 | 0.70 | 81 | 935 | 0.38 | – | – | – | / | / | / | 98 | 730 | 0.61 | 232 | 870 | 0.14 | 235 | 910 | 0.14 | 66 | 973 | 0.00 |
| HEPAR2 | 42 | 1216 | 0.79 | – | – | – | 50 | 1228 | 0.72 | 43 | 855 | 0.81 | – | – | – | / | / | / | 75 | 1693 | 0.57 | 70 | 1663 | 0.62 | 67 | 1606 | 0.64 | 81 | 1085 | 0.67 |
| WIN95PTS | 101 | 611 | 0.58 | – | – | – | 109 | 900 | 0.53 | – | – | – | – | – | – | / | / | / | 135 | 893 | 0.54 | / | / | / | / | / | / | 112 | 992 | 0.00 |
| PATHFINDER | 187 | 9473 | 0.27 | – | – | – | 207 | / | 0.30 | – | – | – | – | – | – | / | / | / | 156 | 5528 | 0.47 | 1345 | 7261 | 0.10 | 1348 | 9226 | 0.08 | – | – | – |
| RANK (SHD) | 1.14 ± 0.12 | | | 5.36 ± 3.23 | | | 2.14 ± 0.41 | | | 3.86 ± 4.41 | | | 7.43 ± 3.67 | | | 8.07 ± 2.78 | | | 6.36 ± 5.80 | | | 4.64 ± 2.66 | | | 5.21 ± 2.74 | | | 5.50 ± 3.11 | | |
| RANK (SID) | 1.57 ± 0.82 | | | 6.43 ± 3.82 | | | 3.57 ± 2.53 | | | 4.64 ± 5.37 | | | 8.14 ± 3.69 | | | 7.64 ± 2.66 | | | 5.29 ± 7.92 | | | 3.93 ± 2.78 | | | 4.57 ± 2.67 | | | 3.43 ± 2.96 | | |
| RANK (F1) | 1.36 ± 0.37 | | | 4.43 ± 4.24 | | | 2.29 ± 0.78 | | | 3.93 ± 4.49 | | | 7.71 ± 3.35 | | | 8.00 ± 3.14 | | | 5.93 ± 7.49 | | | 4.86 ± 2.84 | | | 5.21 ± 2.45 | | | 5.36 ± 2.52 | | |

*Table 2.* Performance comparison of *Intervention Targets Detection* (F1 ↑). ● : Best, ● : Next. /: Crashes, –: Timeout.

| DATASETS | EARTHQUAKE | SURVEY | ASIA | SACHS | CHILD | INSURANCE | WATER | MILDEW | ALARM | BARLEY | HAILFINDER | HEPAR2 | WIN95PTS | PATHFINDER | RANK (F1) |
|---|---|---|---|---|---|---|---|---|---|---|---|---|---|---|---|
| UT-IGSP | 0.50 | 1.00 | 0.44 | 0.33 | 0.35 | 0.17 | / | 0.22 | 0.15 | / | 0.27 | 0.23 | / | 0.04 | 5.14±3.27 |
| CITE | 1.00 | 1.00 | 0.67 | 0.36 | 0.67 | 0.32 | / | 0.38 | 0.56 | / | 0.67 | 0.61 | / | 0.60 | 3.07±2.21 |
| PREDITER | 1.00 | 1.00 | 0.67 | 0.50 | – | – | / | – | – | / | – | – | / | / | 5.21±5.17 |
| JCI-GOLEM | 0.67 | 0.40 | 0.40 | 0.22 | 0.18 | 0.11 | 0.23 | – | – | – | – | – | – | – | 6.29±2.20 |
| JCI-HC | 1.00 | 0.50 | 0.67 | 0.57 | 0.40 | 0.29 | 0.33 | 0.34 | 0.29 | 0.26 | 0.23 | 0.23 | – | – | 4.14±1.98 |
| JCI-BLIP | 1.00 | 1.00 | 0.80 | 0.67 | 0.57 | 0.45 | 0.44 | 0.35 | 0.38 | 0.34 | 0.34 | 0.82 | 0.29 | 0.23 | 2.43±0.67 |
| JCI-PC | 1.00 | 1.00 | 0.80 | 0.57 | 0.28 | 0.50 | – | 0.26 | 0.64 | 0.70 | 0.25 | – | – | – | 3.43±3.10 |
| TICL | 1.00 | 1.00 | 1.00 | 1.00 | 1.00 | 0.83 | 0.86 | 0.58 | 0.82 | 0.83 | 0.76 | 0.90 | 0.50 | 0.74 | 1.00±0.00 |

Meek rules (Meek, 1995) to orient additional edges. *Theoretically*, our method preserves the PC algorithm's asymptotic properties: as sample size grows, the classifiers converges to a theoretically plausible solution, *i.e.*, correct CPDAG (or $\mathcal{I}$-CPDAG in intervention), thus ensuring identifiability (Proof in Appendix D). *Empirically*, we also demonstrate its effectiveness and improvements in Sections 4.1 and 4.4.

## 4. Experiments

In this section, we conduct extensive experiments to investigate the following core research questions (RQs):

**RQ1:** How does the TICL perform in causal structure identification from interventional data? *(Effectiveness)*.

**RQ2:** How the quality and quantity of training data from self-augmentation strategy affect TICL? *(Generalizability)*.

**RQ3:** How does the sampling time and complete running time of TICL compare to other methods? *(Efficiency)*.

**RQ4:** How does the TICL perform compared to others across different intervention settings? *(Versatility)*.

**Benchmarks & Baselines.** We use 14 (semi-)real causal graph datasets from the bnlearn (Scutari, 2010) repository as benchmarks. We compare TICL with methods

designed for intervention data, including score-based methods: GIES (Hauser & Bühlmann, 2012), IGSP (Wang et al., 2017; Yang et al., 2018), UT-IGSP (Squires et al., 2020), and continuous optimization-based methods: ENCO (Lippe et al., 2022), AVICI (Lorch et al., 2022). We also extend observational data methods using the JCI framework to intervention settings, including constraint-based methods like PC (Spirtes & Glymour, 1991), score-based methods like HC (Tsamardinos et al., 2006), BLIP (Scanagatta et al., 2015), and gradient-based methods like GOLEM (Ng et al., 2020). Note that some methods require known intervention targets, marked with *; DCDI (Brouillard et al., 2020) and BaCaDI (Hägele et al., 2023) failed due to their focus on continuous data, and results from CSIvA (Ke et al., 2023b) and SDI (Ke et al., 2023a) are omitted due to inaccessible code. More details are provided in Appendix E.1 and E.2.

**Training Datasets & Metrics.** For the training data, by default we use 400 synthetic causal graph instances sampled via the IS-MCMC and forward sample 10k instances from re-parameterized conditional probability tables. XG-Boost (Chen & Guestrin, 2016) is used as the classifier. The F1-Score evaluates intervention target detection and $\mathcal{I}$-CPDAG discovery. For causal assessment, we also use Structural Hamming Distance (SHD) and Structural Intervention Distance (SID) (Peters & Bühlmann, 2015).

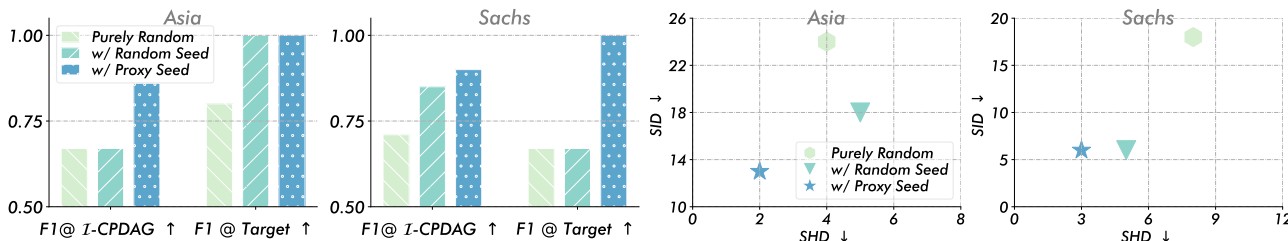

Figure 4. The impact of different synthetic data strategies on the performance of causal structure learning.

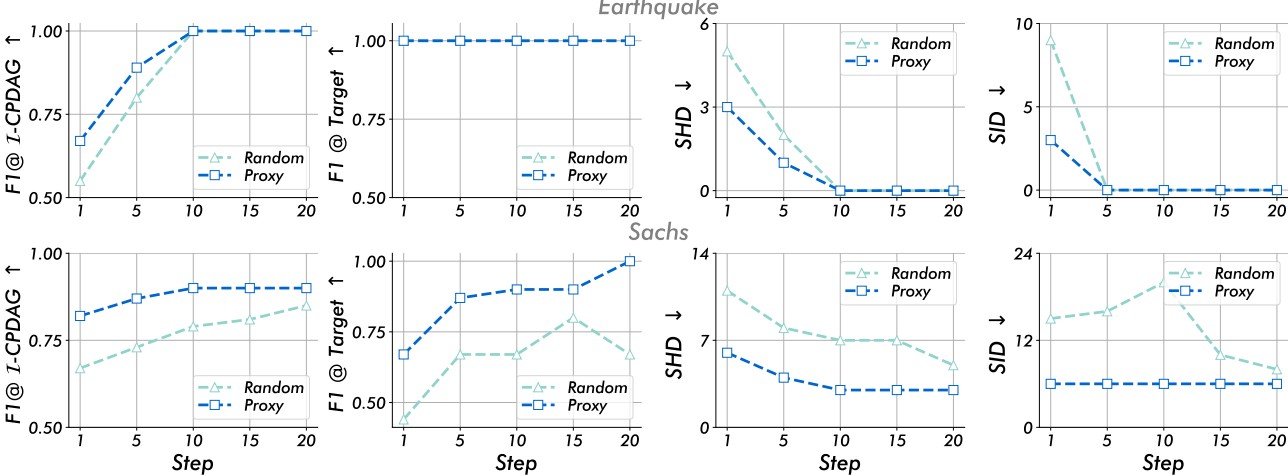

Figure 5. The impact of different initialization strategies on the performance of IS-MCMC convergence.

More details in Appendix E.3 and E.4. Code repository can be found at https://github.com/microsoft/reliableAI/tree/main/causal-kit/TICL.

## 4.1. Causal Structure Identification Performance (RQ1)

Following DCDI and GIES, we conduct multiple intervention experiments for each graph, with the number of experiments being 20% of the nodes. To enhance baseline diversity, we use single-node interventions (*e.g.*, ENCO limited to single-node interventions). Since hard interventions are a special case of soft ones, we choose soft interventions. Finally, we forward-sample 10k samples for both observational and intervention cases to generate test data.

As shown in Tables 1 and 2, TICL achieves competitive results across 14 causal graph datasets. Its advantages are highlighted in two aspects: ❶ *Diverse and Challenging Benchmarking:* Unlike methods tested only on synthetic datasets with 10-30 nodes (Brouillard et al., 2020; Lorch et al., 2022; Hägele et al., 2023; Ke et al., 2023b), we evaluate on the bnlearn benchmark, which includes real-world-inspired causal graphs ranging from small to over 100 nodes. Most methods fail as DAGs grow super exponentially, while TICL demonstrates superior scalability and consistent performance. ❷ *Highly Competitive Results:* In intervention target detection, TICL outperforms all methods, with an average F1-score improvement of 50.21% over the second-best method. For $\mathcal{I}$-CPDAG discovery, TICL improves the

F1 score by 13.62% over the best baseline. The exception is ENCO on Pathfinder, where prior knowledge of known intervention targets yields comparable performance. Overall, TICL leads with an average F1 rank of 1.36 ± 0.37 for causal discovery and 1.00 ± 0.00 for intervention target detection. ❸ Beyond this, we find methods based on the JCI framework also show competitive performance, suggesting it is a promising direction for intervention causal discovery.

## 4.2. Training Data Study: Quantity and Quality (RQ2)

We maintain consistent setups and assess the impact of self-augmented training data quality and quantity for SCL.

**Data Quality.** To evaluate training data quality, we consider three generation strategies: *Purely Random*, generating random graphs (Erdős-Rényi, Scale-Free) with 10-50 nodes; *IS-MCMC w/ Random Seed*, using random graphs as initial seeds for IS-MCMC; and *IS-MCMC w/ Proxy Seed*, using proxy algorithms (*e.g.*, JCI-BLIP) to generate seed graphs for IS-MCMC. We compare performance in Figure 4 and IS-MCMC convergence in Figure 5. We observe that: ❶ *High-quality in-domain data is crucial for supervised causal models.* The three generation strategies can be viewed as progressively approach the test data domain. As shown in Figure 4, both IS-MCMC variants outperform the purely random baseline across all metrics. Among them, the proxy seed method yields the best results, highlighting the value of self-augmentation strategy for test-time training. ❷ *Opti-*

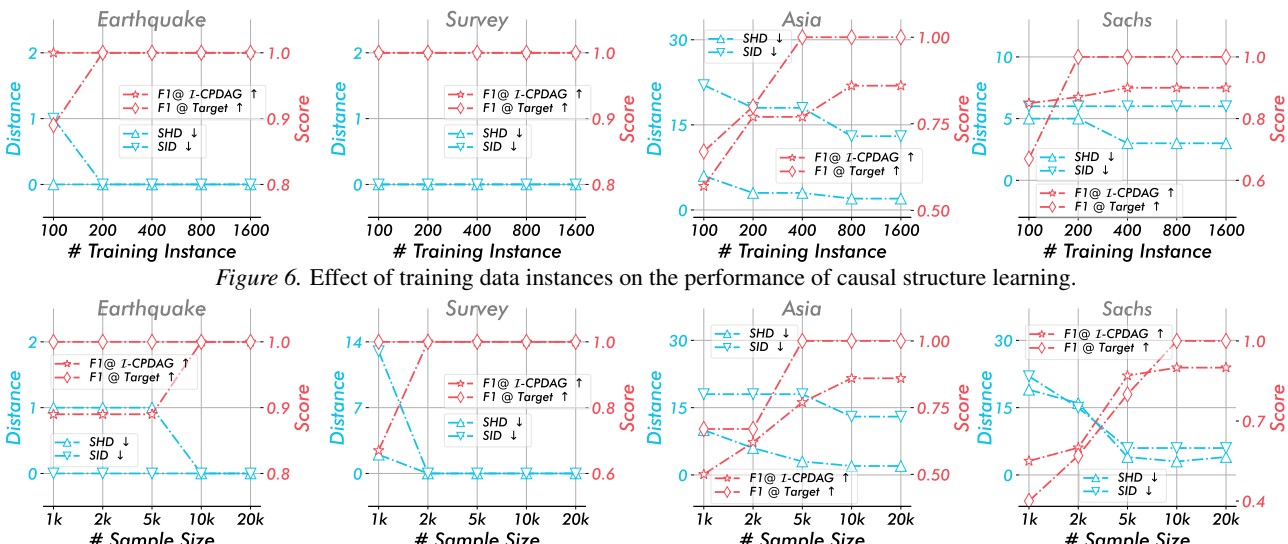

*Figure 6.* Effect of training data instances on the performance of causal structure learning.

*Figure 7.* Effect of training sample size on the performance of causal structure learning.

*mizing posterior-induced data generation process is key to self-augmentation.* Although both IS-MCMC strategies target the same posterior, empirical results (Figure 5) show that proxy seeds converge faster and excel in both causal discovery and target detection. This confirms that our optimization strategy, via effective initialization and intervention constraints, ensures superior efficiency and convergence.

**Data Quantity.** We evaluate `TICL`'s scaling property with training data size: instance count and sample size per instance. Figure 6 shows the impact of instance counts (100-1600), while Figure 7 plots performance with sample sizes (1k–20k) under default intervention settings. We observe that: ❶ *Our `TICL` benefits from more training instances.* As shown in Figure 6, performance improves with more instances across all datasets. Smaller graphs require fewer instances, while larger ones gain more due to increased complexity. ❷ *`TICL` benefits from larger sample sizes.* As shown in Figure 7, increasing sample size stabilizes independence tests, improving various metrics across graphs. These properties further highlight the promise, as our self-augmentation enables the generation of abundant free data.

### 4.3. Sampling and Running Efficiency Study (RQ3)

We maintain the setup and assess the efficiency of `TICL`.

**Sampling Time.** Since our initialization ensures rapid convergence, we evaluate the multi-chain MCMC strategy. As shown in Figure 8 (a), multi-chain MCMC shows minimal time increase as graphs scale, offering significant acceleration. This stems from IS-MCMC's parallel adaptability, confirming the strategy's efficiency.

**Running Time.** We compare the wall-clock times (including proxy time) of all baseline methods. As shown in Figure 8 (b): ❶ *Other learning-based methods are inefficient,*

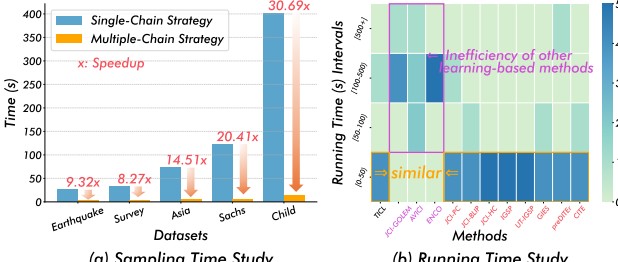

*Figure 8.* Efficiency study of our method.

typically requiring 100-500 seconds due to slow gradient optimization, while non-SCL methods finish within 50 seconds. ❷ *`TICL` matches non-SCL efficiency.* While execution time involves proxy algorithm, this is optional; in high-risk causal discovery, time cost becomes secondary.

### 4.4. Different Intervention Settings Study (RQ4)

We further conducted experiments and analyses on **RQ4** regarding intervention types, intervention ratios, and test sample sizes. Due to the complex diversity of different situations, this has been rarely considered comprehensively before. However, as shown in Figure 1, `TICL` always maintains its superiority under various intervention combinations and a limited number of samples. Due to page limitation, for more results and analysis, please refer to Appendix F.

## 5. Conclusion and Future work

In this paper, we present `TICL` for causal structural learning from interventional data, particularly when faced with generalization challenges in the real-world. We introduced a specific test-time training technique and demonstrated the prospects of JCI for SCL. In the future, we further explore the performance of `TICL` in continuous setting and under fewer assumptions to pursue a causal foundation model.

## Impact Statement

This paper presents work whose goal is to advance the field of machine learning for causal discovery. There are many potential societal consequences of our work, none of which we feel must be specifically highlighted here.

## Acknowledgments

This work is mainly supported by the Guangdong Basic and Applied Basic Research Foundation (No. 2025A1515011994). This work is also supported by the National Natural Science Foundation of China (No. 62402414), Guangdong Provincial Project 2025D03J0014, Guangzhou Municipal Science and Technology Project (No. 2023A03J0011), the Guangzhou Industrial Information and Intelligent Key Laboratory Project (No. 2024A03J0628), and Guangdong Provincial Key Lab of Integrated Communication, Sensing and Computation for Ubiquitous Internet of Things (No. 2023B1212010007).

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

# SUPPLEMENTARY MATERIAL
## TEST-TIME LEARNING OF CAUSAL STRUCTURE FROM INTERVENTIONAL DATA

## TABLE OF CONTENTS

# A. Background Knowledge and Related Work

## A.1. Background Knowledge

In this section, we give basic concepts related to causal graphs and joint causal inference.

### A.1.1. CAUSAL GRAPH-RELATED CONCEPT

**Definition A.1** (Directed Acyclic Graph). A directed acyclic graph (DAG) is a directed graph $\mathcal{G}$ that has no cycles, *i.e.* no directed paths starting and ending at the same vertex.

**Definition A.2** (Skeleton). A undirected graph $\mathcal{K} = (V_{\mathcal{K}}, E_{\mathcal{K}})$ represents the skeleton of a causal DAG $\mathcal{G} = (V_{\mathcal{G}}, E_{\mathcal{G}})$ if $(X \to Y) \in E_{\mathcal{G}} \cup (Y \to X) \in E_{\mathcal{G}} \iff (X - Y) \in E_{\mathcal{K}}$.

**Definition A.3** (UT and V-structures). A triple of variables $\langle X, T, Y \rangle$ in a skeleton is an unshielded triple, or short for UT, if $X$ and $Y$ are adjacent to $T$ but are not adjacent to each other. $\langle X, T, Y \rangle$ can be further oriented to become a v-structure $X \to T \leftarrow Y$, in which $T$ is called the collider.

**Definition A.4** ($PC$). Denote the set of parents and children of $X$ in a skeleton as $PC_X$, in other words, $PC_X$ are the neighbors of $X$ in the skeleton. For convenience, if we discuss $PC_X$ in the context of a UT $\langle X, T, Y \rangle$, we intentionally mean the set of parents and children of $X$ but exclude $T$. Similarly, $PC_T$ excludes $X, Y$.

**Definition A.5** (Vicinity). We define the vicinity of a UT $\langle X, T, Y \rangle$ as $V_{\langle X,T,Y \rangle} := \{X, T, Y\} \cup PC_X \cup PC_Y \cup PC_T$. Vicinity is a generalized version of PC, *i.e.*, the neighbors of $\{X, T, Y\}$ in the skeleton.

**Definition A.6** (Sepsets). Sepsets $\mathcal{S}$ can be define: $\{\mathcal{S} : X \perp Y | S, S \subset PC_X \cup T, \text{ or } S \subset PC_Y \cup T\}$. Under faithfulness assumption, sepsets $\mathcal{S}$ is an ensemble where each item is a subset of variables within the vicinity that d-separates $X$ and $Y$.

**Definition A.7** (Causal Graph). A causal graph $\mathcal{G}$ is a graphical description of a system in terms of cause-effect relationships, *i.e.* the causal mechanism. Specifically, for each edge $(X, Y) \in \mathcal{E}$, $X$ is the direct cause of $Y$, and $Y$ is the direct effect of $X$, satisfying the causal edge assumption, i.e., the value assigned to each variable $X$ is completely determined by the function $\mathcal{F}$ given its parent. Formally, this can be expressed as: $X_i := f(Pa(X_i))$, $\forall X_i \in \mathcal{V}$.

As natural consequence of such definitions, we can define models that entail both the structural representation and the set of functions that regulate the underlying causal mechanism.

**Definition A.8** (Structural Causal Model). A structural causal model (SCM) is defined by the tuple $\mathcal{M} = (\mathcal{V}, \mathcal{U}, \mathcal{F}, P)$, where:

- $\mathcal{V}$ is a set of endogenous variables, i.e. observable variables,
- $\mathcal{U}$ is a set of exogenous variables, i.e. unobservable variables, where $\mathcal{U} \cup \mathcal{V} = \emptyset$
- $\mathcal{F}$ is a set of functions, where each function $f_i \in F$ is defined as $fi := (\mathcal{V} \cup \mathcal{U})^p \to \mathcal{V}$, with $p$ the ariety of $f_i$, so that $f_i$ determines completely the value of $V_i$,
- $P$ is a joint probability distribution over the exogenous variables $P(\mathcal{U}) = \prod_i P(\mathcal{U}_i)$.

**Assumption A.9** (Causally Sufficiency). The set of variables V is said to be causally sufficient if and only if every cause of any subset of V is contained in V itself.

Note that, in our setup, due to the *causality sufficiency*, exogenous variables are ignored.

### A.1.2. JCI ASSUMPTION-RELATED CONCEPT

**Assumption A.0** ("Joint SCM"). The data-generating mechanism is described by a simple SCM $\mathcal{M}$ of the form:

$$\mathcal{M} : \begin{cases} C_k = f_k(\boldsymbol{X}_{\text{PA}_{\mathcal{H}}(k) \cap \mathcal{I}}, \boldsymbol{C}_{\text{PA}_{\mathcal{H}}(k) \cap \mathcal{K}}, \boldsymbol{E}_{\text{PA}_{\mathcal{H}}(k) \cap \mathcal{J}}), & k \in \mathcal{K}, \\ X_i = f_i(\boldsymbol{X}_{\text{PA}_{\mathcal{H}}(i) \cap \mathcal{I}}, \boldsymbol{C}_{\text{PA}_{\mathcal{H}}(i) \cap \mathcal{K}}, \boldsymbol{E}_{\text{PA}_{\mathcal{H}}(i) \cap \mathcal{J}}), & i \in \mathcal{I}, \\ \mathbb{P}(\boldsymbol{E}) = \prod_{j \in \mathcal{J}} \mathbb{P}(E_j), \end{cases} \tag{1}$$

that jointly models the system and the context. Its graph $\mathcal{G}(\mathcal{M})$ has nodes $\mathcal{I} \cup \mathcal{K}$ (corresponding to system variables $\{X_i\}_{i \in \mathcal{I}}$ and context variables $\{C_k\}_{k \in \mathcal{K}}$).

It will always make this assumption in order to facilitate the formulation of JCI, the following three assumptions that we discuss are optional, and their applicability has to be decided based on a case-by-case basis. Typically, when a modeler

decides to distinguish a system from its context, the modeler possesses background knowledge that expresses that the context is exogenous to the system:

**Assumption A.1.** ("Exogeneity") No system variable causes any context variable, i.e.,

$$\forall k \in \mathcal{K}, \forall i \in \mathcal{I}: \quad i \to k \notin \mathcal{G}(\mathcal{M}).$$

**Assumption A.2.** ("Complete randomized context") No context variable is confounded with a system variable, i.e.,

$$\forall k \in \mathcal{K}, \forall i \in \mathcal{I}: \quad i \leftrightarrow k \notin \mathcal{G}(\mathcal{M}).$$

JCI Assumption 1 is often easily justifiable, but the applicability of JCI Assumption 2 may be less obvious in practice. Then, Assumption 3 is further stated, which can be useful whenever both JCI Assumptions 1 and 2 have been made as well.

**Assumption A.3.** ("Generic context model") The context graph[1] $\mathcal{G}(\mathcal{M})_\mathcal{K}$ is of the following special form:

$$\forall k \neq k' \in \mathcal{K}: \quad k \leftrightarrow k' \in \mathcal{G}(\mathcal{M}) \quad \wedge \quad k \to k' \notin \mathcal{G}(\mathcal{M}).$$

The following key result essentially states that when one is only interested in modeling the causal relations involving the system variables (under JCI Assumptions 1 and 2), one does not need to care about the causal relations between the context variables, as long as one correctly models the context distribution.

**Theorem A.4.** *Assume that JCI Assumptions A.0, A.1 and A.2 hold for SCM $\mathcal{M}$:*

$$\mathcal{M}: \begin{cases} C_k = f_k(\boldsymbol{C}_{\mathrm{PA}_\mathcal{H}(k)\cap\mathcal{K}}, \boldsymbol{E}_{\mathrm{PA}_\mathcal{H}(k)\cap\mathcal{J}}), & k \in \mathcal{K}, \\ X_i = f_i(\boldsymbol{X}_{\mathrm{PA}_\mathcal{H}(i)\cap\mathcal{I}}, \boldsymbol{C}_{\mathrm{PA}_\mathcal{H}(i)\cap\mathcal{K}}, \boldsymbol{E}_{\mathrm{PA}_\mathcal{H}(i)\cap\mathcal{J}}), & i \in \mathcal{I}, \\ \mathbb{P}(\boldsymbol{E}) = \prod_{j\in\mathcal{J}} \mathbb{P}(E_j), \end{cases}$$

*For any other SCM $\tilde{\mathcal{M}}$ satisfying JCI Assumptions A.0, A.1 and A.2 that is the same as $\mathcal{M}$ except that it models the context differently, i.e., of the form*

$$\tilde{\mathcal{M}}: \begin{cases} C_k = \tilde{f}_k(\boldsymbol{C}_{\mathrm{PA}_{\tilde{\mathcal{H}}}(k)\cap\mathcal{K}}, \boldsymbol{E}_{\mathrm{PA}_{\tilde{\mathcal{H}}}(k)\cap\tilde{\mathcal{J}}}), & k \in \mathcal{K}, \\ X_i = f_i(\boldsymbol{X}_{\mathrm{PA}_\mathcal{H}(i)\cap\mathcal{I}}, \boldsymbol{C}_{\mathrm{PA}_\mathcal{H}(i)\cap\mathcal{K}}, \boldsymbol{E}_{\mathrm{PA}_\mathcal{H}(i)\cap\mathcal{J}}), & i \in \mathcal{I}, \\ \mathbb{P}(\boldsymbol{E}) = \prod_{j\in\tilde{\mathcal{J}}} \mathbb{P}(E_j), \end{cases}$$

*with $\mathcal{J} \subseteq \tilde{\mathcal{J}}$ and $\mathrm{PA}_\mathcal{H}(i) = \mathrm{PA}_{\tilde{\mathcal{H}}}(i)$ for all $i \in \mathcal{I}$, we have that*

1. *the conditional system graphs coincide: $\mathcal{G}(\mathcal{M})_{(\mathcal{K})} = \mathcal{G}(\tilde{\mathcal{M}})_{(\mathcal{K})}$;*

2. *if $\tilde{\mathcal{M}}$ and $\mathcal{M}$ induce the same context distribution, i.e., $\mathbb{P}_\mathcal{M}(\boldsymbol{C}) = \mathbb{P}_{\tilde{\mathcal{M}}}(\boldsymbol{C})$, then for any perfect intervention on the system variables $(I, \boldsymbol{\xi}_I)$ with $I \subseteq \mathcal{I}$ (including the non-intervention $I = \emptyset$), $\tilde{\mathcal{M}}_{(I,\boldsymbol{\xi}_I)}$ is observationally equivalent to $\mathcal{M}_{(I,\boldsymbol{\xi}_I)}$.*

3. *if the context graphs $\mathcal{G}(\tilde{\mathcal{M}})_\mathcal{K}$ and $\mathcal{G}(\mathcal{M})_\mathcal{K}$ induce the same separations, then also $\mathcal{G}(\tilde{\mathcal{M}})$ and $\mathcal{G}(\mathcal{M})$ induce the same separations (where "separations" can refer to either $d$-separations or $\sigma$-separations).*

The following corollary of Theorem A.4 states that JCI Assumption A.3 can be made without loss of generality for the purposes of constraint-based causal discovery if the context distribution contains no conditional independences:

**Corollary A.5.** *Assume that JCI Assumptions A.0, A.1 and A.2 hold for SCM $\mathcal{M}$. Then there exists an SCM $\tilde{\mathcal{M}}$ that satisfies JCI Assumptions A.0, A.1 and A.2 and A.3, such that*

1. *the conditional system graphs coincide: $\mathcal{G}(\mathcal{M})_{(\mathcal{K})} = \mathcal{G}(\tilde{\mathcal{M}})_{(\mathcal{K})}$;*

2. *for any perfect intervention on the system variables $(I, \boldsymbol{\xi}_I)$ with $I \subseteq \mathcal{I}$ (including the non-intervention $I = \emptyset$), $\tilde{\mathcal{M}}_{(I,\boldsymbol{\xi}_I)}$ is observationally equivalent to $\mathcal{M}_{(I,\boldsymbol{\xi}_I)}$;*

3. *if the context distribution $\mathbb{P}_\mathcal{M}(\boldsymbol{C})$ contains no conditional or marginal independences, then the same $\sigma$-separations hold in $\mathcal{G}(\tilde{\mathcal{M}})$ as in $\mathcal{G}(\mathcal{M})$; if in addition, the Directed Global Markov Property holds for $\mathcal{M}$, then also the same $d$-separations hold in $\mathcal{G}(\tilde{\mathcal{M}})$ as in $\mathcal{G}(\mathcal{M})$.*

---

[1]Remember that $\mathcal{G}(\mathcal{M})_\mathcal{K}$ denotes the subgraph on the context variables $\mathcal{K}$ induced by the causal graph $\mathcal{G}(\mathcal{M})$.

## A.2. Related Work

In this section, we present some of the most relevant work to the key points of this paper.

### A.2.1. INTERVENTIONAL CAUSAL DISCOVERY

Recovering the underlying causal structure from observational and interventional data is a fundamental research problem (Spirtes et al., 2000; Pearl, 2009). When only observational data are available, constraint-based methods identify a directed acyclic graph (DAG) consistent with conditional independence constraints. Notable examples include the PC algorithm (Spirtes & Glymour, 1991) and its variants, such as Conservative-PC (Ramsey et al., 2012) and PC-stable (Colombo et al., 2014). Score-based methods, including GES (Chickering, 2002) and hill-climbing (Koller & Friedman, 2009), search for the optimal DAG under a predefined scoring function combined with constraints. Gradient-based methods further extend score-based approaches by transforming discrete searches into continuous equality constraints, such as NOTEARS (Zheng et al., 2018), GAE (Ng et al., 2019), and GraN-DAG (Yu et al., 2019). *However, without specific assumptions, the identifiability of structures solely derived from observational data is theoretically limited.* Some works have extended causal discovery algorithms to intervention settings (Hauser & Bühlmann, 2012; Wang et al., 2017; Yang et al., 2018; Squires et al., 2020; Zhang et al., 2024; von Kügelgen et al., 2024). For example, GIES (Hauser & Bühlmann, 2012) pioneered the case with known hard interventions, while IGSP (Wang et al., 2017; Yang et al., 2018) introduced a greedy sparse ordering method to handle known general interventions. Building on this, UT-IGSP (Squires et al., 2020) partially addresses the scenario of unknown target interventions. However, given the diversity of intervention data types and experimental strategies, these methods still fail to universally address various intervention scenarios, limiting effective causal identification. *In contrast, our approach offers a unified solution for causal discovery from intervention data by leveraging a joint causal inference framework combined with supervised causal learning methods, achieving practical empirical performance.*

### A.2.2. SUPERVISED CAUSAL LEARNING

Supervised Causal Learning (SCL) is an emerging paradigm of causal discovery, and has demonstrated strong empirical performance. The strength of SCL lies in its ability to learn complex classification mechanisms, contrasting with traditional rule-based logics for detecting causal relations. Early SCL work focused on pairwise causal discovery, such as RCC (Lopez-Paz et al., 2015) and MRCL (Hill et al., 2019). Further efforts have shifted toward multivariate causal discovery, with models like DAG-EQ (Li et al., 2020) and SLdisco (Petersen et al., 2023), which apply to linear causal models. ML4C (Dai et al., 2023) and ML4S (Ma et al., 2022) focus on v-structure detection and skeleton learning, respectively. For interventional data, methods such as CSIvA (Ke et al., 2023b) and AVICI (Lorch et al., 2022) extended SCL to handle known, hard interventions. Additionally, BCNP (Dhir et al., 2025) employs Bayesian meta-learning, treating the posterior estimation problem as a supervised learning task. Furthermore, SiCL (Zhang et al., 2025) reveals that deep neural network architectures exhibit systematic biases, which cannot be eliminated by increasing model size or data volume. This approach achieves DNN-driven SCL through the joint prediction of skeleton matrices and v-tensors. However, these models are limited in generalization. *In contrast, our approach adopts a test-time training paradigm, which introduces a process that acquires training data during test time. This allows the model to "overfit" to the specific biases of the test data, addressing the generalization issue and significantly improving performance in real-world settings.*

### A.2.3. TEST-TIME TRAINING

A body of work (Sun et al., 2020; Wang et al., 2020; 2022; Liu et al., 2021; Hardt & Sun, 2024; Sun et al., 2025; Dalal et al., 2025; Chen & Liang, 2025; Behrouz et al., 2025; Tandon et al., 2025; Behrouz et al., 2026) has explored test-time training paradigm to address the challenge of distribution shift in test data. A common approach is to identify an auxiliary task that aids the model in better adapting to the test data. For instance, TTT (Sun et al., 2020) jointly trains a model for rotation prediction and image classification. TTT++ (Liu et al., 2021) extends this by employing a contrastive learning approach as an auxiliary task for adaptation. In the context of causal discovery, test-time training exhibits distinctive characteristics. Unlike prior works, causal discovery benefits from the ability to *generate test-time training data* in a self-supervised manner. We refer to this approach as *self-augmentation*. This self-augmentation allows test-time training data to capture nuances and biases inherent to the test data, thereby making it well-suited for supervised causal learning. Currently, there is limited work in this area, with the most relevant being ML4S (Ma et al., 2022), which proposes a heuristic for generating vicinal graphs at test time for skeleton learning. *To the best of our knowledge, we are the first to present a systematic approach to test-time adaptation in SCL, and we extend it to a broader context of causal discovery in general interventional settings.*

A.2.4. JOINT CAUSAL INFERENCE

Joint Causal Inference (JCI) (Mooij et al., 2020) presents a joint causal inference framework aiming to integrate multiple observed outcomes collected during different experiments (*i.e.*, contexts). In this framework, the observed variable set is divided into two disjoint sets: system variables and context variables. After that, S-FCI (Li et al., 2023) builds upon JCI by introducing a new constraint-based algorithm, enabling learning from observational and intervention data across multiple domains. (Mascaro & Castelletti, 2023) also provides a graphical representation theory of I-MEC under general interventions and designs compatible priors for Bayesian inference to ensure score equivalence of indistinguishable structures. Although JCI provides a novel framework for addressing intervention problems and simplifies the unification of different intervention settings for supervised causal learning, it does not directly address the critical questions of what the appropriate learning objectives are when applying supervised learning, and how the learning process should be designed. *To this end, we highlight the overlooked connection between identifiability in ICD and SCL. Through the modified PC-like SCL, a two-phase learning method is developed for predicting identifiable causal structures ($\mathcal{I}$-CPDAG) while ensuring theoretical identifiability.*

# B. Inplementation Details

## B.1. Training Data Acquisition via Self-Augmentation

### B.1.1. PARAMETER RE-ESTIMATION FOR CPT

The joint distribution represented by a randomly generated conditional probability table (CPT) may significantly differ from the true joint distribution corresponding to the augmented data, thus undermining the forward-sampling process. Therefore, we propose approximating the joint posterior not only on the structure of the causal graph but also on the parameters of its conditional probability distributions. This approach ensures that the distribution of the augmented graph maintains in-domain "similarity" with the true underlying distribution.

Specifically, we first use maximum likelihood estimation (Myung, 2003) to estimate the conditional probability tables (CPTs) of the initial seed graph. Secondly, during the IS-MCMC iteration process, we propose that the CPTs of the current step's graph are determined by the CPTs from the previous step. For each node in the current step's graph, if its parent nodes remain the same as those in the corresponding node from the previous step's graph, its CPT is directly inherited from the previous step. Otherwise, nodes may lose previous parent nodes due to edge deletion or gain new parent nodes due to edge addition as a result of graph structure transformation operations.

For edge deletions that result in the loss of previous parent nodes, we can adjust the CPT by *marginalization*. For instance, in step $t-1$, if node $X$ has parent nodes $Y$ and $Z$, the corresponding CPT encodes the distribution $P(X|Y,Z)$. In step $t$, if the edge $Y \to X$ is deleted, the new conditional distribution should encode $P(X|Z)$. This can be naturally achieved by marginalizing out the node $Y$ using the law of total probability, *i.e.*, $P(X|Z) = \sum_y P(X|Z, Y = y)$.

For edge additions that result in new parent nodes, the situation becomes more complicated. For instance, in step $t$, if a new edge $U \to X$ is added, introducing a new parent node $U$, the corresponding conditional probability distribution should encode $P(X|Y,Z,U)$. Considering we primarily deal with discrete data, and the *Dirichlet-multinomial* distribution is a natural choice for modeling categorical distributions, we use the Dirichlet distribution to sample different CPTs. Specifically, for each conditional probability distribution $P(X|Y = y, Z = z, U = u)$, we estimate its parameters $\alpha_i$ by maximizing the log-likelihood function of the data, which is given by:

$$
\begin{aligned}
F(\alpha) = \log p(D|\alpha) = & \quad \log \prod_i p(\mathbf{p}_i|\alpha) = \quad \log \prod_i \frac{\Gamma\left(\sum_k \alpha_k\right)}{\prod_k \Gamma(\alpha_k)} \prod_k p_{ik}^{\alpha_k - 1} \\
= & \quad N\left(\log \Gamma\left(\sum_k \alpha_k\right) - \sum_k \log \Gamma(\alpha_k) + \sum_k (\alpha_k - 1) \log \hat{p}_k\right),
\end{aligned}
$$

where $\log \hat{p}_k = \frac{1}{N} \sum_i \log p_{ik}$ represents the observed sufficient statistics, $\Gamma(x)$ denotes the Gamma function and is defined to be: $\int_0^\infty t^{x-1} e^{-t} dt$. Since this function does not have a closed-form solution, we employ a fixed-point iteration technique (Minka, 2000) to estimate the parameters. The core idea is to find an initial guess for $\alpha$ and iteratively refine it to converge to the maximum likelihood estimate. It seek a function $F(\cdot)$ that provides a lower bound for the log-likelihood function:

$$F(\alpha) \geq N\left(\left(\sum_k \alpha_k\right)\Psi\left(\sum_k \alpha_k^{old}\right) - \sum_k \log\Gamma(\alpha_k) + \sum_k \alpha_k \log\hat{p}_k + C\right),$$

where $C$ is a constant dependent on $\alpha$, ensuring that the function is optimized at $\alpha$. By setting the gradient to zero, a new iterative value for $\alpha$ is derived:

$$\alpha_k^i = \Psi^{-1}\left(\Psi\left(\sum_k \alpha_k^{i-1}\right) + \log\hat{p}_k\right),$$

where inverse digamma function $\Psi^{-1}$ can be efficiently solved using the Newton-Raphson method (Ypma, 1995). Subsequently, for each possible value $x$ of the variable $X$, we sample from *Dirichlet*$(\beta\alpha_i)$ to obtain $P(X|Y = y, Z = z, U = u)$, thereby forming our target distribution $P(X|Y, Z, U)$. Here, $\beta$ is a hyperparameter that adjusts the variance, set to 0.25.

### B.1.2. INTERVENTIONAL STRUCTURAL-MARKOV CHAIN MONTE CARLO

---

**Algorithm 1** Standard `IS-MCMC` Algorithm Workflow

---

**Input:**
    Chain Length: $T$,    Expected Training Sample Size: $N$
**Output:**
    Synthetic training data set: $\{(\mathcal{G}^{(1)}, \mathcal{D}^{(1)}), (\mathcal{G}^{(2)}, \mathcal{D}^{(2)}), \ldots\}, (\mathcal{G}^{(N)}, \mathcal{D}^{(N)})$
**Pipeline:**

1. Initialize Seed Graph $\mathcal{G}^{(0)} \sim \mathcal{G}_{\mathcal{I}}$

2. Estimating CPT parameters by maximum likelihood $\Theta^{(0)} = \text{MLE}(\mathcal{G}^{(0)}, \mathcal{D}_{\mathcal{I}})$

3. **for** iteration $t = 1, 2, \ldots$ **do**:

    (a) Propose: $\mathcal{G}^{cand} \sim q(\mathcal{G}^{(t)} \mid \mathcal{G}^{(t-1)})$ with Intervention Constraint.
    (b) Parameter estimate for current graph: $\Theta^{cand} = \text{Estimation}(\mathcal{G}^{cand}, \mathcal{D}_{\mathcal{I}})$
    (c) Calculate Acceptance Probability:$\alpha(\mathcal{G}^{cand} \mid \mathcal{G}^{(t-1)}) = \min\left\{1, \frac{q(\mathcal{G}^{(t-1)}|\mathcal{G}^{cand})P(\mathcal{G}^{cand}|\mathcal{D}_{\mathcal{I}})}{q(\mathcal{G}^{cand}|\mathcal{G}^{(t-1)})P(\mathcal{G}^{(t-1)}|\mathcal{D}_{\mathcal{I}})}\right\}$
    (d) Randomly Sample: $u \sim \text{Uniform}[0, 1]$
    (e) Accept the proposal: $\mathcal{G}^{(t)} \leftarrow \mathcal{G}^{cand}$, **if** $u < \alpha$ **then** Reject the proposal: $\mathcal{G}^{(t)} \leftarrow \mathcal{G}^{t-1}, \Theta^{(t)} \leftarrow \Theta^{(t-1)}$
    (f) Forward Sampling data: $\mathcal{D}^{(t)} \leftarrow \text{Sampling}(\mathcal{G}^{(t)}, \Theta^{(t)})$

---

---

**Algorithm 2** Optimized `IS-MCMC` Algorithm Workflow

---

❶  Initialize Seed Graph with Proxy Algorithm: $\mathcal{G}^{(0)} \leftarrow \text{proxy}(\mathcal{D}_{\mathcal{I}})$    ▷ *Good Initial State*
    Estimating CPT parameters by maximum likelihood: $\Theta^{(0)} = \text{MLE}(\mathcal{G}^{(0)}, \mathcal{D}_{\mathcal{I}})$

**Parallel** $i \in 1, 2, \ldots, N$ **do**:    ▷ *Efficient Parallel Chains*
  **for** iteration $t = 1, 2, \ldots, L$ **do**:

  ❷  Propose: $\mathcal{G}^{cand} \sim q(\mathcal{G}^{(t)} \mid \mathcal{G}^{(t-1)})$ with Intervention Constraint.    ▷ *Intervention Constraints*

  ❸  Re-parameterize according to the previous graph ( B.1.1): $\Theta^{cand} = \text{Estimation}(\mathcal{G}^{cand}, \Theta^{(t-1)})$    ▷ *Parameters Reuse*

  ❹  Calculate Acceptance Probability:$\alpha(\mathcal{G}^{cand} \mid \mathcal{G}^{(t-1)}) = \min\left\{1, \frac{q(\mathcal{G}^{(t-1)}|\mathcal{G}^{cand})P(\mathcal{G}^{cand}|\mathcal{D}_{\mathcal{I}})}{q(\mathcal{G}^{cand}|\mathcal{G}^{(t-1)})P(\mathcal{G}^{(t-1)}|\mathcal{D}_{\mathcal{I}})}\right\}$

  ❺  Randomly Sample: $u \sim \text{Uniform}[0, 1]$

    Accept proposal: $\mathcal{G}^{(t)} \leftarrow \mathcal{G}^{cand}, \Theta^{(t)} \leftarrow \Theta^{cand}$ **if** $u < \alpha$ **else** Reject proposal: $\mathcal{G}^{(t)} \leftarrow \mathcal{G}^{t-1}, \Theta^{(t)} \leftarrow \Theta^{(t-1)}$

  ❻  Forward Sampling data: $\mathcal{D}^{(t)} \leftarrow \text{Sampling}(\mathcal{G}^{(t)}, \Theta^{(t)})$

---

We first present the standard version of the IS-MCMC algorithm, as outlined in Algorithm 1. The first step involves initializing the seed graph, which is typically sampled from the prior distribution of the augmented graph. The second step

estimates the parameters of the conditional probability tables using maximum likelihood estimation based on the initial graph structure and the augmented data. The main loop of the algorithm consists of three parts: (1) generating a proposed (or candidate) sample graph from the proposal distribution with intervention constraint; (2) calculating the acceptance probability using the acceptance function $\alpha(\cdot)$, based on the proposal distribution and the posterior probability; (3) accepting the candidate sample with probability $\alpha$ (the acceptance probability), or rejecting the candidate sample with probability $1 - \alpha$.

However, as the number of nodes increases, the computational efficiency of the standard IS-MCMC algorithm becomes untenable. Therefore, for the IS-MCMC algorithm applied to large graphs, we enhance it from two aspects. For the initial random seed graph in the first step, we recommend using proxy algorithms such as JCI+BLIP, which can provide a good starting point that is closer to the target graph structure, thereby accelerating the convergence rate. For the iterative MCMC process in the second step, we suggest parallelizing the computation by simultaneously executing multiple chains. Each chain performs consecutive edge perturbations, followed by sequential accept-reject sampling, to conduct approximate computations. The optimized IS-MCMC workflow is shown in Algorithm 2. If not specified otherwise, we use the optimized IS-MCMC method as the default setting.

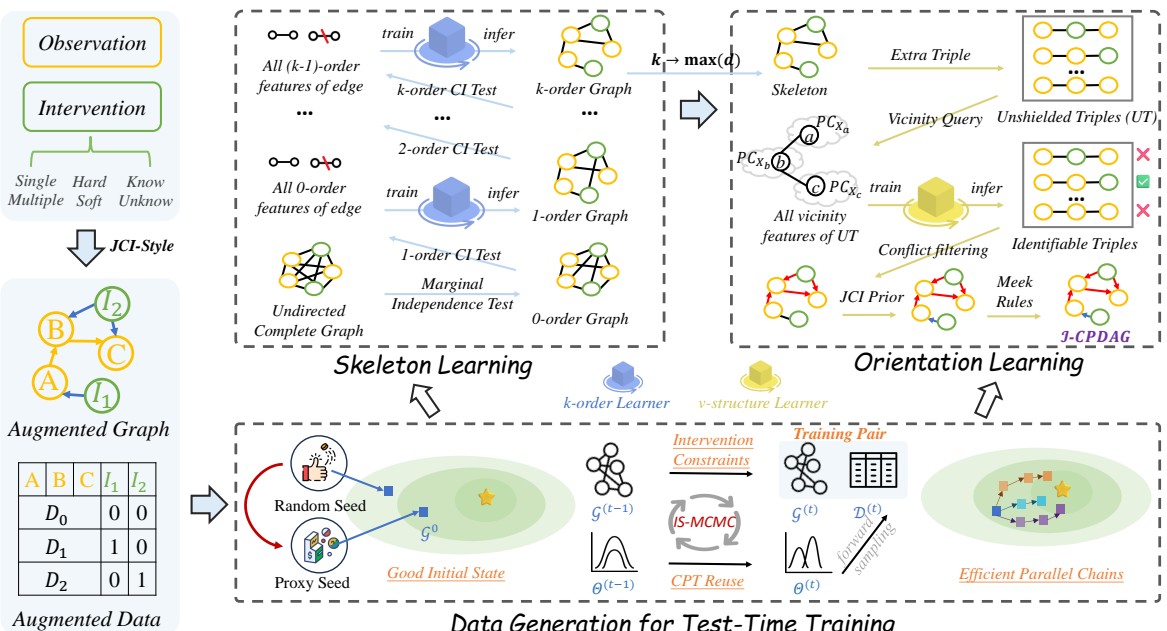

*Figure 9.* The detailed framework of `TICL`.

## B.2. Two-phase Supervised Causal Learning

The two-phase supervised learning approach of `TICL` is inspired by the PC algorithm (Spirtes & Glymour, 1991). We first describe the PC algorithm in detail and then reinterpret it from a machine learning perspective, providing a formalization. We also provide detailed feature engineering and techniques. As shown in Figure 9, we further present a more detailed framework diagram, which includes the specific process of two-phase learning.

### B.2.1. REVISITING THE PC ALGORITHM

The PC algorithm consists of three main phases, as shown in Algorithm 3. Phase 1 identifies the skeleton and the separating sets, determining the existence of edges. Phase 2 orients the unshielded triples in the skeleton based on the separating sets, establishing edge directionality. Finally, phase 3 further refines the edge directionality using heuristic Meek Rules.

In phase 1, we start with a complete undirected graph $\mathcal{C}$. This graph is then sparsified through iterative conditional tests information **CI**, where an edge $X_i - X_j$ is removed if $X_i$ is conditionally independent of $X_j$ given some subset $\mathcal{S}$ of the remaining variables of the current $k$-order graph. These conditional independence queries proceed in a cascading manner, making the algorithm computationally efficient for high-dimensional sparse graphs since we only need to query conditional

---

**Algorithm 3** Pipeline of `PC` Algorithm (Spirtes & Glymour, 1991)

---

**Require:**
    Conditional Independence Information **CI** among Variables.
    Undirected Complete Graph $\mathcal{C}$ Obtained by Variables.

**Output:**
    Separation Sets: $\mathcal{SS}$ (Temporary Product)
    Skeleton: $\mathcal{K}$ (Intermediate Product)
    Unshielded Triple Sets of Skeleton: $\mathcal{U}$ (Temporary Product)
    Partially Directed Acyclic Graph, *i.e.*, PDAG: $\mathcal{P}$ (Intermediate Product)
    Completed Partially Directed Acyclic Graph, *i.e.*, CPDAG: $\mathcal{G}$

**Pipeline:**

1. Adjacency Determination: Find the skeleton $\mathcal{K}$ and separation sets $\mathcal{SS}$ based on undirected complete graph $\mathcal{C}$ and conditional independence information **CI** using Algorithm 4.

2. Orientation Determination: Orient unshielded triples $\mathcal{U}$ in the skeleton $\mathcal{K}$ to derive partially directed acyclic graph $\mathcal{P}$ based on the separation sets $\mathcal{SS}$ using Algorithm 5.

3. In $\mathcal{P}$ orient as many of the remaining undirected edges as possible by repeated application of rules **R1**-**R3** to derive completed partially directed acyclic graph $\mathcal{G}$.

---

independencies up to order $d - 1$, where $d$ is the maximal in degree of the underlying DAG. We summarize this process in Algorithm 4.

---

**Algorithm 4** Pipeline of Adjacency Determination / Phrase 1 of the `PC` Algorithm (Kalisch & Bühlman, 2007)

---

**Require:**
    Undirected Complete Graph $\mathcal{C}$ Induced by Variables.
    Sequential Order $k$.

**Output:**
    Skeleton: $\mathcal{K}$
    Separation Sets: $\mathcal{SS}$

**Pipeline:**
    $k = 1$
    **repeat**

        **for** each adjacent pair $(X_i, X_j)$ in $\mathcal{C}$ **do**
          **for** each subset $\mathcal{S} \subseteq adj(\mathcal{C}, X_i)\backslash\{X_j\}$ or $adj(\mathcal{C}, X_j)\backslash\{X_i\}$ with $|\mathcal{S}| = k$ **do**
            **if** $X_i$ and $X_j$ are conditionally independent given $\mathcal{S}$ **then**
              Delete edge $X_i - X_j$ from $\mathcal{C}$
              Let separation set $\mathcal{SS}_{(X_i, X_j)} = \mathcal{SS}_{(X_j, X_i)} = \mathcal{S}$
            **end if**
          **end for**
        **end for**                       ▷ This process can be supervised for learning

    $k = k + 1$
    **until** all adjacent pairs $(X_i, X_j)$ in $\mathcal{C}$ satisfy $|adj(\mathcal{C}, X_i)\backslash\{X_j\}| < k$

---

In phase 2, it aims to identify V-structures. Specifically, it considers all unshielded triples $\mathcal{U}$ in the skeleton $\mathcal{K}$ and orients an unshielded triple $(X_a, X_c, X_b)$ into a V-structure if and only if $X_c$ is not in the separating set of $X_a$ and $X_b$. We also summarize this process in Algorithm 5.

---

**Algorithm 5** Pipeline of Orientation Determination / Phrase 2 of the `PC` Algorithm (Kalisch & Bühlman, 2007)

**Require:**
    Skeleton: $\mathcal{K}$
    Separation Sets: $\mathcal{SS}$
    Unshielded Triple Sets of Skeleton: $\mathcal{U}$
**Output:**
    Partially Directed Acyclic Graph, *i.e.*, PDAG: $\mathcal{P}$
**Pipeline:**

> **for** each non-adjacent pair $(X_a, X_b)$ with common neighbour $X_c$ in $\mathcal{U}$ **do**
>     **if** $X_c \notin \mathcal{SS}_{(X_a,X_b)}$ **then**
>         Replace $X_a - X_c - X_b$ in $\mathcal{K}$ by $X_a \rightarrow X_c \leftarrow X_b$
>     **end if**
> **end for**          ▷ This process can be supervised for learning

    $\mathcal{P} = \mathcal{S}$

---

In phase 3, heuristic meek rules (Meek, 1995) are further applied iteratively to orient as many of the remaining undirected edges as possible. It contains the following three rules:

- **R1**: If $X_a \rightarrow X_b - X_c$ exists, change $X_b - X_c$ to $X_b \leftarrow X_c$ (to avoid creating a new V-structure).

- **R2**: If $X_a \rightarrow X_b \rightarrow X_c$ exists, change $X_a - X_c$ to $X_a \rightarrow X_c$ (otherwise a directed cycle is created).

- **R3**: If there are $X_a - X_{c1} \rightarrow X_b$, $X_a - X_{c2} \rightarrow X_b$, and $X_{c1}, X_{c2}$ are not adjacent, change $X_a - X_b$ to $X_a \rightarrow X_b$ (otherwise a new v-structure or a directed cycle is created).

### B.2.2. CONNECTING PC AND SUPERVISED MACHINE LEARNING

In addition to phase 3, phase 1 and 2 can be considered as classification tasks regarding the determination of the presence of edges and the orientation of unshielded triples based on extracted conditionally independent features. We continue below with a detailed explanation.

■ From a machine learning perspective, we formalize phase 1 as follows:

**Task:** For the current $k$-oreder graph, classify whether there is an edge between vertices $X_i$ and $X_j$.

**Featurization:** Query a subset $\mathcal{S}$ of the current remaining variables of $k$-oreder graph, and calculate the conditional dependence between $X_i$ and $X_j$ given $\mathcal{S}$:

$$F_{(X_i, X_j)}^{(k)} = \min_{\mathcal{S} \subseteq \mathbf{X} \setminus \{X_i, X_j\}} \{X_i \sim X_j | \mathcal{S}\}$$

**Classifier:** Train a binary classifier for $(k+1)$-order graph using current $k$-order features:

$$C_{skeleton}^{(k+1)}(F_{(X_i,X_j)}^{(k)}) := \begin{cases} adjacent & F_{(X_i,X_j)}^{(k)} \neq 0 \\ non-adjacent & F_{(X_i,X_j)}^{(k)} = 0 \end{cases}$$

■ From a machine learning perspective, we formalize phase 2 as follows:

**Task:** For all unshielded triples $\mathcal{U}$, classify whether each triple $< X_a, X_c, X_b >$ is a v-structure.

**Featurization:** Query all separating sets $\mathcal{SS}$ satisfying $X_a \perp X_b|\mathcal{SS}$, and calculate the existence Boolean feature:

$$F_{<X_a,X_c,X_b>} = \begin{cases} 1 & X_c \in \mathcal{SS} \\ 0 & X_c \notin \mathcal{SS} \end{cases}$$

**Classifier:** Train a binary classifier using features:

$$C_{orientation}(F_{<X_a,X_c,X_b>}) := \begin{cases} v\text{-}structure & F_{<X_a,X_c,X_b>} = 0 \\ non-v\text{-}structure & F_{<X_a,X_c,X_b>} \neq 0 \end{cases}$$

**Theorem B.1.** *(Spirtes and Glymour (Spirtes et al., 2000)) Let the distribution of* $\mathbf{X}$ *be faithful to a DAG* $\mathcal{G}$*, and assume that we are given perfect conditional independence information about all pairs of variables* $(X_i, X_j)$ *in* $\mathbf{X}$ *given subsets* $\mathcal{S} \subseteq \mathbf{X} \setminus \{X_i, X_j\}$. *The output of PC algorithm is the CPDAG that represents* $\mathcal{G}$.

However, in practical applications, conditional independence tests relying on data estimates can encounter issues like limited sample size, data quality issues, and intricate dependency structures (Günther et al., 2022). Thus, it is advisable to employ more systematic featurization procedures to enhance classification performance in a more effective and robust manner.

### B.2.3. SKELETON AND ORIENTATION FEATURE ENGINEERING

In the realm of skeleton learning, drawing upon the experience of (Cheng et al., 2002; Ding et al., 2020; Xiang & Kim, 2013; Ma et al., 2022), we have extracted primary features of the two categories: quantitative $k$-order conditional dependencies and local structural information. For orientation learning, again relying on the experience of (M K & K, 2016; Zanga et al., 2022; Dai et al., 2023), we have extracted main features of the two categories: quantitative unshielded triplet conditional dependencies and local structural information. The intuition behind it is detailed below.

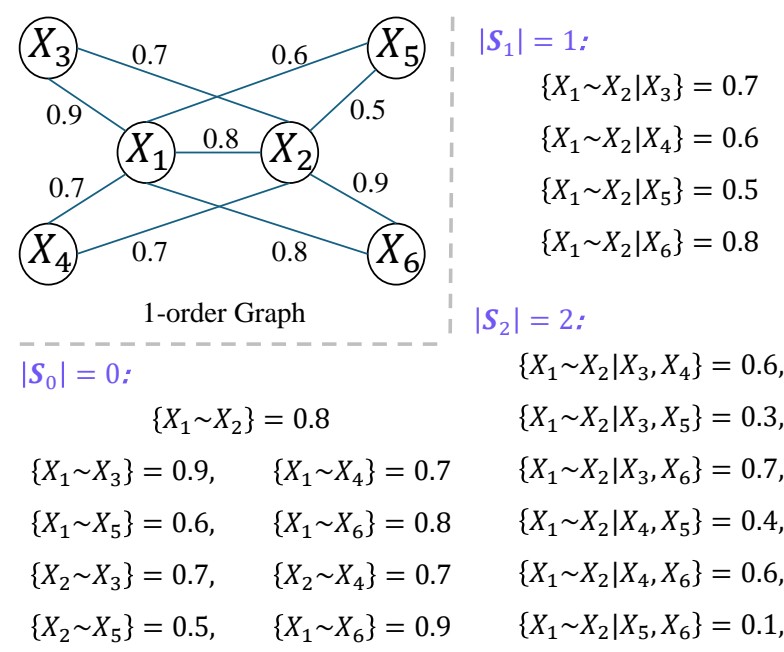

*Figure 10.* Feature extraction example of edge $X_1 - X_2$ in 1-order skeleton graph.

In skeleton inference, we first look for useful features of the type of conditional test information. Intuitively, for the numerical vectors resulting from the conditional tests performed on the current node pairs, which indicate the conditional dependency between nodes, we consider them as primary features. Additionally, when conducting higher-order conditional tests, if the dependency relationships decrease significantly, it suggests that they may be blocked by new nodes, so this residual conditional dependency can be seen as another complementary feature. Considering the structural aspect, conditional

*Table 3.* The features extracted for skeletal learning. As shown in Figure 10, $k = 2$, is used to illustrate the design intuition and computational examples.

| | | SKELETON FEATURE | | |
|---|---|---|---|---|
| TYPE | NAME | CALCULATE | EXAMPLE RAW RESULTS | DIMENSION |
| CI TEST INFORMATION | $k$-ORDER CONDITIONAL DEPENDENCE | $\{X_i \sim X_j\|\mathcal{S}_2\}, \ \|\mathcal{S}_2\|= 2$ | [0.6, 0.3, 0.7, 0.4, 0.6, 0.1] | NOT FIXED $\to$ 15 |
| | RESIDUAL CONDITIONAL DEPENDENCE | $\{X_i \sim X_j\|\mathcal{S}_1\} - \min_{X_q}\{X_i \sim X_j\|\mathcal{S}_2 \cup \{X_q\}\}$, $\|\mathcal{S}_1\|= 1, \ \|\mathcal{S}_2\|= 2$ | [0.4, 0.2, 0.4 ,0.7] | NOT FIXED $\to$ 15 |
| STRUCTURAL INFORMATION | COMPETITIVENESS | $\frac{\|\{X_q\|X_q \in nbd(X_i), C_{k-1}(X_i,X_j) > C_{k-1}(X_i,X_q)\}\|}{\|nbd_{\mathcal{G}_{k-1}}(X_i)\|-1}$, $\frac{\|\{X_q\|X_q \in nbd(X_j), C_{k-1}(X_j,X_i) > C_{k-1}(X_j,X_q)\}\|}{\|nbd_{\mathcal{G}_{k-1}}(X_j)\|-1}$, $C_{k-1}(X_i, X_j)$ | [0.5, 0.75, 0.8] | 3 |
| | DEGREE | $deg(X_i), deg(X_j)$ | [4, 4] | 2 |
| | DENSITY | $\frac{\|nbd(X_i) \cap nbd(X_j)\|}{\min\left(\|nbd(X_i)\|, \|nbd(X_j)\|\right)}$ | [1] | 1 |

independence may be influenced by the information of the local graph structure in which the current node pairs are located. Generally, the sparser the local graph is, the less likely it is to be disconnected by conditional independence tests. Therefore, we select three main features for measurement, including the relative competitiveness of adjacent edges between node pairs, the degrees of node pairs, and the overlapping density of adjacent edges of node pairs. In Figure 10, we provide an example of a 1-order skeleton graph, based on which we further summarize the types of feature variables and calculation methods in Table 3.

In the unshielded triplet orientation, we also seek useful features of the condition test information type. Unlike relying solely on specific triplet condition test information, we extend it to the neighborhood. Specifically, in addition to testing variables $X_a$ and $X_b$ themselves, we also extend the set variables of their parents and children, $PC_{X_a}$ and $PC_{X_b}$, as testing variable domains. For the condition variables, in addition to the empty set $\emptyset$, **SS**, $X_c$, we further extend the set variables of the parents and children of $X_c$ to the condition domain, including their element-wise union **SS** $\vee X_c$, **SS** $\vee PC_{X_c}$. Therefore, by selecting a pair of variables from the variable domain and a condition from the condition domain, we obtain a total of $4 \times 6 = 24$ neighborhood conditional dependency features. Furthermore, considering structural information, we overlap numerical and scale numerical measures of the size and overlap features of the sets of the variable domain and the condition domain to reflect the potential sparsity of the local graph structure. As shown in Figure 11, we provide an example of directional feature extraction, based on which we further summarize the types of feature variables and calculation methods in Table 4.

It is noted that the conditional dependence features mentioned above may be dynamically changing, which does not conform to the fixed-size features required by most traditional machine learning models. However, this can be easily addressed through classical kernel mean embedding techniques (Smola et al., 2007) to obtain fixed-length embedding features:

$$\frac{1}{|F|} \sum_{z \in F} (\cos(< w_j, z > +b_j))_{j=1}^m \in \mathbb{R}^m.$$

Here, $m = 15$, which signifies that each extended feature $F$ now possesses a fresh embedding dimension of 15. Furthermore, we incorporate an extra set of five statistics comprising maximum, minimum, mean, standard deviation, and set size. Consequently, we merge all these derived features to construct the input feature for the model.

Additionally, various methods exist for assessing conditional dependence, such as determining the p-value through testing for conditional independence or using conditional mutual information (Cover, 1999). When dealing with categorical variables,

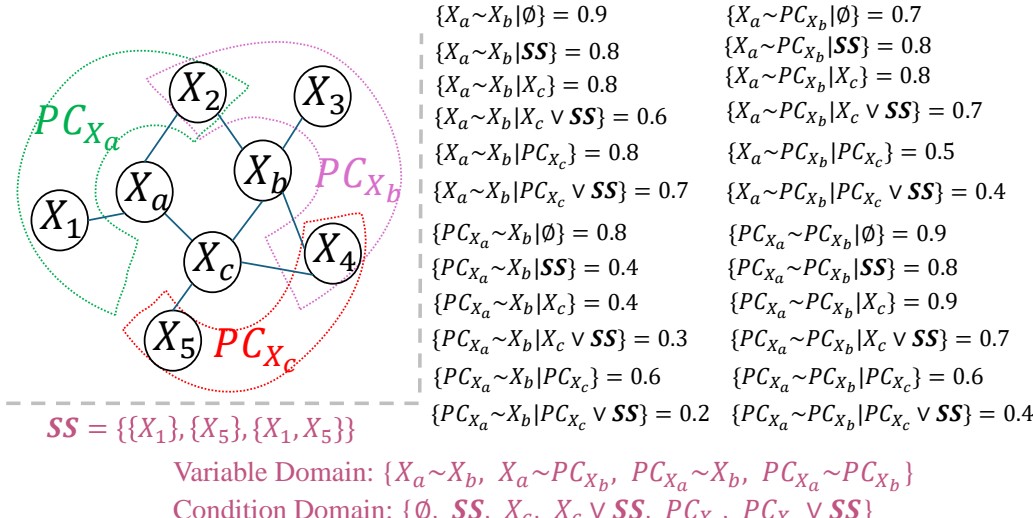

$$\{X_a \sim X_b | \emptyset\} = 0.9 \qquad \{X_a \sim PC_{X_b} | \emptyset\} = 0.7$$
$$\{X_a \sim X_b | \mathbf{SS}\} = 0.8 \qquad \{X_a \sim PC_{X_b} | \mathbf{SS}\} = 0.8$$
$$\{X_a \sim X_b | X_c\} = 0.8 \qquad \{X_a \sim PC_{X_b} | X_c\} = 0.8$$
$$\{X_a \sim X_b | X_c \vee \mathbf{SS}\} = 0.6 \qquad \{X_a \sim PC_{X_b} | X_c \vee \mathbf{SS}\} = 0.7$$
$$\{X_a \sim X_b | PC_{X_c}\} = 0.8 \qquad \{X_a \sim PC_{X_b} | PC_{X_c}\} = 0.5$$
$$\{X_a \sim X_b | PC_{X_c} \vee \mathbf{SS}\} = 0.7 \qquad \{X_a \sim PC_{X_b} | PC_{X_c} \vee \mathbf{SS}\} = 0.4$$

$$\{PC_{X_a} \sim X_b | \emptyset\} = 0.8 \qquad \{PC_{X_a} \sim PC_{X_b} | \emptyset\} = 0.9$$
$$\{PC_{X_a} \sim X_b | \mathbf{SS}\} = 0.4 \qquad \{PC_{X_a} \sim PC_{X_b} | \mathbf{SS}\} = 0.8$$
$$\{PC_{X_a} \sim X_b | X_c\} = 0.4 \qquad \{PC_{X_a} \sim PC_{X_b} | X_c\} = 0.9$$
$$\{PC_{X_a} \sim X_b | X_c \vee \mathbf{SS}\} = 0.3 \qquad \{PC_{X_a} \sim PC_{X_b} | X_c \vee \mathbf{SS}\} = 0.7$$
$$\{PC_{X_a} \sim X_b | PC_{X_c}\} = 0.6 \qquad \{PC_{X_a} \sim PC_{X_b} | PC_{X_c}\} = 0.6$$
$$\{PC_{X_a} \sim X_b | PC_{X_c} \vee \mathbf{SS}\} = 0.2 \quad \{PC_{X_a} \sim PC_{X_b} | PC_{X_c} \vee \mathbf{SS}\} = 0.4$$

$$\mathbf{SS} = \{\{X_1\}, \{X_5\}, \{X_1, X_5\}\}$$

Variable Domain: $\{X_a \sim X_b, \ X_a \sim PC_{X_b}, \ PC_{X_a} \sim X_b, \ PC_{X_a} \sim PC_{X_b}\}$
Condition Domain: $\{\emptyset, \ \mathbf{SS}, \ X_c, \ X_c \vee \mathbf{SS}, \ PC_{X_c}, \ PC_{X_c} \vee \mathbf{SS}\}$

*Figure 11.* Orientation feature extraction example of unshielded triple $X_a - X_c - X_b$.

*Table 4.* The features extracted for orientation learning, as shown in Figure 11, are used to illustrate the design intuition and computational examples.

| | | ORIENTATION FEATURE | | |
|---|---|---|---|---|
| TYPE | NAME | CALCULATE | EXAMPLE RAW RESULTS | DIMENSION |
| CI TEST INFORMATION | VICINITY CONDITIONAL DEPENDENCE | $\{\text{VARIABLE} \mid \text{CONDITION}\}$, VARIABLE: $\{X_a \sim X_b, X_a \sim PC_{X_b}, PC_{X_a} \sim X_b, PC_{X_a} \sim PC_{X_b}\}$ CONDITION: $\{\emptyset, SS, X_c, X_c \vee SS, PC_{X_c}, PC_{X_c} \vee SS\}$ | [0.9, 0.8, 0.8, 0.6, 0.8, 0.7, 0.7, 0.8, 0.8, 0.7, 0.5, 0.4, 0.8, 0.4, 0.4, 0.3, 0.6, 0.2, 0.9, 0.8, 0.9, 0.7, 0.6, 0.4] | 24 |
| STRUCTURAL INFORMATION | OVERLAP | $\frac{|set_1 \cap set_1|}{\min(|set_1|, |set_2|)}$, $set \in PC_{X_a}, PC_{X_b}, PC_{X_c}, \mathbf{SS}$, WE ALSO ADD $set_1 = X_c, set_2 = \mathbf{SS}$ | [0.5, 0, 0.5, 0.5, 0, 0.5 ,1] | 7 |
| | SCALING | $\#PC_{X_a}, \#PC_{X_b}, \#PC_{X_c}$, $\#\mathbf{SS}, \frac{1}{\#\mathbf{SS}} \sum_{SS \in \mathbf{ss}} \#SS$ | $[2, 3, 2, 3, \frac{4}{3}]$ | 5 |

the G2 test is a suitable option (Agresti, 2012). In our approach, we employ an approximate version of the G2 statistic and rely on p-values to evaluate conditional dependence.

It is worth noting that the p-value can become insignificant due to the double-precision bit limit in computers. To overcome this issue, transforming p-values can be employed to mitigate such limitations and offer a means to evaluate conditional dependency. To begin with, we define the complementary error function as:

$$g(z) = 1 - \frac{2}{\sqrt{\pi}} \int_0^z e^{-t^2} \, \mathrm{d}t,$$

and utilize the quantity $z$ as the inverse of $g$:

$$z = g^{-1}(x).$$

By applying the non-linear transformation of $g^{-1}$ to a given p-value $x$, we derive a re-scaled quantity that enhances the

assessment of conditional dependency. In essence, $z$ can be interpreted as a measure akin to z-scores in a standard normal distribution; for example, if the p-value is 0.01, then $z = 3$, as a value of 3-sigma signifies that the probability of data falling within a 3-sigma range in a normal distribution is 0.99.

## C. Convergence of Inteventional Structural-MCMC

In this section, we first confirm the convergence of the posterior distribution in the IS-MCMC algorithm based on the Structure MCMC framework. Furthermore, we provide a detailed discussion of the key factors considered in optimizing the IS-MCMC process. Besides, we also provide visualization experiments on convergence analysis.

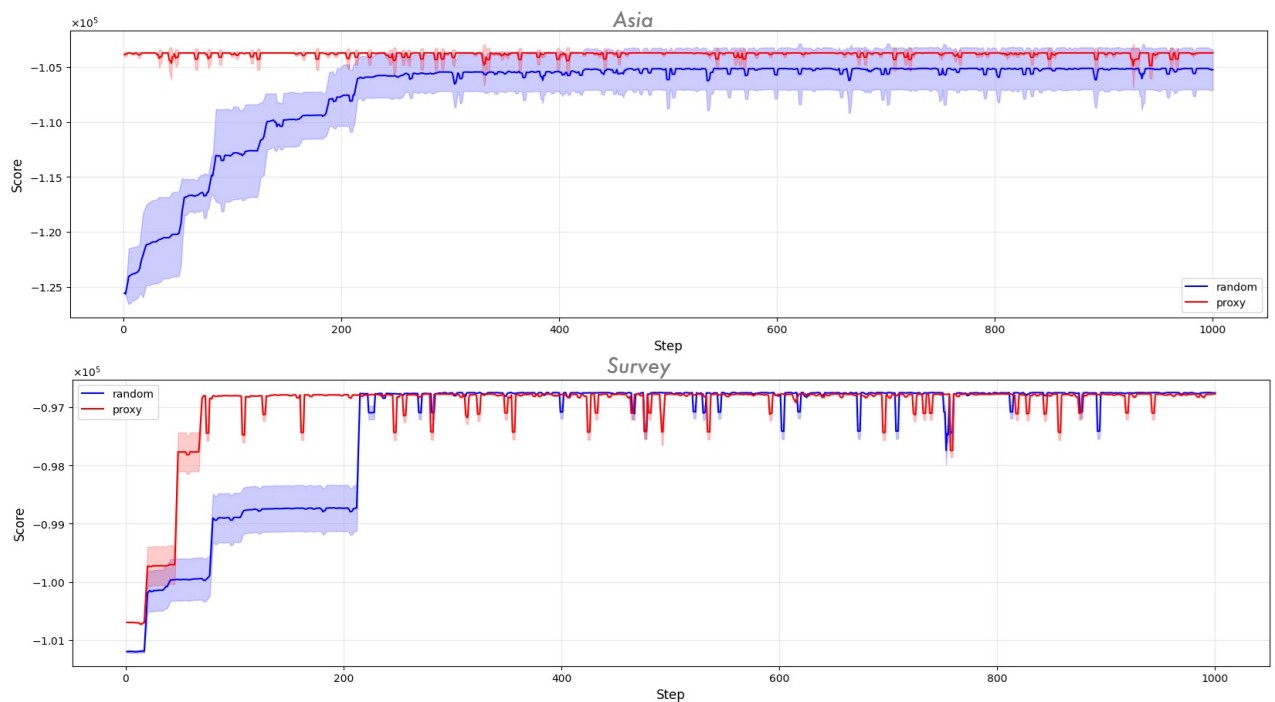

*Figure 12.* Convergence Visualization Analysis on different datasets.

### C.1. Guarantee of Posterior Distribution Convergence

Our IS-MCMC algorithm follows a standardized Structure MCMC process (Madigan et al., 1995; Su & Borsuk, 2016; Kuipers & Moffa, 2017) in the intervention-augmented graph space. Please refer to Algorithm 2, which corresponds to Steps ❶ - ❻ in the main text as a general introduction to the MCMC process.

By employing the Metropolis-Hastings sampler within the standardized Structure MCMC framework, the Markov chain is guaranteed to have a stationary distribution equal to the posterior distribution $P(\mathcal{G} \mid \mathcal{D})$ (Su & Borsuk, 2016).

It is important to note that, under the premise of this theoretical guarantee, the novelty of our work lies in proposing the use of the posterior distribution as the target for acquiring training samples and optimizing feasible solutions in intervention scenarios.

### C.2. Optimization Considerations for IS-MCMC

However, two critical factors need to be addressed. First, managing the time complexity of this process is crucial for maintaining efficient inference. Second, ensuring the effectiveness of the IS-MCMC process on the augmented graph.

- **From the TTT-driven perspective on efficiency:** Since we leverage the "test-time" phase, the convergence speed of MCMC is particularly important. We observed that a good initial state enables the Markov chain to converge faster, as evidenced by Figure 5 in the experimental section of our paper. Additionally, we modified the algorithm to use

parallel multi-chain sampling, allowing for more efficient exploration of the graph space, as shown in Figure 8 of our experimental results.

- **From the JCI-driven perspective on adaptability:** The MCMC process operates on augmented graphs rather than standard causal graphs. Thus, additional constraints are required to ensure the validity of the augmented graph. To address this, we introduced intervention constraints to ensure consistency of the augmented graph within the JCI framework.

In summary, our IS-MCMC algorithm effectively addresses multiple challenges under the TTT + JCI framework and demonstrates that using posterior estimates as training data is highly beneficial for test-time SCL.

### C.3. Convergence Visualization Analysis of IS-MCMC

As shown in Figure 12, we plotted the goodness-of-fit scores using random seed and proxy seed strategies on the Asia and Survey datasets, respectively. By treating the initial state as low-quality, the IS-MCMC strategy gradually improves the score until it converges to a stationary state. The proxy seed accelerates this process, so in practice, our method can significantly benefit from this process, especially in test-time training.

## D. Identifiability Theory of Two-Phase SCL

### D.1. Intuitive Analysis

#### D.1.1. OBSERVATION

We use the observations from phase 2 as an illustration. As mentioned in Section B.2, we formalize the v-structure orientation of PC as a specific classification problem, that is, determining whether the non-shielded triple $\langle X_a, X_c, X_b \rangle$ forms a v-structure by classifier $C_{PC}$.

The logic of PC's orientation is clearly asymptotically correct (*i.e.*, the **CI** test becomes fully accurate when the number of records goes to infinity). However, in practice, when the number of records is finite, its empirical performance is not satisfactory. Thus, further enhancements, such as *Majority-PC* (MPC) (Colombo et al., 2014), have been developed to address this limitation.

MPC is a sample-based enhancement of PC's orientation, achieving better performance with finite samples. Instead of identifying only one separating set $\mathcal{SS}$, MPC finds all possible separating sets $\mathcal{SS}$ and counts how many of them contain $X_c$. The logic can be recast as follows:

**Featurization:** Finds all separating sets $\mathcal{SS}$ of $X_a, X_b$, and defines a real-valued feature:

$$F_{<X_a, X_c, X_b>} = \frac{|\{S_i | X_c \in S_i \in \mathcal{SS}\}|}{|\mathcal{SS}|}$$

**Classifier:** Train a binary classifier using features:

$$C_{MPC}(F_{<X_a, X_c, X_b>}) := \begin{cases} v\text{-}structure & F_{<X_a, X_c, X_b>} \le 0.5 \\ non-v\text{-}structure & F_{<X_a, X_c, X_b>} > 0.5 \end{cases}$$

Theoretically, both PC and MPC, as "hand-crafted" classifiers, are asymptotically correct. However, in practice, MPC exhibits greater complexity in its classification mechanism compared to PC, resulting in improved empirical performance. Nonetheless, from a machine learning perspective, both PC and MPC remain "simple" in terms of their feature representation and classification strategies.

#### D.1.2. MOTIVATION

The primary motivation for modifying PC into a PC-like SCL method is to leverage *both theoretical identifiability and enhanced empirical performance*:

- **Theoretical Guarantees.** Our PC-like SCL approach *retains* the asymptotic properties of the original PC algorithm. Specifically, the method detects the correct CPDAG (or $\mathcal{I}$-CPDAG in the interventional setting) when the sample size approaches infinity. This ensures theoretical identifiability.

- **Empirical Performance.** In finite-sample scenarios, where PC's reliance on conditional independence (CI) tests can lead to errors, the SCL approach outperforms traditional methods. By combining feature-rich representations with a robust classification mechanism, SCL achieves superior empirical results, as demonstrated in our paper, by comparing `TICL` with PC-JCI or other non-SCL methods. Similar empirical evidences are also presented in prior works (Li et al., 2020; Ma et al., 2022; Dai et al., 2023).

### D.2. Advantage

Building on these observations and motivation analysis, our goal is to design a PC-like SCL algorithm that maintains theoretical asymptotic correctness while encouraging a more systematic feature representation and enabling the learning of more sophisticated classification mechanisms. This approach aims to make our method more "robust" than PC, achieving superior empirical performance. Specifically, Our PC-Like SCL method generalizes these ideas:

- **Featurization:** SCL extracts a richer set of features that capture conditional dependencies and structural patterns around $\langle X_a, X_c, X_b \rangle$.

- **Classifier:** Instead of relying on heuristic rules like $C_{PC}$ or $C_{MPC}$, we train a classifier on synthetic data. The model learns complex interactions among features, avoiding the limitations of error-prone conditional independence (CI) tests.

In the asymptotic regime, the learned classifier converges to an equivalent, theoretically correct solution like $C_{PC}$ or $C_{MPC}$. However, in practical settings, SCL's broader feature set and data-driven optimization enable significantly better empirical performance. For example, Table 1 and 2 compare JCI+SCL with JCI-PC and other non-SCL methods, clearly showing SCL's superiority.

### D.3. Theoretical Analysis

#### D.3.1. RESTATING THE PC-LIKE SCL PROCESS

The PC-like SCL approach proposed in our paper consists of two phases:

- **Phase 1: Skeleton Learning.** A binary classifier $C_1$ takes the feature set (as detailed in Table 3 and Figure 10) as input and predicts the existence of edges between all pairs of nodes.

- **Phase 2: Orientation Learning.** A binary classifier $C_2$ takes the feature set (as detailed in Table 4 and Figure 11) as input and predicts the v-structure for each unshielded triple (UT), using the skeleton learned from Phase 1.

Finally, Meek's rules are applied to orient as many causal directions as possible.

We aim to prove that both $C_1$ and $C_2$ are asymptotically correct (*i.e.*, they output results equivalent to those of the PC algorithm when the sample size approaches infinity). For simplicity, we demonstrate the proof for $C_2$, as the proof for $C_1$ follows the same principle.

#### D.3.2. ASYMPTOTIC CORRECTNESS OF THE PC-LIKE SCL APPROACH

**Definition D.1** (Overlap Coefficient). $\mathrm{OLP}(\mathbf{A}, \mathbf{B}) := |\mathbf{A} \cap \mathbf{B}| / \min(|\mathbf{A}|, |\mathbf{B}|)$, where $\mathbf{A}$ and $\mathbf{B}$ are two sets of variables.

**Definition D.2** (Discriminative Predicate). A discriminative predicate is a binary predicate function defined over the domain of $C_2$'s feature vector. It can be regarded as a special classifier with a predefined mechanism.

**Definition D.3** (Strong Predicate). A strong discriminative predicate satisfies the following two criteria when applied to a UT's feature vector: (a) It evaluates to **'true'** if the UT forms a v-structure. (b) It evaluates to **'false'** if the UT does not form a v-structure.

A strong predicate is ***asymptotically correct*** because its output aligns with that of the PC algorithm when the sample size approaches infinity. The key statement here is that there exists at least one strong predicate, meaning we can construct a static classifier using the proposed feature set to achieve asymptotic correctness.

**Lemma D.4** (Existence of a Strong Predicate). *For a canonical dataset with infinite samples, the predicate* $\mathrm{OLP}(X_c, \mathcal{S}) = 0$ *is a strong discriminative predicate. It corresponds precisely to the v-structure detection logic of the PC algorithm.*

*Proof.* According to the PC algorithm's asymptotic correctness theorem:

For a canonical dataset with infinite samples (assuming faithfulness and perfect conditional independence information):

- If an unshielded triple $\langle X_a, X_c, X_b \rangle$ forms a v-structure, then $X_c$ does **not** belong to any separation set of $(X_a, X_b)$.

- If $\langle X_a, X_c, X_b \rangle$ does not form a v-structure, then $X_c$ belongs to **all** separation sets of $(X_a, X_b)$.

In this setting (infinite samples), there is no ambiguity: $X_c$ is either in all separation sets or in none of them.

Now consider the predicate $\mathrm{OLP}(X_c, \mathcal{S}) = 0$, which evaluates to **true** if and only if $X_c \notin \mathcal{S}$ for all $\mathcal{S} \in \mathcal{SS}$.

- This implies that $\langle X_a, X_c, X_b \rangle$ is a v-structure.

- Therefore, $\mathrm{OLP}(X_c, \mathcal{S}) = 0$ is a strong predicate because it perfectly discriminates v-structures from non-v-structures.

$\square$

**Theorem D.5** (Asymptotic Correctness of $C_2$). *By employing a learning model with the universal approximation property, the classifier $C_2$ is asymptotically correct when classifying a canonical dataset with infinite samples.*

*Proof.* From Lemma D.4, there exists a strong discriminative predicate $P$ that achieves zero loss on a canonical dataset with infinite samples.

- Given $P$ as the ground truth for v-structure detection, a machine learning model with universal approximation capability can approximate $P$ to arbitrary precision, achieving performance no worse than $P$.
- Additionally, there may exist multiple strong predicates that satisfy the criteria of Definition D.3. Therefore, $C_2$ can converge to any one of these strong predicates in the asymptotic regime.

$\square$

**Summary:** The theoretical guarantees demonstrate that the PC-like SCL approach, by leveraging the universal approximation capabilities of learning-based classifiers, retains the asymptotic correctness of the original PC algorithm while enabling superior empirical performance on finite data.

# E. Experimental Details

## E.1. Benchmark

Our experiments are carried out on 14 different causal graph datasets inspired by real-world applications from bayesian network repository [2]. All discrete networks undergo thorough quality checks and necessary repairs, ensuring that the sum of all conditional probability distributions is 1, there are no single-level virtual nodes, and there are no dependencies. The statistics of these bayesian networks are shown in Table 5. We also provide visualizations of some causal graphs, as shown in Figure 13.

## E.2. Baselines

The development history of causal discovery from interventional data has been a gradual and incremental process. Here is a detailed introduction to some key works in this field:

- **GIES (Hauser & Bühlmann, 2012):** *This paper first extends the concept of Markov equivalence of Directed Acyclic Graphs for the first time to the case of interventional distributions arising from multiple interventional experiments.* It further demonstrates that under reasonable assumptions of interventional experiments, the intervened Markov equivalence defines a more refined DAG partition than the observed Markov equivalence, thereby enhancing the identifiability of causal models. Moreover, they generalize the greedy equivalence search algorithm, proposing a greedy interventional equivalence search algorithm for regularized maximum likelihood estimation under such intervened conditions.
- **IGSP (Wang et al., 2017; Yang et al., 2018):** In this paper, the authors propose two algorithms that utilize both observational and interventional data with consistency guarantees, and prove their consistency under the faithfulness assumption. These algorithms are intervention-adapted versions of the Greedy SP algorithm, and they are non-parametric, which makes them applicable to the analysis of non-Gaussian data as well. Subsequently, *they first extend these*

---

[2]https://www.bnlearn.com/bnrepository/

*Table 5.* Statistics and description of bayesian networks we used.

| NETWORK | #NODES | #EDGES | #PARAMETERS | MAX IN-DEGREE | AVG. DEGREE |
|---|---|---|---|---|---|
| EARTHQUAKE | 5 | 4 | 10 | 2 | 1.60 |
| SURVEY | 6 | 6 | 21 | 2 | 2.00 |
| ASIA | 8 | 8 | 18 | 2 | 2.00 |
| SACHS | 11 | 17 | 178 | 3 | 3.09 |
| CHILD | 20 | 25 | 230 | 2 | 2.50 |
| INSURANCE | 27 | 52 | 1,008 | 3 | 3.85 |
| WATER | 32 | 66 | 10,083 | 5 | 4.12 |
| MILDEW | 35 | 46 | 540,150 | 3 | 2.63 |
| ALARM | 37 | 46 | 509 | 4 | 2.49 |
| BARLEY | 48 | 84 | 114,005 | 4 | 3.50 |
| HAILFINDER | 56 | 66 | 2,656 | 4 | 2.36 |
| HEPAR2 | 70 | 123 | 1,453 | 6 | 3.51 |
| WIN95PTS | 76 | 112 | 574 | 7 | 2.95 |
| PATHFINDER | 109 | 195 | 72,079 | 5 | 3.58 |

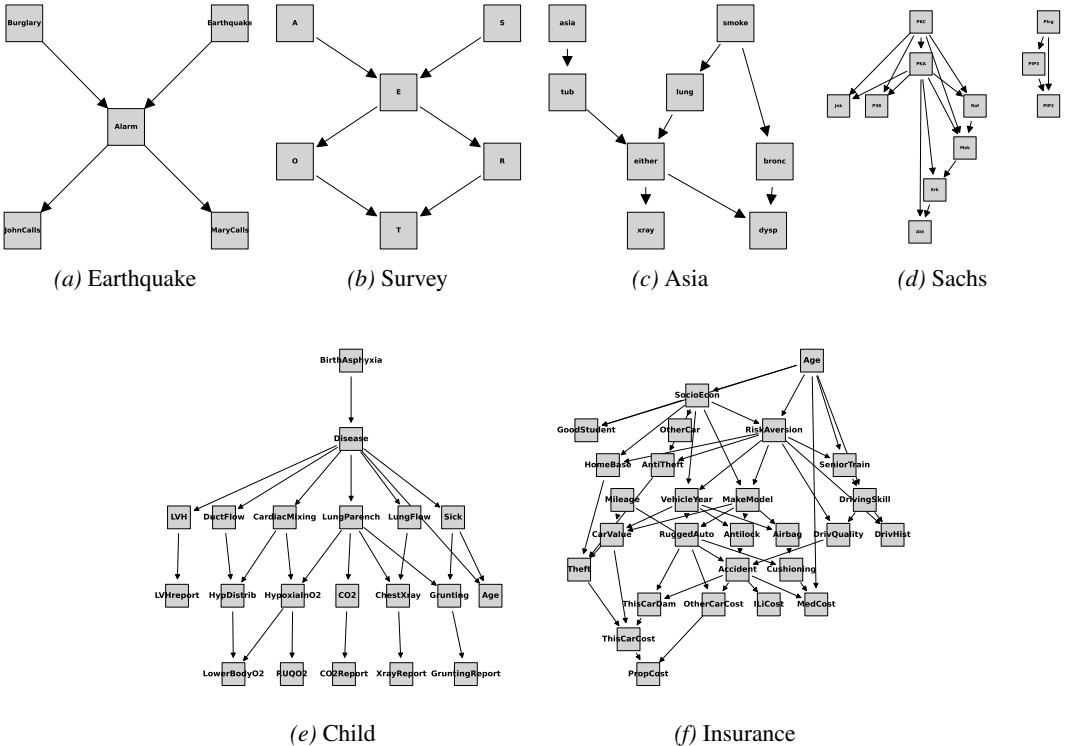

*(a)* Earthquake  *(b)* Survey  *(c)* Asia  *(d)* Sachs

*(e)* Child  *(f)* Insurance

*Figure 13.* Graph Visualizations.

*identifiability results to general interventions that can modify the dependencies between the target variable and its causes without eliminating them,* and propose the first consistent algorithm for learning a DAG in such an environment.

- **UT-IGSP (Squires et al., 2020):** In this paper, *the authors further extend interventions to unknown scenarios, that is, the problem of estimating causal Directed Acyclic Graph models from a mixture of observational and interventional data when the intervention targets are partially or completely unknown.* They describe the intervened Markov equivalence classes

of DAGs that can be identified from interventional data with unknown intervention targets. Additionally, they propose a provably consistent algorithm for learning the intervened Markov equivalence class from such data. The algorithm greedily searches the permutation space to minimize a novel scoring function. This algorithm is also non-parametric.

- **DCDI (Brouillard et al., 2020):** In this paper, *the authors first introduce a theoretically grounded method for learning causal structures based on neural networks that can utilize interventional data.* They present two instances of this approach: one that relies on normalizing flows as a universal density approximator. They also demonstrate that the precise maximization of the proposed score will identify the $\mathcal{I}$-Markov equivalence classes of causal graphs for both known and unknown target settings.

- **ENCO (Lippe et al., 2022):** In this paper, the authors explore score-based methods for causal discovery from observational and interventional data. They argue that such methods often require constrained optimization to enforce acyclicity or lack convergence guarantees. Consequently, they formulate graph search as an optimization of the likelihood of independent edges, where the direction of edges is modeled as separate parameters. Thus, they provide convergence guarantees when interventions on all variables are available, without the need to constrain the acyclicity of the scoring function.

- **BaCaDI (Hägele et al., 2023):** In this paper, the authors aim to discuss the issue of causal discovery in the realistic scenario where interventional data is scarce. To address this shortcoming, they provide a principled Bayesian approach that operates within the continuous space of causal Bayesian networks (CBNs) and the latent probability representations of interventions. This enables them to approximate complex joint posteriors through efficient, gradient-based particle variational inference techniques, making it applicable to causal systems with many variables.

Recently, Supervised Causal Discovery, which aims to compress data and map between causal relations using pre-synthesized causal graph datasets, has shown impressive performance in test data. We introduce some key works on supervised causal discovery from intervention data:

- **CSIvA (Ke et al., 2023b):** In this paper, the authors believe that meta-learning enables models to generalize well to data from natural causal Bayesian networks, even with relatively few assumptions made during training on synthetic data. *Therefore, they introduce for the first time a supervised approach to tackle the problem of causal structure induction.* This method maps datasets composed of both observational and interventional samples to a structure. By introducing a novel transformer architecture, they aim to discover relations between variables across samples.

- **SDI (Ke et al., 2019; 2023a):** In this paper, the authors introduce a novel neural network-based method for causal discovery from interventional data, capable of handling unknown interventions. By utilizing two sets of distinct parameters to model the causal mechanisms and the structure of the causal graph, experimental evidence indicates that this method can generalize to unseen interventions and can effectively perform partial graph discovery.

- **AVICI (Lorch et al., 2022):** In this work, the authors posit that designing appropriate scores or tests that capture prior knowledge is challenging. Therefore, they propose amortizing the learning of causal structures, which involves training a variational inference model to directly predict causal structures from observational or interventional data. This approach leverages permutation invariance to exhibit robust generalization capabilities, particularly in the challenging field of genomics.

Due to the limitations inherent in the type of methods and the constraints of the assumptions made, they are generally unable to effectively process interventional data across various experimental scenarios. Here, we further summarize, based on the experiments of their paper, the basic setup of their causal discovery from interventional data, as depicted in Table 6:

Most existing intervention dataset learning methods have typically required a known set of intervention targets, which is often a strong assumption. Therefore, when intervention targets are unknown, this naturally poses a challenge: how to identify intervention nodes individually becomes a key issue in structural learning. This task is of practical significance. For instance, in the realm of gene editing, techniques can cleave off-target genomic sites. Evaluating and detecting off-target effects accurately and devising corresponding strategies constitute significant research directions in current gene editing studies (Manghwar et al., 2020). Limited attention has been given to exploring methods for identifying intervention targets. We provide an introduction as follows:

- **UT-IGSP (Squires et al., 2020):** As mentioned before, the UT-IGSP algorithm learns the intervention targets while learning the causal structure. Although the intervention refines the search space, greedy search of the sparsest permutations is very slow in the case of high-dimensional data, especially when using non-Gaussian conditional independence test.

- **CITE (Varici et al., 2021):** In this paper, the authors primarily address the problem of estimating unknown intervention targets in causal directed acyclic graphs based on observational and interventional data. The focus lies on soft interventions within the framework of linear structural equation models. They propose a scalable and efficient algorithm capable of

*Table 6.* Summary of basic settings for various methods of causal discovery from intervention data

| METHODS | GROUN-TRUTH DAG | | TESTING DATA | | | INTERVENTION EXPERIMENT | | |
|---|---|---|---|---|---|---|---|---|
| | TYPE | #VARIABLE | TYPE | #OBS. | #INT. / PER EXP. | TYPE | #INT. FAMILY | #INT. TARGET |
| GIES | RANDOM GRAPH (2007) (LINEAR GAUSSIAN SEM) | $10 \sim 50$ | CONTINUE | $50 \sim 10$K | $50 \sim 10$K | SINGLE / MULTIPLE KNOW HARD | #VAR. * {.0 / .2 / .4 / .6 / .8 / 1.0} | $1 \sim 4$ |
| IGSP | ER (LINEAR GAUSSIAN SEM) | $10 \sim 20$ | CONTINUE | $1$K $\sim 100$K | $1$K $\sim 100$K | SINGLE / MULTIPLE KNOW HARD / SOFT | $1 \sim 2$ | #VAR. * {0.4 / 0.5} |
| UT-IGSP | ER (LINEAR GAUSSIAN SEM) | 20 | CONTINUE | $1$K $\sim 5$K | $1 \sim 5$K | SINGLE / MULTIPLE KNOW / UNKNOWN HARD / SOFT | KNOW: (5) + UNKNOW: (5) | KNOW: (1) + UNKNOWN: (0 $\sim$ 3) |
| DCDI | ER (LGM / ANM / NN) | 10-20 | CONTINUE | $10$K | $10$K | SINGLE / MULTIPLE KNOW / UNKNOWN HARD / SOFT | #VARIABLE | #VAR. * 0.1 |
| SDI | RANDOM GRAPH (NN) BNLEARN | $3 \sim 13$ | DISCRETE | 2560 (FROM CODE) | 1111 (FROM CODE) | SINGLE KNOW SOFT | #VARIABLE | 1 |
| CSIvA | ER (LGM / ANM / NN) (MLP / DIRICHLET) | $30 \sim 80$ | CONTINUE | $100 \sim 1.5$K | $100 \sim 1.5$K | SINGLE KNOW / UNKNOWN HARD / SOFT | #VARIABLE | 1 |
| ENCO | RANDOM GRAPH (NN) BNLEARN | 25 | DISCRETE | $1$K $\sim 100$K | $20 \sim 200$ | SINGLE KNOW SOFT | #VARIABLE | 1 |
| AVICI | ER / SF / WS / SBM / GRG (LINEAR / RANDOM FOURIER) | 30 | CONTINUE | 200 (FROM CODE) | 20 (FROM CODE) | SINGLE KNOW HARD / SOFT | #VAR. * .5 | 1 |
| BaCaDI | ER / SF / SERGIO (2020) (LINEAR / NOLINEAR GAUSSIAN) | 20 | CONTINUE | 100 | 10 | SINGLE KNOW / UNKNOWN HARD / SOFT | #VARIABLE | 1 |
| **TICL** | BNLEARN | 5-109 | DISCRETE | $2$K $\sim 10$K | $1$K $\sim 10$K | SINGLE / MULTIPLE KNOW / UNKNOWN HARD / SOFT | #VAR. * {.0 / .2 / .4 / .8 } | $1 \sim 3$ |

consistently identifying all intervention targets. The key idea is to estimate the intervention sites based on the difference between precision matrices associated with the observed data set and the intervened data set.

- **PreDITEr** (Varici et al., 2022): After that, the authors of this paper further extend the previous method by eliminating the need to learn the entire causal model, focusing solely on learning intervention targets. The key point is to leverage the sparse changes imposed on the precision matrix of the linear model by intervention measures, composed of a series of precision difference estimation steps. Additionally, they infer the knowledge required to refine observational Markov equivalence classes into interventional MECs.

- **LIT** (Yang et al., 2024): In this paper, the authors tackle for the first time the problem of identifying unknown intervention targets in a multi-environment setting. They further consider cases within the intervention target set that allow for potential confounding factors. They propose a two-stage algorithm to recover exogenous noise and match it with the corresponding endogenous variables. Under the assumption of causal sufficiency, intervention targets can be uniquely identified. In cases where potential confounding factors exist, a candidate intervention target set is provided, offering more information compared to previous works.

As shown in Table 7, we provide a comparison to briefly illustrate the types of unknown intervention experiments supported by the above intervention target detection algorithms.

*Table 7.* Comparison of situations (more suitable) supported by intervention target detection methods.

| METHODS | CAUSAL INSUFFICIENCY | INTERV. FREQUENCY | | INTERV. MECHANISM | | DATA TYPE | | DOMAIN | |
|---|---|---|---|---|---|---|---|---|---|
| | | SINGLE | MULTIPLE | HARD | SOFT | CONTINUE | DISCRETE | SINGLE | MULTIPLE |
| UT-IGSP | ✗ | ✓ | ✓ | ✓ | ✓ | ✓ | ∼ | ✓ | ✓ |
| CITE | ✗ | ✓ | ✓ | ∼ | ✓ | ✓ | ∼ | ✓ | ✗ |
| PreDITEr | ✗ | ✓ | ✓ | ∼ | ✓ | ✓ | ∼ | ✓ | ✗ |
| LIT | ✓ | ✓ | ✓ | ∼ | ✓ | ✓ | ∼ | ✓ | ✓ |

To ensure fair experimentation, algorithms and hyperparameter tuning are involved in each of our experiments. For all baseline algorithms, we explore critical hyperparameters that govern their interactions, and report the best results. Moreover, we use open source code for all algorithms for evaluation, including code from the authors as well as various popular toolkits.

The public code for the baseline is also included at the URL below.

Firstly, we introduce a method specifically designed for causal discovery from observational and interventional data:

- **GIES:** - Score Function: Gaussian BIC, https://github.com/juangamella/gies
- **IGSP:** - Significance $\alpha \in \{1e3, 1e4\}$, CI test $\in \{\text{Gaussian}, \text{kci}, \text{hsic}\}$, https://github.com/uhlerlab/graphical_model_learning
- **UT-IGSP:** - Significance $\alpha \in \{1e3, 1e4\}$, CI test $\in \{\text{Gaussian}, \text{kci}, \text{hsic}\}$, https://uhlerlab.github.io/causaldag/utigsp.html
- **ENCO:** - Sparsity Regularizer $\lambda_{sparse} \in \{0.002, 0.02\}$, Epochs $\in \{50, 100\}$, https://github.com/phlippe/ENCO
- **AVICI:** - Model $\in \{\text{scm-v0}, \text{linear}, \text{rff}, \text{grn}\}$, https://github.com/larslorch/avici

As mentioned in the main text, joint causal inference offers a robust theoretical framework that effectively combines interventional and observational data, simplifying causal discovery algorithms to operate solely on observational setting. We select four distinct algorithmic combinations, with parameter settings as follows:

- **PC** - variant $\in \{\text{original, stable}\}$, Significance $\alpha \in \{0.05, 0.01\}$, CI test $\in \{\text{fisherz}, \text{g2}, \text{chi2}\}$, https://github.com/huawei-noah/trustworthyAI/tree/master/gcastle
- **HC** - Score Function: Bdeu Score, https://github.com/pgmpy/pgmpy
- **BLIP** - Time $\in \{60, 300, \text{\#Variable}\}$, https://cran.r-project.org/web/packages/r.blip/
- **GOLEM** - $\lambda_1 \in \{2e\text{-}2, 2e\text{-}3\}$, $\omega \in \{0.2, 0.3\}$, https://github.com/ignavierng/golem

Finally, for the specific task of intervention target detection, the configuration parameters for the comparative methods are as follows:

- **CITE** - $\lambda_{l1} \in \{1e\text{-}1, 5e\text{-}1\}$, $\text{Parent}_{l1} \in \{5e\text{-}3, 1e\text{-}2, 2e\text{-}2, 3e\text{-}2, 4e\text{-}2, 5e\text{-}2, 6e\text{-}2, 8e\text{-}2, 9e\text{-}2, 1e\text{-}1\}$, https://github.com/bvarici/intervention-estimation
- **PreDITEr** - $\lambda_{l1} \in \{1e\text{-}1, 3e\text{-}1\}$, $\lambda_{pasp} \in \{1e\text{-}1, 2e\text{-}1\}$, https://github.com/bvarici/uai2022-intervention-estimation-latents

In addition to our selected baselines, we also enumerate other classic baselines relevant to causal discovery from intervention and explain why they were not chosen for our study:

- **DCDI:** This method is a well-known and widely used approach for causal discovery using neural networks, relying on normalizing flows as a generic density estimator. Therefore, it is primarily applicable to continuous data types. We found in experimental testing that it almost never converges and operates at an unbearably slow pace in the case of discrete data. Hence, we disregard its results.
- **SDI:** This method is the first to use neural networks and adapt to discrete data types for intervention causal discovery. However, due to its high complexity, it is often challenging to scale to large-scale node graphs. Additionally, according to the original repository code, its hybrid C-language requirement for compiler adaptation leads to abnormal errors. Hence, we disregard its results.
- **CSIvA:** The lack of accessible code and complex parameter settings make it impossible to replicate the findings of the paper.
- **BaCaDI:** Similar reasons to DCDI.
- **LIT:** Similar reasons to CSIvA.

### E.3. Metrics

We design different reasonable metrics for two different tasks, *i.e.*, $\mathcal{I}$-CPDAG discovery and intervention target detection. The detailed introduction is as follows:

$\mathcal{I}$**-CPDAG discovery.** We first calculate Structural Hamming Distance (SHD) at $\mathcal{I}$-CPDAG level. Specifically, SHD is computed between the learned $\mathcal{I}$-CPDAG($\hat{\mathcal{G}}$) and ground truth $\mathcal{I}$-CPDAG($\mathcal{G}$), *i.e.*, the smallest number of edge additions, deletions, direction reversals and type changes (directed vs. undirected) to convert the output $\mathcal{I}$-CPDAG to ground truth $\mathcal{I}$-CPDAG. As is shown in Table 8, SHD is equal to the sum of the number of ✗s in the table.

Different graphs may lead to different causal inference statements and different intervention distributions. To quantify this discrepancy, we use the (pre-) distance between $\mathcal{I}$-CPDAG($\hat{\mathcal{G}}$)s, known as Structural Intervention Distance (SID). The SID is solely based on graphical criteria and quantifies the closeness between two causal graphs according to their corresponding

*Table 8.* SHD calculation details.

| IN PREDICT→ IN GROUND-TRUTH↓ | IDENTIFIABLE (DIRECTED) | | UNIDENTIFIABLE (UNDIRECTED) | MISSING IN SKELETON |
|---|---|---|---|---|
| | RIGHT | WRONG | | |
| IDENTIFIABLE | ① ✓ | ② ✗ | ③ ✗ | ④ ✗ |
| UNIDENTIFIABLE | ⑤ ✗ | | ⑥ ✓ | ⑦ ✗ |
| NONEXIST | ⑧ ✗ | | ⑨ ✗ | ⑩ ✓ |

causal inference statements. Thus, it is highly suitable for assessing graphs used for the computation of interventions. Formally:

$$\mathrm{SID} : (\hat{\mathcal{G}}, \mathcal{G}) \mapsto \#\{(i,j), i \neq j \,|\, the\ intervention\ distribution\ from\ i\ to\ j$$
$$is\ falsely\ estimated\ by\ \hat{\mathcal{G}}\ with\ respect\ to\ \mathcal{G}\}$$

For more detailed calculation algorithms, we recommend readers to refer to (Peters & Bühlmann, 2015) carefully.

F1-score is then calculated based on the identifiable edges of $\mathcal{I}\text{-CPDAG}(\hat{G})$ and $\mathcal{I}\text{-CPDAG}(G)$, where the accuracy (precision) is equal to True Positive Rate (TPR) and the recall (recall) is equal to 1 - False Discovery Rate (FDR). Details about the specific calculation can also refer to Table 8:

$$\text{Precision} = \text{TPR} = \frac{①}{① + ② + ③ + ④},$$

$$\text{Recall} = 1 - \text{FDR} = \frac{①}{① + ② + ⑤ + ⑧},$$

$$\text{F1-Score} = \frac{2 \times \text{Precision} \times \text{Recall}}{\text{Precision} + \text{Recall}},$$

**Intervention Target Detection.** We then use the F1-Score metric to measure the detection of intervention targets. Let the set of all intervention family edges be denoted as $\hat{\mathcal{I}}$, and all predicted intervention target edges be denoted as $\mathcal{I}$. Then we can formally define:

$$\text{Precision} = \frac{\#(\mathcal{I} \cap \hat{\mathcal{I}})}{\#\mathcal{I}},$$

$$\text{Recall} = \frac{\#(\mathcal{I} \cap \hat{\mathcal{I}})}{\#\hat{\mathcal{I}}},$$

$$\text{F1-Score} = \frac{2 \times \text{Precision} \times \text{Recall}}{\text{Precision} + \text{Recall}}.$$

### E.4. Experiments Setting

**Computational Resources.** For conducting experiments in this work, we employed the IS-MCMC algorithm to pre-generate training data for each test dataset. Subsequently, we trained two types of models, namely the skeleton model and the orientation model, based on the data. The skeleton model comprises multiple cascaded models, with a default maximum of 4 order. Our benchmark environment consists of a Linux server equipped with a $1\times$ AMD EPYC 7763 128-Core Processor CPU (512GB memory) and $4\times$ NVIDIA RTX A6000 (48GB memory) GPUs. To carry out benchmark testing experiments, all baselines are set to run for a duration of 12 hours by default, with specific timings contingent upon the method. It is noteworthy that our method does not necessitate GPU computation, although optimization may further expedite processes. Other methods, depending on their implementation choices, may opt for GPU acceleration.

*Table 9.* Basic configuration of our experiments and methods

| CATEGORY | DETAIL | HYPERPARAMETERS OR SETTINGS |
|---|---|---|
| TRAINING DATA | #TRAINING INSTANCES | 400 (DEFAULT)
$100 \sim 1600$ (OTHERS, SEE 4.2) |
| | #SIMPLES / PER INSTANCE | 10K (DEFAULT)
$1\text{K} \sim 20\text{K}$ (OTHERS, SEE 4.2) |
| TESTING DATA | #OBSERVATION SAMPLE | 10K (DEFAULT)
$2\text{K} \sim 10\text{K}$ (OTHERS, SEE F.2.2) |
| | #INTERVENTION SAMPLE | 10K (DEFAULT)
$1\text{K} \sim 10\text{K}$ (OTHERS, SEE F.2.1) |
| INTERVENTION EXPERIMENT | INTERVENTION TYPE | SINGLE + SOFT + UNKNOW (DEFAULT)
SINGLE + HARD + UNKNOW (OTHERS, SEE F.1.1)
MULTIPLE + SOFT + UNKNOW (OTHERS, SEE F.1.2) |
| | #INTERVENTION EXP. | #VAR. $\times$ 0.2 (DEFAULT)
#VAR. $\times \{0.0, 0.4, 0.8\}$ (OTHERS, SEE F.3) |
| INITIAL GRAPH SEED | PROXY ALGORITHM | JCI-BLIP (DEFAULT)
RANDOM GRAPH (OTHERS, SEE 4.2) |
| MODEL THRESHOLD | SKELETON THRESHOLD | $0.5 \sim 0.75$ |
| | ORIENTATION THRESHOLD | 0.1 |

**Basic Configurations.** As shown in Table 9, we provide the basic parameter configurations for our `TICL`, along with the fundamental details of the synthetic data and test data used in various types of experiments in this paper. Furthermore, we provide some basic conceptual explanations. The training data refers to the pre-sampled synthetic data, which includes pairs of instances used for training. These pairs consist of re-parameterized synthetic graphs and sample data tables obtained through forward sampling based on conditional probability tables. As for the test data, we directly retrieve the causal graph structure and parameterized conditional probability tables from bnlearn. Then, the data obtained directly through forward sampling is referred to as observed samples. Otherwise, we can conduct various types of intervention experiments, including single-node intervention or multi-node intervention (*i.e.*, intervening on one or multiple nodes at a time), soft intervention or hard intervention (*i.e.*, whether to eliminate dependencies of parent nodes; for soft intervention, we replace probability distributions sampled from the Dirichlet distribution with parameters $\alpha \sim U[0.2, 1.0]$), and known intervention or unknown intervention targets (*i.e.*, whether we have prior knowledge of intervention target nodes). For the starting point of IS-MCMC sampling, we default to selecting JCI+BLIP as the proxy algorithm to obtain the initial graph seed. We use the `xgb.XGBClassifier()` API provided by `Scikit-learn`[3] as the classifier for both skeleton and orientation models. Different threshold parameters are set for skeleton identification and orientation identification, while all other hyper-parameters are set to default values. The code is available at https://github.com/microsoft/reliableAI/tree/main/causal-kit/TICL.

# F. Additional Experiments (RQ4)

## F.1. Effect of Intervention Type

### F.1.1. PREFECT INTERVENTIONS

We follow the default parameter settings, replacing soft interventions with hard interventions, *i.e.*, perfect interventions, to eliminate the dependency of the parent node. The results of $\mathcal{I}$-CPDAG structure discovery are shown in Table 10, and the results of intervention target identification are shown in Table 11. The results indicate that our method consistently outperforms on almost all datasets and metrics.

---

[3]https://scikit-learn.org/stable/

*Table 10.* Performance comparison of $\mathcal{I}$-*CPDAG* under perfect intervention  (F1 Score ↑ / SHD ↓ / SID ↓).

| DATASETS #NODES / #EDGES | METRIC | TICL | JCI-GOLEM | JCI-PC | JCI-BLIP | JCI-HC | AVICI* | ENCO* | IGSP* | GIES | UT-IGSP |
|---|---|---|---|---|---|---|---|---|---|---|---|
| EARTHQUAKE 5 / 4 | F1 SCORE | 1.00 | 0.00 | 1.00 | 1.00 | 0.00 | 0.29 | 0.67 | 1.00 | 1.00 | 1.00 |
| | SHD | 0 | 6 | 0 | 0 | 4 | 4 | 2 | 0 | 0 | 0 |
| | SID | 0 | 13 | 0 | 0 | 20 | 10 | 2 | 0 | 0 | 0 |
| SURVEY 6 / 6 | F1 SCORE | 1.00 | 0.40 | 0.40 | 0.73 | 0.73 | 0.00 | 0.44 | 0.91 | 0.91 | 0.91 |
| | SHD | 0 | 7 | 4 | 2 | 2 | 6 | 4 | 1 | 1 | 1 |
| | SID | 0 | 23 | 18 | 9 | 9 | 17 | 16 | 4 | 4 | 4 |
| ASIA 8 / 8 | F1 SCORE | 0.86 | 0.00 | 0.40 | 0.86 | 0.12 | 0.00 | 0.22 | 0.77 | 0.71 | 0.43 |
| | SHD | 2 | 14 | 8 | 2 | 13 | 8 | 10 | 3 | 4 | 6 |
| | SID | 12 | 36 | 34 | 12 | 47 | 34 | 28 | 10 | 18 | 24 |
| SACHS 11 / 17 | F1 SCORE | 0.79 | 0.00 | 0.44 | 0.79 | 0.71 | / | 0.25 | 0.56 | 0.51 | 0.56 |
| | SHD | 6 | 20 | 17 | 6 | 8 | / | 36 | 13 | 18 | 13 |
| | SID | 27 | 61 | 30 | 27 | 29 | / | 35 | 28 | 34 | 32 |
| CHILD 20 / 25 | F1 SCORE | 0.76 | 0.24 | 0.25 | 0.75 | 0.65 | / | 0.07 | 0.49 | 0.38 | 0.53 |
| | SHD | 10 | 34 | 45 | 11 | 16 | / | 105 | 29 | 53 | 29 |
| | SID | 160 | 301 | 233 | 188 | 226 | / | 210 | 125 | 140 | 112 |
| INSURANCE 27 / 52 | F1 SCORE | 0.72 | 0.15 | 0.47 | 0.71 | 0.47 | / | 0.11 | 0.37 | 0.39 | 0.37 |
| | SHD | 22 | 73 | 64 | 24 | 43 | / | 171 | 77 | 96 | 78 |
| | SID | 325 | 671 | 414 | 390 | 421 | / | 536 | 442 | 316 | 452 |
| WATER 32 / 66 | F1 SCORE | 0.58 | 0.14 | – | 0.50 | 0.30 | / | 0.24 | / | / | / |
| | SHD | 42 | 77 | – | 44 | 71 | / | 80 | / | / | / |
| | SID | 419 | 566 | – | 457 | 642 | / | 508 | / | / | / |

*Table 11.* Performance comparison of *Intervention Targets Detection* under perfect intervention  (F1 Score ↑).

| DATASETS | EARTHQUAKE | SURVEY | ASIA | SACHS | CHILD | INSURANCE | WATER |
|---|---|---|---|---|---|---|---|
| | 1 | 1 | 2 | 2 | 4 | 5 | 6 |
| UT-IGSP | 0.40 | 0.50 | 0.44 | 0.40 | 0.36 | 0.22 | / |
| CITE | 1.00 | 1.00 | 1.00 | 0.57 | 0.40 | 0.55 | / |
| PREDITER | 0.67 | / | 1.00 | 0.67 | – | – | / |
| JCI-GOLEM | 0.67 | 1.00 | 0.00 | 0.18 | 0.22 | 0.28 | 0.21 |
| JCI-HC | 0.50 | 1.00 | 0.36 | 0.67 | 0.47 | 0.34 | 0.32 |
| JCI-BLIP | 1.00 | 1.00 | 0.80 | 0.80 | 0.53 | 0.44 | 0.42 |
| JCI-PC | 1.00 | 0.67 | 0.67 | 0.47 | 0.47 | 0.45 | – |
| TICL | 1.00 | 1.00 | 1.00 | 1.00 | 1.00 | 1.00 | 1.00 |

### F.1.2. MULTIPLE INTERVENTIONS

Next, we consider conducting multi-node interventions, where for each intervention experiment, we randomly select 1 to 3 nodes for intervention, with the selection of nodes being without replacement, while the rest follow the default parameter settings. The results of $\mathcal{I}$-CPDAG structure discovery are shown in Table 12, and the results of intervention target identification are shown in Table 13. The results indicate that our method still maintains a consistent lead in almost all datasets and metrics. Additionally, it is worth noting that in the case of interventions on multiple nodes simultaneously, we observed that the performance does not necessarily improve compared to single-node interventions, which may be due to the complexity introduced by intervening on multiple nodes at the same time.

*Table 12.* Performance comparison of $\mathcal{I}$-*CPDAG* under multiple interventions (F1 Score ↑ / SHD ↓ / SID ↓).

| DATASETS #NODES / #EDGES | METRIC | TICL | JCI-GOLEM | JCI-PC | JCI-BLIP | JCI-HC | IGSP* | GIES | UT-IGSP |
|---|---|---|---|---|---|---|---|---|---|
| EARTHQUAKE 5 / 4 | F1 SCORE | 0.60 | 0.00 | 0.55 | 1.00 | 0.25 | 1.00 | 0.40 | 1.00 |
| | SHD | 4 | 5 | 4 | 0 | 6 | 0 | 7 | 0 |
| | SID | 9 | 18 | 3 | 0 | 15 | 0 | 6 | 0 |
| SURVEY 6 / 6 | F1 SCORE | 0.83 | 0.00 | 0.40 | 0.91 | 0.83 | 0.91 | 0.00 | 0.91 |
| | SHD | 2 | 9 | 4 | 1 | 1 | 1 | 6 | 1 |
| | SID | 10 | 30 | 18 | 4 | 6 | 4 | 30 | 4 |
| ASIA 8 / 8 | F1 SCORE | 0.86 | 0.08 | 0.56 | 0.95 | 0.56 | 0.46 | 0.44 | 0.77 |
| | SHD | 2 | 9 | 7 | 1 | 7 | 5 | 7 | 3 |
| | SID | 8 | 33 | 17 | 5 | 21 | 20 | 25 | 10 |
| SACHS 11 / 17 | F1 SCORE | 0.94 | 0.16 | 0.56 | 0.90 | 0.54 | 0.51 | 0.32 | 0.72 |
| | SHD | 2 | 18 | 12 | 3 | 12 | 14 | 19 | 10 |
| | SID | 8 | 61 | 38 | 21 | 48 | 35 | 43 | 15 |
| CHILD 20 / 25 | F1 SCORE | 0.61 | 0.18 | 0.26 | 0.60 | 0.44 | 0.60 | 0.33 | 0.51 |
| | SHD | 14 | 44 | 45 | 15 | 24 | 25 | 47 | 29 |
| | SID | 243 | 308 | 229 | 251 | 304 | 72 | 170 | 117 |
| INSURANCE 27 / 52 | F1 SCORE | 0.73 | 0.16 | 0.36 | 0.69 | 0.52 | 0.34 | 0.45 | 0.33 |
| | SHD | 22 | 63 | 72 | 27 | 38 | 81 | 85 | 81 |
| | SID | 369 | 624 | 472 | 398 | 425 | 470 | 276 | 453 |
| WATER 32 / 66 | F1 SCORE | 0.62 | 0.20 | – | 0.60 | 0.24 | / | / | / |
| | SHD | 39 | 81 | – | 42 | 71 | / | / | / |
| | SID | 436 | 605 | – | 478 | 655 | / | / | / |

*Table 13.* Performance comparison of *Intervention Targets Detection* under multiple interventions (F1 Score ↑).

| DATASETS | EARTHQUAKE | SURVEY | ASIA | SACHS | CHILD | INSURANCE | WATER |
|---|---|---|---|---|---|---|---|
| | 3 | 3 | 6 | 6 | 4 | 5 | 12 |
| UT-IGSP | 0.86 | 0.86 | 0.67 | 0.57 | 0.80 | 0.24 | / |
| CITE | 0.80 | 1.00 | 0.67 | 0.43 | 0.33 | 0.71 | / |
| PREDITER | 0.80 | 0.50 | / | 0.29 | – | – | / |
| JCI-GOLEM | 0.33 | 0.80 | 0.62 | 0.47 | 0.21 | 0.14 | 0.23 |
| JCI-HC | 0.57 | 1.00 | 0.67 | 0.67 | 0.50 | 0.33 | 0.38 |
| JCI-BLIP | 1.00 | 1.00 | 0.86 | 0.86 | 0.44 | 0.42 | 0.57 |
| JCI-PC | 0.80 | 1.00 | 0.92 | 0.75 | 0.50 | 0.33 | – |
| TICL | 0.80 | 1.00 | 1.00 | 0.86 | 1.00 | 0.86 | 1.00 |

## F.2. Effect of Sample Size

### F.2.1. LIMITED INTERVENTIONAL DATA SAMPLE SIZES

Intervention experiments in the real world are often unrealistic or costly, such as gene knockout experiments, where only a small amount of experimental data may be available. Therefore, we further consider the case of limited intervention data sample size. Specifically, we reduce the sample size of each intervention experiment from 10,000 to 1,000 to simulate this scenario, while keeping other parameters at their default settings. The results of $\mathcal{I}$-CPDAG structure discovery are shown in Table 14, and the results of intervention target identification are shown in Table 17. The results indicate that most methods show a certain degree of performance decline. Nevertheless, our method still maintains a significant lead on almost all datasets and metrics, demonstrating the robustness and superiority of our TICL approach.

### F.2.2. LIMITED OBSERVATIONAL DATA SAMPLE SIZES

Similarly, we also consider the case of limited sample size of observational data. Specifically, we reduce the sample size of the observational data from 10,000 to 2,000 to simulate this scenario, while keeping the rest of the parameters at their default settings. The results of $\mathcal{I}$-CPDAG structure discovery are shown in Table 16, and the results of intervention target

*Table 14.* Performance comparison of $\mathcal{I}$-*CPDAG* under limited interventional data sample sizes (F1 ↑ / SHD ↓ / SID ↓).

| DATASETS #NODES / #EDGES | METRIC | TICL | JCI-GOLEM | JCI-PC | JCI-BLIP | JCI-HC | AVICI* | ENCO* | IGSP* | GIES | UT-IGSP |
|---|---|---|---|---|---|---|---|---|---|---|---|
| EARTHQUAKE 5 / 4 | F1 SCORE | 1.00 | 0.00 | 0.89 | 1.00 | 1.00 | 0.00 | 0.67 | 1.00 | 1.00 | 1.00 |
| | SHD | 0 | 7 | 1 | 0 | 1 | 4 | 2 | 0 | 0 | 0 |
| | SID | 0 | 12 | 0 | 0 | 0 | 10 | 2 | 0 | 0 | 0 |
| SURVEY 6 / 6 | F1 SCORE | 0.73 | 0.00 | 0.33 | 0.73 | 0.00 | 0.00 | 0.60 | 0.91 | 0.91 | 0.91 |
| | SHD | 2 | 11 | 6 | 2 | 7 | 5 | 3 | 1 | 1 | 1 |
| | SID | 9 | 30 | 22 | 9 | 30 | 17 | 14 | 4 | 4 | 4 |
| ASIA 8 / 8 | F1 SCORE | 0.80 | 0.20 | 0.33 | 0.80 | 0.33 | 0.00 | 0.70 | 0.77 | 0.71 | 0.77 |
| | SHD | 3 | 11 | 7 | 3 | 11 | 8 | 6 | 3 | 4 | 3 |
| | SID | 8 | 38 | 38 | 19 | 34 | 37 | 9 | 10 | 18 | 10 |
| SACHS 11 / 17 | F1 SCORE | 0.74 | 0.00 | 0.44 | 0.21 | 0.38 | 0.08 | 0.17 | 0.67 | 0.32 | 0.62 |
| | SHD | 7 | 23 | 17 | 15 | 13 | 17 | 41 | 11 | 21 | 12 |
| | SID | 31 | 62 | 42 | 55 | 48 | 60 | 47 | 24 | 40 | 30 |
| CHILD 20 / 25 | F1 SCORE | 0.80 | 0.16 | 0.25 | 0.75 | 0.54 | 0.56 | 0.14 | 0.51 | 0.47 | 0.49 |
| | SHD | 10 | 41 | 39 | 10 | 20 | 16 | 96 | 28 | 36 | 28 |
| | SID | 168 | 315 | 277 | 185 | 249 | 188 | 192 | 128 | 125 | 135 |
| INSURANCE 27 / 52 | F1 SCORE | 0.64 | 0.06 | 0.37 | 0.60 | 0.57 | 0.17 | 0.12 | 0.30 | 0.53 | 0.38 |
| | SHD | 25 | 65 | 66 | 25 | 37 | 51 | 162 | 82 | 58 | 75 |
| | SID | 395 | 677 | 544 | 404 | 423 | 625 | 539 | 451 | 246 | 438 |
| WATER 32 / 66 | F1 SCORE | 0.47 | 0.19 | – | 0.40 | 0.45 | / | 0.47 | / | / | / |
| | SHD | 47 | 71 | – | 51 | 52 | / | 53 | / | / | / |
| | SID | 496 | 545 | – | 540 | 398 | / | 473 | / | / | / |

*Table 15.* Performance comparison of *Intervention Targets Detection* under limited interventional data sample sizes (F1 ↑).

| DATASETS | EARTHQUAKE | SURVEY | ASIA | SACHS | CHILD | INSURANCE | WATER |
|---|---|---|---|---|---|---|---|
| | 1 | 1 | 2 | 2 | 4 | 5 | 6 |
| UT-IGSP | 0.33 | 1.00 | 0.57 | 1.00 | 0.36 | 0.27 | / |
| CITE | 1.00 | 0.67 | 0.67 | 1.00 | 0.40 | 0.22 | / |
| PREDITER | 0.67 | 1.00 | / | 0.40 | 0.33 | – | / |
| JCI-GOLEM | 0.00 | 1.00 | 0.00 | 0.27 | 0.15 | 0.11 | 0.00 |
| JCI-HC | 1.00 | 0.67 | 0.50 | 0.80 | 0.40 | 0.37 | 0.36 |
| JCI-BLIP | 1.00 | 1.00 | 0.67 | 0.67 | 0.40 | 0.44 | 0.40 |
| JCI-PC | 1.00 | 1.00 | 0.67 | 0.80 | 0.44 | 0.32 | – |
| TICL | 1.00 | 1.00 | 1.00 | 1.00 | 1.00 | 0.91 | 1.00 |

identification are shown in Table 17. The results are similar, indicating the importance of observational data for causal structure identification. We also see that our method is still in a good leading position.

## F.3. Effect of Intervention Experiment Size

Finally, we investigated the impact of different intervention trial frequencies on causal structure discovery. Specifically, we set intervention frequencies at 0%, 20%, 40%, and 80% of the target graph nodes, where 0% means inferring causal structure solely from observational data. We conducted experiments on the Child dataset (20 nodes) using seven methods, including ours, and the results of I-CPDAG structure discovery at different intervention frequencies are shown in Figure 14. The experiments align with intuition, showing that with an increasing number of experiments, all methods exhibit improvements in various metrics, with our method standing out more prominently.

*Table 16.* Performance comparison of $\mathcal{I}$-*CPDAG* under limited observational data sample sizes  (F1 ↑ / SHD ↓ / SID ↓).

| DATASETS #NODES / #EDGES | METRIC | TICL | JCI-GOLEM | JCI-PC | JCI-BLIP | JCI-HC | AVICI* | ENCO* | IGSP* | GIES | UT-IGSP |
|---|---|---|---|---|---|---|---|---|---|---|---|
| EARTHQUAKE 5 / 4 | F1 SCORE | 0.86 | 0.00 | 0.86 | 1.00 | 1.00 | 0.17 | 0.57 | 1.00 | 0.00 | 1.00 |
| | SHD | 1 | 6 | 1 | 0 | 0 | 8 | 3 | 0 | 4 | 0 |
| | SID | 1 | 16 | 1 | 0 | 0 | 13 | 2 | 0 | 16 | 0 |
| SURVEY 6 / 6 | F1 SCORE | 0.73 | 0.00 | 0.40 | 0.73 | 0.29 | 0.00 | 0.25 | 0.50 | 0.36 | 0.50 |
| | SHD | 2 | 12 | 4 | 2 | 6 | 7 | 5 | 4 | 4 | 4 |
| | SID | 11 | 30 | 18 | 9 | 29 | 22 | 18 | 11 | 20 | 11 |
| ASIA 8 / 8 | F1 SCORE | 0.77 | 0.00 | 0.29 | 0.67 | 0.53 | 0.38 | 0.71 | 0.77 | 0.40 | 0.77 |
| | SHD | 4 | 10 | 7 | 4 | 5 | 8 | 4 | 3 | 6 | 3 |
| | SID | 18 | 46 | 39 | 18 | 31 | 34 | 14 | 10 | 42 | 10 |
| SACHS 11 / 17 | F1 SCORE | 0.45 | 0.09 | 0.63 | 0.21 | 0.36 | 0.27 | 0.25 | 0.65 | 0.35 | 0.47 |
| | SHD | 12 | 17 | 13 | 15 | 14 | 14 | 29 | 9 | 22 | 12 |
| | SID | 49 | 54 | 24 | 55 | 53 | 57 | 45 | 18 | 37 | 39 |
| CHILD 20 / 25 | F1 SCORE | 0.75 | 0.20 | 0.17 | 0.62 | 0.75 | / | 0.11 | 0.69 | 0.38 | 0.65 |
| | SHD | 10 | 44 | 50 | 17 | 11 | / | 110 | 16 | 53 | 17 |
| | SID | 181 | 290 | 258 | 266 | 186 | / | 220 | 62 | 139 | 99 |
| INSURANCE 27 / 52 | F1 SCORE | 0.61 | 0.20 | 0.38 | 0.62 | 0.52 | / | 0.15 | 0.54 | 0.41 | 0.56 |
| | SHD | 29 | 69 | 64 | 30 | 38 | / | 127 | 40 | 89 | 39 |
| | SID | 424 | 630 | 538 | 451 | 453 | / | 540 | 364 | 347 | 428 |
| WATER 32 / 66 | F1 SCORE | 0.50 | 0.21 | – | 0.49 | 0.35 | / | 0.40 | / | / | / |
| | SHD | 44 | 81 | – | 46 | 61 | / | 57 | / | / | / |
| | SID | 507 | 620 | – | 510 | 532 | / | 527 | / | / | / |

*Table 17.* Performance comparison of *Intervention Targets Detection* under limited observational data sample sizes  (F1 ↑).

| DATASETS | EARTHQUAKE | SURVEY | ASIA | SACHS | CHILD | INSURANCE | WATER |
|---|---|---|---|---|---|---|---|
| | 1 | 1 | 2 | 2 | 4 | 5 | 6 |
| UT-IGSP | 0.33 | 0.67 | 0.40 | 0.67 | 0.81 | 0.24 | / |
| CITE | / | 1.00 | / | 1.00 | 0.33 | 0.38 | / |
| PREDITER | / | 1.00 | / | / | 0.40 | – | / |
| JCI-GOLEM | 0.00 | 0.00 | 0.00 | 0.27 | 0.15 | 0.27 | 0.20 |
| JCI-HC | 1.00 | 0.67 | 0.44 | 0.50 | 0.57 | 0.37 | 0.30 |
| JCI-BLIP | 1.00 | 1.00 | 0.67 | 0.67 | 0.44 | 0.44 | 0.37 |
| JCI-PC | 1.00 | 0.67 | 0.40 | 0.57 | 0.29 | 0.30 | – |
| TICL | 1.00 | 0.67 | 1.00 | 1.00 | 0.86 | 0.59 | 1.00 |

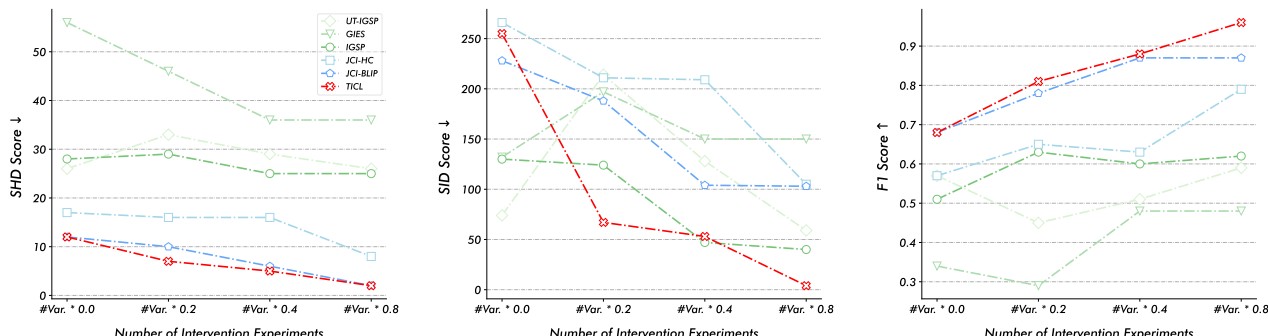

*Figure 14.* Exploration of the impact of different size of intervention experiments on performance

# G. More Discussion / Limitation / Future Work

### G.1. Supervised Causal Discovery

We advocate for causal discovery from intervention data in supervised paradigm. Identifiable causal relationships between causal graphs can be treated as ground truth labels, and corresponding data can be obtained through forward sampling. Importantly, this sampling strategy is cost-free, allowing us to obtain an infinite number of samples for training a model to compress the mapping between data and causal relationships. As the main study on data quantity, typically, the more data, the better performance. Furthermore, although we can freely generate parameterized causal graphs, for specific systems of interest, such as the aforementioned study on data quality, we consider training data derived using posterior distributions to be useful for causal discovery of the data of interest.

### G.2. Self-augmentation *vs.* Pre-training

We noticed that all the SCL methods in the literature, during the training phase, integrate or generate datasets containing various causal mechanisms and their related data samples through simulators (if available), however, such generation mechanisms are entirely dependent on manual specification. Ideally, it is preferable to predefine rich synthetic graph structures (possible examples include ER / SF / Low Rank / Random graphs, etc.) and synthetic function types (including linear, quadratic, nonlinear, etc.) as much as possible. We refer to the process of synthesizing data during the training phase and then training them together as "pre-training", similar to today's language model pre-training, using massive training data to achieve generalization in different domains. However, causal structure learning is a high-risk field, where people often pay more attention to performance. The potential danger of this pre-training method is that if the causal mechanisms of the system of interest have never been seen before, there may be out-of-distribution generalization issues. Therefore, in this paper, we advocate acquiring training data more relevant to the test data after accessing the test data, providing high-quality training data by observing the posterior estimates of causal graphs to avoid the "domain shift" problem. We call this self-augmentation method, similar to in-context learning, adjusting on specific testing instance of interest in order to achieve high performance that is usable. As discussed in the TTT (Gandelsman et al., 2022) literature, fixed models can introduce significant bias in new testing scenarios, which is particularly challenging for causal structure learning due to the combinatorial diversity of interventions. In contrast, learning each test instance from scratch results in extremely high variance. TICL occupies this critical middle ground: it reduces the bias from static training by generating adaptive data tailored to the test distribution while controlling variance through a surrogate algorithm. That said, a complete theoretical treatment of generalization in causal structure learning—incorporating structure, samples, and interventions—is highly challenging and beyond our current scope. We consider this a direction for future research.

### G.3. Discrete *vs.* Continuous Data

In this paper, we mainly focus on causal structure identification from discrete data. The *Markov completeness* theorem states that for discrete or linear Gaussian data, we can only identify causal graphs up to their CPDAG. For continuous data satisfying linear non-Gaussian mechanisms or additive noise assumptions, we can identify more causal directions. However, our goal is to properly handle intervention data by using a supervised learning paradigm to learn identifiable causal structures. Therefore, more considerations may be needed to design corresponding learning tasks for continuous data. Nevertheless, in practice, our method can be naturally extended to continuous data by utilizing standard algorithms from existing literature, such as the Hilbert-Schmidt Independence Criterion (Gretton et al., 2007) to compute conditional dependencies of numerical variables. Despite this, more attention may need to be paid, which we consider as future work. In addition, like many other methods, TICL relies on standard causal discovery assumptions to obtain theoretical guarantees and focuses on the TTT of SCL. Further relaxation of these assumptions is left for future work.

