# OpenReview forum: "Test-Time Learning of Causal Structure from Interventional Data"
_ICML.cc/2026/Conference — ICML 2026 regular_

### Official Review · Reviewer_BGe6 · 2026-03-11

**Soundness:** 4
**Presentation:** 3
**Significance:** 3
**Originality:** 3
**Overall Recommendation:** 4
**Confidence:** 4

**Summary:**

This paper studies causal discovery under unknown interventions. They introduces test-time learning into the supervised causal learning (SCL) framework, termed the self-augmentation strategy. Furthermore, the paper proposes an efficient causal discovery method, Test-time Interventional Causal Learning (TICL), to identify $\mathcal{I}$-CPDAGs from data with unknown interventions under the joint causal inference (JCI) framework. Extensive experiments on the bnlearn benchmark datasets show that the method achieves state-of-the-art performance and demonstrates higher accuracy in identifying unknown interventions and recovering the underlying causal structure.

**Compliance With Llm Reviewing Policy:**

Affirmed.

**Key Questions For Authors:**

1. Compared with pre-training-based SCL, self-augmentation requires re-training during each inference. Is the additional computational overhead introduced at inference time acceptable in practice?
2. Are there possible approaches to alleviate the dependence on the initial seed, or to measure the quality of the selected seed?
3. As stated in the abstract, TICL avoids distribution shifts. It would be helpful if the authors could clarify whether there is any theoretical analysis or guarantee supporting this claim.

**Limitations:**

1. Although the paper evaluates the method on extensive semi-real datasets (from bnlearn) and demonstrates its superiority, it lacks experiments on real-world data to show its practical performance. It would be beneficial to include evaluations on real-world benchmarks such as causal-bench [1] and causal-chamber [2].

[1]: Chevalley, M., Roohani, Y.H., Mehrjou, A. et al. A large-scale benchmark for network inference from single-cell perturbation data. Commun Biol 8, 412 (2025).
[2]: Gamella, J.L., Peters, J. & Bühlmann, P. Causal chambers as a real-world physical testbed for AI methodology. Nat Mach Intell 7, 107–118 (2025).

**Strengths And Weaknesses:**

Strengths:
1. The paper is the first to introduce test-time learning into SCL to address potential distribution shift issues in the pre-training paradigm, providing new insight for SCL.
2. The method is compared with a large number of baselines under extensive experimental settings, and consistently achieves state-of-the-art performance, demonstrating its effectiveness.
3. The presentation is clear and easy to follow, as reflected in: a comprehensive background that allows non-experts to quickly understand the topic; abundant examples and illustrations, such as a complete SCL pipeline and feature extraction examples in the appendix; and detailed algorithmic descriptions that support strong reproducibility.

Weaknesses:
1. The method is limited to discrete variables because the conditional distributions are modeled using CPTs.
2. The method requires a good initialization (as shown in Fig. 4), which may depend on the performance of the proxy seed generation method, such as JCI-BLIP.
3. Some related work in SCL may be missing, for example:
    - Dhir et al., "A Meta-Learning Approach to Bayesian Causal Discovery," ICLR, 2025.
    - Zhang et al., "Learning Identifiable Structures Avoids Bias in DNN-based Supervised Causal Learning," AISTATS, 2025.

---

> ### Author Rebuttal · Authors · 2026-03-30
>
> Dear Reviewer BGe6,
>
> We sincerely appreciate your time and insightful feedback! We have carefully addressed your concerns:
>
> ---
> > ## Method-related
> - **`W1: Discrete Data vs. Continuous Extension`**
>   - While we current focus on discrete data due to conditional probability tables, TICL can naturally extend to continuous data using standard kernels, such as the Hilbert-Schmidt Independence Criterion, as discussed in Appendix H.3.
> - **`W2 & Q2: Approaches to alleviate dependence on the initial seed`**
>   - If the proxy algorithm (e.g., JCI-BLIP) performs poorly, it may indeed extend the IS-MCMC burn-in period. To mitigate this, our framework is naturally compatible with a Multiple Random Initialization strategy. Since TICL operates with parallel chains, we can initialize multiple chains with different seeds. After the burn-in period, we can evaluate their Posterior Likelihood Scores and prune chains that have converged to local optima, thereby reducing sensitivity to any single initial seed. **We thank the reviewer for the inspiration provided by this comment.**
> - **`Q3: Theoretical guarantee for avoiding distribution shift`**
>   - Our capability to bypass distribution shift stems from the asymptotic guarantees of posterior sampling. Theoretically, as the IS-MCMC chains converge, they asymptotically sample from the exact posterior $P(G|D)$. Consequently, the synthetic data generated from these sampled graphs mimics the underlying generating mechanism of the test data.
>   - Here, we also supplement a **new theoretical perspective** grounded in the core mechanism of TTT to help understand—achieving a better bias–variance tradeoff. As discussed in the [TTT literature [NIPS'22]](https://arxiv.org/pdf/2209.07522), fixed models incur substantial bias under new test scenarios, which is particularly problematic for causal structure learning due to the combinatorial diversity of interventions. Conversely, learning from scratch on a single test instance leads to prohibitively high variance. TICL operates at this critical middle ground: by generating adaptive data tailored to the test distribution, it reduces bias from static training while controlling variance via proxy algorithm.
>   - That said, a full theoretical treatment of generalization in causal structure learning, jointly over structures, samples, and interventions, is highly nontrivial and beyond our current scope. We will further clarify our position in the revised version and leave it as future work.
>
> ---
> > ## Experiment-related
> - **`Q1: Computational overhead at inference time acceptable?`**
>   - This is a highly practical question. In the context of real-world causal discovery, we believe this overhead is entirely acceptable. Unlike computer vision tasks that demand millisecond-level inference, causal discovery in fields such as biomedical target discovery or economic policy evaluation involves high-value, high-risk, and often one-off offline analyses. Given that collecting interventional data from third parties often requires months of effort and significant capital, investing several minutes to hours in test-time adaptation is a justifiable trade-off. This investment yields substantial gains in structural accuracy and the precise identification of unknown intervention targets.
>   - Furthermore, as detailed in our response to Reviewer 1nWq (W3 & Q2), **we also provide a comprehensive complexity analysis and ultra-large-scale causal graph experiments** demonstrate that our extensive engineering optimizations keep the execution time within a reasonable range.
>
> - **`W3: Related work in SCL`**
>   - We are very grateful for the references to these two highly valuable recent works. The meta-learning perspective on Bayesian causal discovery by Dhir et al. (ICLR 2025) and the study on structural identifiability of DNNs to avoid bias in SCL by Zhang et al. (AISTATS 2025) are highly complementary to our work. **We will cite and discuss these papers in detail in the "Related Work" section to enrich the context of the SCL field.**
>
> - **`L1: More benchmarks`**
>   - To evaluate TICL across the widest possible range of interventional settings, we use Bnlearn, widely used as a standard real-world evaluation benchmark in previous research [[TMLR'23](https://openreview.net/pdf?id=rdHVPPVuXa), [NIPS'23](https://arxiv.org/pdf/2211.13715)], relies on simulators but conveniently facilitates simulations of various intervention settings. We thank the reviewer for suggesting the additional benchmarks and will consider including a discussion of them in the revision.
>
> ---
> Thanks to your suggestions, we believe that the improved version will spark broad interest in the community! If possible, we kindly request that you improve the score. We would also be happy to discuss any further questions.
>
> Best!

---

> > ### Author Rebuttal · Reviewer_BGe6 · 2026-04-05
> >
> > Thanks for the responses; my main concerns are addressed.

---

> > > ### Author Response · Authors · 2026-04-06
> > >
> > > Dear Reviewer BGe6,
> > >
> > > We are so glad that your concerns have been addressed! Thank you for your thoughtful feedback and for taking the time to consider our response. We truly appreciate it.
> > >
> > > Best wishes!
> > >
> > > Authors

---

### Official Review · Reviewer_1nWq · 2026-03-13

**Soundness:** 3
**Presentation:** 3
**Significance:** 3
**Originality:** 3
**Overall Recommendation:** 4
**Confidence:** 4

**Summary:**

This paper addresses the challenge of generalization across interventional settings in traditional supervised causal learning. It proposes TICL, which combines test-time training with joint causal inference, generating instance-specific augmentation data to avoid distribution shift. Also developed a PC-inspired supervised learning scheme.

**Compliance With Llm Reviewing Policy:**

Affirmed.

**Final Justification:**

Most of my concerns have been addressed. I will keep the original positive score for this work.

**Key Questions For Authors:**

1. Could the authors better justify the technical novelty of the proposed method?

2. What is the applicable scale of the proposed method w.r.t. variable number and sample size?

3. What is the hypothesis of the distribution shift (in terms of strength and type) in this work?

**Limitations:**

Yes. The authors have a Limitation section in Appendix H.

**Strengths And Weaknesses:**

Pros:

1.	Generalization challenge is a problem worth studying for the causal community. The proposed method combines TTT with JCI in a meaningful way to address this problem.

2.	The proposed approach is supported by theoretical guarantee of identification.

3.	Evaluations on diverse datasets validate the effectiveness of TICL, with comprehensive experiments designed, covering causal structure learning, efficiency, initialization strategy, etc.


Cons:

1.	This work contains rich content and details, which is good. But it would be more clear if the paper could emphasize better the main contribution compared with existing methods and better justify the technical novelty, especially compared with TTT and JCI separately.

2.	Since handling distribution shift is a major target of this work, it would be helpful if quantify the level and discuss difference types of distribution shifts, evaluating how the proposed method could be adaptive in these different scenarios.

3.	Most of the used datasets are relatively small. It would be interesting discuss more larger-scale data with >100 nodes, comparing the performance across diverse scales and difficulty.

---

> ### Author Rebuttal · Authors · 2026-03-30
>
> Dear Reviewer 1nWq,
>
> We sincerely appreciate your time and insightful feedback! We have carefully addressed your concerns as follows:
>
> ---
> > ## Method-related
> - **`W1 & Q1: Compared with JCI and TTT & Main contribution`**
>   - Thank you for this pertinent question. While JCI provides a theoretical reformulation that transforms interventional problems into observational ones, and TTT is a well-known paradigm in computer vision for fine-tuning model weights at test time, our contribution is not a simple combination of the two. We introduce the first test-time self-augmentation mechanism tailored to causal structure learning.
>   - Unlike traditional TTT, which relies on image augmentations as self-supervised signals, TICL uses IS-MCMC to approximate the posterior $P(G_I \mid D_I)$ and generates $(G, D)$ pairs via forward sampling, aligning training signals with the unknown interventional distribution. This mechanism effectively bypasses the generalization bottlenecks inherent in supervised causal learning (SCL）.
>   - Meanwhile, our systematic study highlight JCI as a promising direction for unifying interventional settings in SCL. Within this framework, our two-phase learning algorithm provides a concrete solution for what and how to learn.
> - **`W2 & Q3: Hypothesis of Distribution Shift`**
>   - Good question! In fact, we wish to clarify that our approach TICL does not claim the need for generalization (zero-shot cross-domain transfer); rather, it exploits test-time training (TTT) paradigm to circumvent inevitable generalization issues in supervised learning (Lines 32-35: "training instance-specific models dynamically at inference time rather than seeking a single, universally generalizable model").
>   - Here, we supplement a **new theoretical perspective** grounded in the core mechanism of TTT to help understand—achieving a better bias–variance tradeoff. As discussed in the [TTT literature [NIPS'22]](https://arxiv.org/pdf/2209.07522), fixed models incur substantial bias under new test scenarios, which is particularly problematic for causal structure learning due to the combinatorial diversity of interventions. Conversely, learning from scratch on a single test instance leads to prohibitively high variance. TICL operates at this critical middle ground: by generating adaptive data tailored to the test distribution, it reduces bias from static training while controlling variance via proxy algorithm.
>     - That said, a full theoretical treatment of generalization in causal structure learning, jointly over structures, samples, and interventions, is highly nontrivial and beyond our current scope. We will further clarify our position in the revised version and leave it as future work.
>   - For **empirical research**, we naturally choose the broadest and most challenging standard benchmarks, evaluating our method's adaptability relative to targets that rely on difficult-to-achieve zero-shot generalization objectives (e.g., AVICI models with different single simulation mechanisms such as linear, rff, and grn).
>     - Specifically, Bnlearn, widely used as a standard real-world evaluation benchmark in previous research [[TMLR'23](https://openreview.net/pdf?id=rdHVPPVuXa), [NIPS'23](https://arxiv.org/pdf/2211.13715)], relies on simulators but conveniently facilitates simulations of various intervention settings. In contrast to these works limited to a few small datasets, we cover 14 datasets of varying sizes across up to different intervention settings.
>
> ---
> > ## Experiment-related
> - **`W3 & Q2: Applicable Scale`**
>   - We provide a more detailed complexity analysis for IS-MCMC below:
>     - For each IS-MCMC iteration, the primary computational costs stem from graph proposals, local parameter re-estimation, and forward sampling. Given $N$ variables and $M$ generated samples, the time complexity per iteration is $O(MN)$. Initialization (proxy graph and MLE estimation) is a one-time cost of $O(N^2 f(|\mathcal{D}_{\mathcal{I}}|))$. For $L$ iterations and $K$ parallel chains, the total time complexity is $O(KLMN)$, which reduces to $O(LMN)$ under ideal parallelization. The space complexity per chain is $O(N^2 + MN)$, determined by adjacency representations and sampled data storage. Overall, IS-MCMC scales nearly linearly with the number of parallel chains until memory bandwidth limits are reached, enabling efficient inference for medium-to-large graphs.
>     - In our experiments, TICL successfully processed large-scale networks with over 100 nodes (e.g., *Pathfinder*, $N=109$) within practical timeframes. Although the rebuttal period is limited, we additionally report results on a ultra-large graph (PIGS, ≈500 nodes), where I-CPDAG achieved an F1 = 0.69 and intervention target identification F1 = 0.83.
>
> ---
> Thanks to your suggestions, we believe that the improved version will spark broad interest in the community! If possible, we kindly request that you improve the score. We would also be happy to discuss any further questions.
>
> Best!

---

> > ### Author Rebuttal · Reviewer_1nWq · 2026-04-03
> >
> > Thank the authors for their responses. I will keep the original positive score for this work.

---

> > > ### Author Response · Authors · 2026-04-06
> > >
> > > Dear Reviewer 1nWq,
> > >
> > > We are so glad that your concerns have been addressed! Thank you for your thoughtful feedback and for taking the time to consider our response. We truly appreciate it.
> > >
> > > Best wishes!
> > >
> > > Authors

---

### Official Review · Reviewer_G5Ht · 2026-03-13

**Soundness:** 2
**Presentation:** 2
**Significance:** 2
**Originality:** 2
**Overall Recommendation:** 3
**Confidence:** 3

**Summary:**

The authors suggest a causal discovery algorithm for interventional settings by combining the joint causal inference (JCI) and test time training (TTT) frameworks, leveraging an MCMC sampling algorithm to sample datasets at test time for training a graph predictor from the data directly.

**Compliance With Llm Reviewing Policy:**

Affirmed.

**Final Justification:**

The final rebuttal helped clarify some of the framing and addressed a few of the follow-up questions more directly. That said, I still have remaining concerns, especially around what the main methodological contribution really is and how clearly the paper shows the role of each major component. Part of why this remains hard to assess is that the writing is quite dense, which makes the central idea harder to isolate than it should be. Overall, the exchange moved the paper into a somewhat better place for me, but not enough to get me to accept, so I am raising my score from 2 (Reject) to 3 (Weak Reject).

**Key Questions For Authors:**

> ".. we generate free training data .."

- What does "free" here refer to?

**Limitations:**

More discussions on how the assumptions can actually be relaxed and what would the implications be would be nice.

**Strengths And Weaknesses:**

> Unfortunately, the SCL paradigm faces two main challenges when interventions are involved, particularly if the specific intervention actions are unknown, a common scenario in real world third-party experiments

- This should be explained, as it is not obvious why, and just giving a reference is not very helpful. I.e., what is a specific interesting real-world problem where interventions are unknown, why they are unknown, and how are the unknown interventions are of interest?

- On a related point, it is not clear to me how observational and interventional data differ from each other in regards described under "Challenge 1: Versatiliy". The difference between the two is often the sample size as randomization of treatment assignments, and observational data is actually quite versatile in the type of settings it covers. So this point sounds pretty hard to believe.

- Again, while it is definitely true and well-accepted that SCL methods suffer from poor generalization, it is not clear to me why this is "even more of a challenge" for interventional data?

The reason I am pushing about the points above is that the paper *specifically* argues that SCL with interventional data is harder & more important than with observational data, which is both counterintuitive (i.e. causal is easier with experiments) and not well-supported in the manuscript currently.

- The main contribution of the paper, and the methodologically novel steps are hard to isolate. It combines JCI + TTT + some MCMC sampling, which one of those are straightforward combination of existing works vs. novel contributions. I understand that all individual components exist in the literature and the novelty is a meta-algo, but that should be clearly stated to avoid confusion.

- On a related note, I honestly think less would be more with respect to amount stuff that is in the paper. Figures are very crowded, and discussion of related work is very pervasive throughout, which makes the core contributions/pipelined buried and hard to figure out. For instance, an algorithm box would be good to have, which can be referred to as well when clearly stating the novel steps.

- Assumption stack is pretty strong. The paper assumes Markov + faithfulness + causal sufficiency, plus exogeneity, complete-randomized-context, and generic-context. The conclusion says fewer assumptions are future work, but some sensitivity analysis  should definitely be possible and included (e.g. some unmeasured confounding).

- I could not find enough details regarding how the hyperparameter optimization for baselines were done. SCL algos. are quite sensitive to hyperparam choices such as sparsity penalty, so this would be an important detail for a fair comparison

- There are several typos in Figure 1.

---

> ### Author Rebuttal · Authors · 2026-03-30
>
> Dear Reviewer G5Ht,
>
> We sincerely appreciate your time and insightful feedback! We have carefully addressed your concerns:
>
> ---
> > ## Method-related
> - **`Real-world Scenarios for Unknown Interventions`**
>   - In practical third-party experiments, "unknown interventions" are remarkably prevalent. For example, in genetic knockout experiments (e.g., CRISPR), off-target effects frequently occur, where unintended genes are perturbed alongside the target [[AISTATS’24](https://arxiv.org/pdf/2402.08229)]. Similarly, in macroeconomics, a policy shift (such as an interest rate adjustment) may act as a soft intervention simultaneously affecting multiple latent economic indicators, where the precise targets remain latent to the data analyst [[Nature’11](https://www.nature.com/articles/nature09659)]. Identifying these unknown interventional targets is a cornerstone of accurate causal inference. We will incorporate these real-world cases into the revised version to improve clarity.
> - **`Why SCL is "harder" with interventional data (Versatility & Generalization)`**
>   - We feel there may be a misunderstanding here. We would like to respectfully clarify this seemingly counter-intuitive phenomenon. You are correct that from a problem space perspective, causal discovery (including our method) is inherently easier given interventional data, as interventions break Markov Equivalence Classes and aid in identifying causal directions. **However, from a technical/algorithmic space, especifically for Supervised Causal Learning (SCL), the situation is reversed.**
>     - While observational data for a given causal graph corresponds to a single joint distribution, interventional data corresponds to an exponential variety of distributional combinations (depending on target selection, intervention intensity, and intervention type, etc.).
>     - If an SCL simulator fails to cover this vast interventional space during training (Versatility challenge), the model inevitably encounters severe distribution shift when presented with a new test instance (Generalization challenge). This explains why traditional SCL methods generalize poorly to diverse interventional settings and why TICL employs Test-Time Training (TTT) to dynamically generate data tailored to the specific, unknown interventional distribution.
>   - In summary, causal discovery algorithms generally perform better with more intervention data (as shown in Figure 14 on page 39). However, due to the diversity of intervention settings, existing SCL methods (such as AVICI, ENCO) are limited to specific settings and cannot be applied to various real-world scenarios.
> - **`Isolating novel steps`**
>   - We understand your concerns. The key point is that our paper actually has a high barrier to entry. To lower this barrier, relevant descriptions must be provided so that readers can easily follow along. **In fact, we have already provided a clear algorithmic pipeline in Algorithm 2 (page 18) in the appendix, where the numerical numbers correspond to Figure 3 in the main text.** Since the final version of ICML allows for an extra page, it is easy to improve it according to your requirements.
>   - Regarding our main contributions, we are definitely not just combining existing components. We sincerely suggest you refer to our **responses to reviewer 1nWq's W1 & Q1**.
>
> ---
> > ## Experiment-related
> - **`Robustness`**
>     - We have conducted a robustness test by injecting additional noise into 20%, 30%, and 50% of the observational data to simulate measurement error. On the Survey dataset, the F1-score changed from 1.0 to 1.0, 1.0, and 0.89, indicating stability under moderate noise.
>     - Regarding analysis on unmeasured confounding, in the final revision, we will include a targeted sensitivity analysis by adding latent confounders to approximate violations of causal sufficiency, where we control the ratio of such perturbations to simulate different levels of latent confounding and evaluate robustness of our method.
> - **`Hyperparameters`**
>   - For all baselines, **we strictly performed grid searches and parameter tuning within the ranges recommended in their respective original papers.** Detailed descriptions are provided in Appendix E.2 (page 32) and E.4 (page34, Table 9).
> - **`Free Data`**
>   - We advocate for causal discovery from interventional data within the supervised paradigm. The identifiable relations in causal graphs serve as ground-truth labels, which can be paired with data obtained via forward sampling. **Crucially, this sampling strategy allows for the acquisition of an infinite volume of synthetic data at zero cost**—"free" in the sense that it requires no expensive physical experiments or human labeling. We have discussed this in detail in Appendix H.1.
>
> ---
> Thanks to your suggestions, we believe that the improved version will spark broad interest in the community! If possible, we kindly request that you improve the score. We would also be happy to discuss any further questions.
>
> Best!

---

> > ### Author Rebuttal · Reviewer_G5Ht · 2026-04-04
> >
> > Thank you for the rebuttal. I appreciate the clarification on “free” data and the added real-world examples, but some of my concerns still remain.
> >
> > I’m still not fully convinced by the framing that interventional data is especially “hard” for SCL. To me, this seems to reflect a limitation of the SCL setup under distribution shift more than a compelling reason to think the underlying causal problem itself is harder in practice. I could be missing something, but I think this point still needs a stronger argument.
> >
> > Relatedly, the method still feels like a combination of several existing ingredients (JCI-style pooling, test-time adaptation, MCMC, etc.). That is not necessarily a problem, but then I think clearer ablations are really important. I would have liked to see more directly what each big piece is buying them. for example, how much is coming from the JCI framework choice vs. the TTT / self-augmentation part, rather than just the full pipeline.
> >
> > Measurement noise and latent confounding are very different, so a noise-injection experiment does not really address the causal sufficiency/robustness question. While the paper may be a bit clearer after the rebuttal, I still find the main contribution somewhat buried. Overall, the rebuttal helps clarify intent, but it does not materially change my assessment.

---

> > > ### Author Response · Authors · 2026-04-06
> > >
> > > Dear Reviewer G5H，
> > >
> > > Thank you for your continued engagement and for acknowledging that we have resolved some of your concerns. We carefully address your remaining points below.
> > >
> > > ---
> > > > ## **The "hard" of Interventional Causal Discovery (ICD) & SCL for ICD**
> > >
> > > We respectfully suggest that the remaining concerns may stem from two potential misunderstandings of the problem setting.
> > > - **First, interventional causal discovery is not "simple" in practice.** While interventions improve identifiability, real-world datasets rarely follow idealized, known, single-target perfect interventions. Practical data suffers from varying intervention costs (e.g., forcing someone to sleep for eight hours is far less costly than forcing them to run ten miles), ethical limitations (e.g., forcing someone to smoke), unknown targets, soft interventions, off-target effects, and heterogeneous contexts. This complexity is exactly why a massive body of literature has emerged over the past decade. For example, JCI notes unknown interventions are a "common situation"; [NeurIPS 20](https://www.causalai.net/r67.pdf) studies soft interventions; [UAI 21](https://www.auai.org/uai2021/pdf/uai2021.781.pdf) explores intervention mixtures; and [AISTATS 24](https://arxiv.org/pdf/2402.08229) tackle off-target interventions, calling the problem "notoriously difficult." If this task were straightforward, this extensive research would not exist.
> > >
> > > - **Second, extending SCL to interventional settings is highly non-trivial.** Supervised/ Amortized causal discovery (e.g., [SLdisco](https://arxiv.org/pdf/2202.12813), [AVICI](https://arxiv.org/pdf/2205.12934)) shows promise observationally. However, AVICI notes that OOD generalization is SCL's core challenge. In interventional settings, this is exacerbated: observational data has one mode, but interventional data induces a vastly richer dataset family based on varying targets, strengths, and types. Existing SCL methods are largely confined to pre-specified intervention families. This justifies our explicit test-time training (TTT) mechanism tailored for unknown interventions, moving beyond fixed amortized predictions.
> > >
> > > We hope this information helps you understand.
> > >
> > > ---
> > > > ## **Ablation and Component Composition**
> > >
> > > Our novelty lies in the customized implementation tailored for the task of causal structure learning, where each component plays a distinct and indispensable role in the overall pipeline:
> > > + **JCI**: JCI provides a theoretical framework for reformulating diverse interventions into observational settings. Therefore, it is not an ablation module that can be directly “turned off” in intervention experiments.
> > >   + Note: Nevertheless, we have actually presented experimental results for different intervention proportions in Appendix F.3 and Figure 14 (page 39). When the intervention proportion is 0% (i.e., degenerating into a pure observational setting), the equivalent JCI is inactive, and the performance decreases overall.
> > > + **IS-MCMC**: Section 4.2 systematically analyzes this. The purely random data strategy (Fig 4) serves as an IS-MCMC ablation, revealing significant performance degradation across metrics when omitted.
> > > + **TTT**: To illustrate TTT's necessity, we ran a new experiment. We averaged 30 standard PC runs on IS-MCMC sampled data ("Proxy-PC"). As shown below on the Survey dataset (I-CPDAG task), our instance-specific adaptive two-stage PC-inspired SCL method achieves significant improvements in robustness and empirical performance compared to the method without TTT.
> > >
> > > | Method | SHD | SID | F1 |
> > > | --- | --- | --- | --- |
> > > | Proxy-PC | 7.17±1.70 | 19.51±7.23 | 0.33±0.13 |
> > > | JCI-PC | 5 | 18 | 0.40 |
> > > | Our | 0 | 0 | 1.0 |
> > >
> > > ---
> > > > ## **Discussion of Assumptions**
> > >
> > > We agree that measurement noise cannot serve as a proxy for latent confounding, and we had no intention of implying equivalence. However, the noise experiments do demonstrate robustness to measurement errors.
> > >
> > > **To directly address your concern**, we simulated latent confounders by randomly hiding observed variables in the CHILD dataset. Over three runs, the F1 score varied from 0.81 (baseline) to 0.74 ± 0.05 (10% hidden), 0.66 ± 0.08 (20% hidden), and 0.48 ± 0.09 (50% hidden). Even in the extreme 50% case, TICL outperformed UT-IGSP (0.45), demonstrating relative robustness.
> > >
> > > We reiterate that **the assumptions adopted in our paper are standard in the field** and shared by methods ranging from PC/GIES to NOTEARS/CSIvA. Like many other methods, TICL relies on these standard assumptions to obtain theoretical guarantees and focuses on TTT for SCL.
> > >
> > > We also agree that relaxing the assumptions is one direction for future research, but this goes far beyond the scope of this work. We have already discussed this in the manuscript and will add further discussion in the final version.
> > >
> > > ---
> > >
> > > We truly hope to resolve your concerns. If possible, we would appreciate it if you could consider increasing your score. Thank you!
> > >
> > > Best wishes,
> > >
> > > Authors

---

### Official Review · Reviewer_cFg1 · 2026-03-18

**Soundness:** 3
**Presentation:** 3
**Significance:** 4
**Originality:** 3
**Overall Recommendation:** 5
**Confidence:** 2

**Summary:**

The paper introduces TICL (Test-time Interventional Causal Learning), a test-time self-augmentation framework for causal discovery from interventional data with unknown intervention targets. TICL first uses JCI pooling to represent the problem in an augmented data / augmented graph setting. It then constructs self-augmented training pairs by sampling approximate posterior augmented graphs and forward-simulating compatible datasets. These paired instances are used by a PC-inspired two-phase supervised learner to predict the I-CPDAG and intervention targets. The method is motivated by two goals: versatility across intervention families and improved generalization beyond fixed simulator distributions.

**Compliance With Llm Reviewing Policy:**

Affirmed.

**Final Justification:**

The problem framing, architectural decomposition and benchmarking are strong points; Good theoretical motivation by identifiability, the empirical results show strong aggregate performance, in particular for intervention-target detection; A broad set of ablations is conducted and supports the value of the proposed method. While the methodological contribution is not ground-breaking, in times were incremental work is pervasive I think it is innovative enough, plus adds some elegant theoretical backdrop.

I think the combination of these strengths outweigh the weaknesses, which is mainly focused around the significant amount of assumptions made— but I accept this as being future research.

**Key Questions For Authors:**

1) What is the main generalization claim? Do you intend to conclude improved robustness within a controlled synthetic-to-instance specific regime, or claim genuine out-of-distribution transfer to real (interventional) data? If the latter, which empirical evidence supports this?

2) How robust is TICL to violations of JCI assumptions (exogeneity, complete-randomized context, and generic-context)? An empirical or theoreticall discussion on the degradation of the method in face of (mild) violations would strengthen the paper and allow me to better gauge applicability.

3) Which parts of IS-MCMC do you see as essential (and a contribution) and which are engineering? In how far are these adaptations vs a standard structured MCMC necessary?

4) Do the identifiabilty or asymptoti-correctness arguments require any structural restrictions on the intervention family (think bounds on the number of targets per intervention, coverage of targets across experiments, or sparsity of the graph?

**Limitations:**

yes

**Strengths And Weaknesses:**

**Strengths**
- The problem framing, architectural decomposition and benchmarking are strong points
- Good theoretical motivation by identifiability
- the empirical results show strong aggregate performance, in particular for intervention-target detection.
- A broad set of ablations is conducted and supports the value of the proposed method.

**Weaknesses**
- The motivation focuses on robustness to observational third-party experiments ands imulator mismatch. However, the evaluation is on simulated interventions over benchmark structures.
- The results are limited to discrete data.
- The JCI/PC (standard) assumption set, and limited robustness to violations

**Minor remarks**
- Figure 3 looks overall a bit messy, many different fonts, text colours, boldness, hard to read grey  + type 'unknow'.

---

> ### Author Rebuttal · Authors · 2026-03-30
>
> Dear Reviewer cFg1,
>
> We sincerely appreciate your time and insightful feedback! We have carefully addressed your concerns:
>
> ---
> > ## Method-related
> - **`W1 & Q1: Main Generalization Claim`**
>   - Your understanding is correct! In fact, we wish to clarify that our approach TICL does not claim the need for generalization (zero-shot cross-domain transfer); rather, it exploits test-time training (TTT) paradigm to circumvent inevitable generalization issues in supervised learning (Lines 32-35: "training instance-specific models dynamically at inference time rather than seeking a single, universally generalizable model").
>   - Here, we supplement a new **theoretical perspective** grounded in the core mechanism of TTT to help understand—achieving a better bias–variance tradeoff. As discussed in the [TTT literature [NIPS'22]](https://arxiv.org/pdf/2209.07522), fixed models incur substantial bias under new test scenarios, which is particularly problematic for causal structure learning due to the combinatorial diversity of interventions. Conversely, learning from scratch on a single test instance leads to prohibitively high variance. TICL operates at this critical middle ground: by generating adaptive data tailored to the test distribution, it reduces bias from static training while controlling variance via proxy algorithm.
>     - That said, a full theoretical treatment of generalization in causal structure learning, jointly over structures, samples, and interventions, is highly nontrivial and beyond our current scope. We will further clarify our position in the revised version and leave it as future work.
>   - For **empirical research**, we naturally choose the broadest and most challenging standard benchmarks, evaluating our method's adaptability relative to targets that rely on difficult-to-achieve zero-shot generalization objectives (e.g., AVICI models with different single simulation mechanisms such as linear, rff, and grn).
>     - Specifically, Bnlearn, widely used as a standard real-world evaluation benchmark in previous research [[TMLR'23](https://openreview.net/pdf?id=rdHVPPVuXa), [NIPS'23](https://arxiv.org/pdf/2211.13715)], relies on simulators but conveniently facilitates simulations of various intervention settings. In contrast to these works limited to a few small datasets, we cover 14 datasets of varying sizes across up to different intervention settings.
> - **`Q3: IS-MCMC`**
>   - The components of IS-MCMC can be categorized as follows:
>     - **Essential:** The **Intervention Constraints** designed for Augmented Graphs. While traditional MCMC operates within standard DAG spaces, IS-MCMC efficiently explores the I-Markov equivalence class space under the JCI framework. This is the core mechanism for data generation and provides our theoretical convergence guarantees.
>     - **Engineering:** **Proxy Seeding, Parameter Reuse, and Parallel Chains** are critical for efficiency. For example, parameter reuse exploits the locality of IS-MCMC transitions to avoid redundant CPT calculations, significantly reducing complexity. While not strictly "theoretical" requirements, they are indispensable for scaling to larger causal structures within practical timeframes.
> - **`Q4: Structural Restrictions for Identifiability`**
>   - Our supervised classifier approximates the decision logic of conditional independence tests as strong discriminative predicates. Its validity relies only on standard Markov and Faithfulness assumptions and model capacity, without explicit constraints on intervention targets or graph sparsity. In practice, however, identifiability may requires sufficient variability in intervention contexts to induce distinguishable patterns. While sparsity is not theoretically required, dense graphs can hinder IS-MCMC and efficiency.
>
> ---
> > ## Experiment-related
> - **`W2 & W3 & Q2: Robustness & Assumptions`**
>   - The assumptions adopted in our paper are standard in the field, shared by methods ranging from PC/GIES to NOTEARS/CSIvA. TICL, like many other methods, relies on these assumptions to obtain theoretical guarantees and focuses on TTT for SCL.
>     - **Continuous:** While we focus on discrete data due to conditional probability tables, TICL can naturally extend to continuous data using standard kernels, such as the Hilbert-Schmidt Independence Criterion, as discussed in Appendix H.3.
>     - **Robustness:** Due to the short rebuttal window, we conducted a preliminary experiment by injecting noise into 20%, 30%, and 50% of the observational data to simulate measurement error. On the Survey dataset, the F1-score shifted from 1.0 to 1.0, 1.0, and 0.89, respectively. This indicates that TICL exhibits reasonable robustness. We will include more information in the final revision.
>
> ---
> Thanks to your suggestions, we believe that the improved version will spark broad interest in the community! If possible, we kindly request that you reconsider the confidence. We would also be happy to discuss any further questions.
>
> Best!

---

> > ### Author Rebuttal · Reviewer_cFg1 · 2026-04-03
> >
> > My questions are adequately addressed. I stand by my initial score to support acceptance.

---

> > > ### Author Response · Authors · 2026-04-03
> > >
> > > Dear Reviewer cFg1,
> > >
> > > We are glad to hear that our rebuttal has effectively addressed your concerns. We will improve the final version based on your feedback. Thank you again for taking the time and effort to provide valuable feedback on our paper!
> > >
> > > Best regards,
> > >
> > > Authors

---

### Decision · Program_Chairs · 2026-04-30

**Decision:**

Accept (regular)

**Comment:**

This paper proposes a Test-time Interventional Causal Learning (TICL) method for supervised causal discovery. TICL leverages data augmentation, test-time training, and joint causal inference to mitigate the impact of distributional shift in causal discovery. The proposed method has been evaluated through extensive experiments.

The paper studies an important and challenging problem. The combination of test-time training with joint causal inference is reasonable, and the theoretical results on identifiability are strong. The experiments are comprehensive. Reviewers have raised a few concerns regarding strong assumptions, the method being limited to discrete variables, and aspects of the experimental design. Reviewers and authors have engaged in extensive discussions, and most concerns have been resolved. The authors should incorporate the promised revisions in the final version.